# Evaluating acute image ordering for real-world patient cases via language model alignment with radiological guidelines

Michael S. Yao [1,2], Allison Chae [2], Piya Saraiya[3], Charles E. Kahn Jr.[2,3], Walter R. Witschey [2,3], James C. Gee [3], Hersh Sagreiya [2,3,5] & Osbert Bastani [4,5] ✉

## Abstract

**Background** Diagnostic imaging studies are increasingly important in the management of acutely presenting patients. However, ordering appropriate imaging studies in the emergency department is a challenging task with a high degree of variability among healthcare providers. To address this issue, recent work has investigated whether generative AI and large language models can be leveraged to recommend diagnostic imaging studies in accordance with evidence-based medical guidelines. However, it remains challenging to ensure that these tools can provide recommendations that correctly align with medical guidelines, especially given the limited diagnostic information available in acute care settings.

**Methods** In this study, we introduce a framework to intelligently leverage language models by recommending imaging studies for patient cases that align with the American College of Radiology's Appropriateness Criteria, a set of evidence-based guidelines. To power our experiments, we introduce RadCases, a dataset of over 1500 annotated case summaries reflecting common patient presentations, and apply our framework to enable state-of-the-art language models to reason about appropriate imaging choices.

**Results** Using our framework, state-of-the-art language models achieve accuracy comparable to clinicians in ordering imaging studies. Furthermore, we demonstrate that our language model-based pipeline can be used as an intelligent assistant by clinicians to support image ordering workflows and improve the accuracy of acute image ordering according to the American College of Radiology's Appropriateness Criteria.

**Conclusions** Our work demonstrates and validates a strategy to leverage AI-based software to improve trustworthy clinical decision-making in alignment with expert evidence-based guidelines.

## Plain language summary

Emergency room doctors often need to quickly decide which medical scans, such as X-rays or CT scans, to order for patients. However, these decisions can vary significantly among doctors. In this study, we looked at whether generative artificial intelligence (AI) can help recommend which scans patients should receive. We created a dataset of real patient cases to help AI tools follow expert medical guidelines when suggesting scans. Our results show that AI tools can accurately choose the right scans for patients and can also be helpful assistants for clinicians. In the future, we hope this work can support faster, more accurate decision-making and reduce unnecessary tests.

Ordering diagnostic imaging studies is an increasingly common task in the emergency department (ED) and other acute-care settings, and is associated with high cognitive burden for clinicians[1–4]. While diagnostic imaging can play a crucial role in the acute workup of patients, ordering imaging studies with limited clinical utility has raised concerns regarding resource utilization, radiation exposure, and financial burden to both patients and healthcare systems[5–7]. Recent estimates suggest that up to 30% of diagnostic

imaging studies ordered in the ED setting could be replaced with more appropriate alternatives at the time the order was placed[6,8].

Multiple factors contribute to the challenge of ordering appropriate and clinically indicated imaging studies. Importantly, emergency medicine physicians often need to make rapid diagnostic decisions with limited clinical context while simultaneously managing high patient volumes and complex patient presentations[9,10]. Furthermore, there is considerable variability

[1]Department of Bioengineering, University of Pennsylvania, Philadelphia, PA, USA. [2]Perelman School of Medicine, University of Pennsylvania, Philadelphia, PA, USA. [3]Department of Radiology, University of Pennsylvania, Philadelphia, PA, USA. [4]Department of Computer and Information Science, University of Pennsylvania, Philadelphia, PA, USA. [5]These authors jointly supervised this work: Hersh Sagreiya, Osbert Bastani. ✉e-mail: obastani@seas.upenn.edu

between healthcare providers in imaging ordering patterns: recent studies have documented significant differences in the utilization rates of different imaging studies between individual emergency physicians, suggesting that factors beyond strict clinical necessity influence imaging decisions[11–15].

To help clinicians make more informed, evidence-based decisions in image ordering and simultaneously address inter-provider variability in imaging practices, the American College of Radiology (ACR) released the ACR Appropriateness Criteria® (ACR AC), which are a set of evidence-based guidelines that assist referring physicians in ordering the most appropriate diagnostic imaging studies for specific clinical conditions[16]. As of June 2024, the ACR AC contains 224 unique imaging topics (i.e., patient scenarios).

However, despite the widespread availability of the ACR AC, low utilization of these guidelines remains a challenge in many emergency departments and inpatient settings[3,17]. Bautista et al. showed that there is low utilization of the ACR AC by clinicians in practice: less than 1% of physicians interviewed in their study use the ACR AC as a first-line resource when ordering diagnostic imaging studies[18,19]. The limited usage of the ACR AC may be partly due to how the Appropriateness Criteria are made accessible to clinicians; the evidence-based criteria are dense and can be difficult to interpret even for physician experts—especially in acute healthcare settings such as the emergency department where decision making is both time-sensitive and critical.

To address this problem, recent work has investigated the potential utility of generative artificial intelligence (AI) tools to synthesize dense passages of evidence-based guidelines to offer clinical decision support (CDS) in physician workflows[20–26]. In particular, large language models (LLMs) are generative AI models trained on large corpora of textual data to achieve impressive performance on tasks such as language translation, summarization, and text generation[27–30]. However, despite the success of these models in natural language tasks, LLMs have been shown to struggle in challenging domain-specific tasks requiring human expertise and specialized training, such as in medicine, law, and engineering[31–36]. As a result, accessing the potential benefits of LLMs in these domains—such as for recommending appropriate imaging studies for patients—continues to be an ongoing challenge, deterring widespread adoption of generative AI models in clinical medicine[37,38].

Prior work has examined the ability of LLMs to rapidly process and contextualize large volumes of information to help make the ACR Appropriateness Criteria more accessible for real-time CDS. For example, Nazario-Johnson et al.[20] and Zaki et al.[21] evaluate the alignment of LLMs with the ACR Appropriateness Criteria; however, both studies leverage inputs that are not representative of the vernacular used in real-world clinical workflows. Other studies, including Savage et al.[22]; Kim et al.[39]; Jin et al.[40]; Rau et al.[41]; and Krithara et al.[42], work with more realistic examples of real-world patient descriptions; however, these LLM inputs either (1) assume that all relevant medical information is provided to make a diagnostic decision; or (2) are phrased as multiple-choice questions. Neither of these characteristics are representative of how clinicians might use LLMs for CDS in practice, especially in acute emergency medicine settings that are notably characterized by incomplete patient information. Finally, Liu et al.[43]; Zhang et al.[44]; Zhang et al.[45]; Zhang et al.[46]; and Singhal et al.[47] introduce a number of performant models for medical tasks; however, these models again assume access to a relatively complete picture of the patient's clinical status and past medical history, which is rarely the case for acutely presenting patients in the emergency department.

In this work, we investigate how state-of-the-art LLMs can be used as CDS tools to help clinicians order guideline-recommended imaging studies according to the ACR AC. In Fig. 1, we illustrate this problem by demonstrating how a state-of-the-art language model, such as Claude Sonnet-3.5, fails to accurately recommend appropriate imaging that align with the ACR AC for a variety of input patient descriptions. Given these initial findings, we hypothesize that while LLMs may struggle to directly recommend imaging studies for patients (a domain-specific task), they may excel at describing patient conditions and presentations (phrased as ACR AC Topics)[47–49]. In this light, we apply LLMs to analyze patient case summaries and map them to topic categories from the ACR AC. We represent these case summaries as "patient one-liners," which are concise summaries of patient presentations commonly used by clinicians to communicate relevant details quickly to other healthcare providers[50,51]. Importantly, our dataset of patient one-liners are representative of both the vernacular and limited patient context available in real-world text written by clinicians. Given a patient one-liner, we can then programmatically query the ACR AC based on the LLM-recommended topic category (without any explicit LLM usage) to

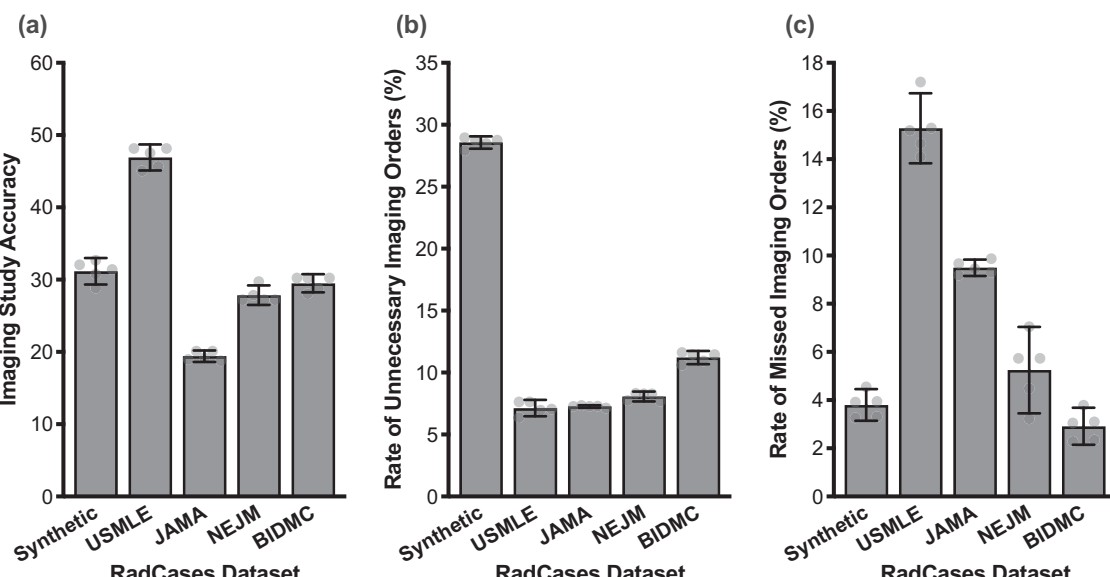

**Fig. 1 | Large Language Models Struggle with Diagnostic Imaging Ordering.** We evaluate Claude Sonnet-3.5, a state-of-the-art language model, on its ability to order imaging studies given an input patient case description, or "one-liner." The LLM is evaluated on five representative subsets of the RadCases dataset introduced in our work. To demonstrate the difficulty of ordering diagnostic imaging studies in practice, we show that **a** Claude Sonnet-3.5 frequently orders imaging studies that are not aligned with the ACR Appropriateness Criteria. **b** The language model also frequently orders unnecessary imaging studies, and **c** can incorrectly forego imaging even when it is clinically warranted. In our work, we introduce an LLM inference strategy to improve the performance of language models according to these important clinical metrics. Error bars represent ± 95% CI over $n$ = 5 independent experimental runs.

determine the optimal imaging study for a patient. We show how this proposed method can enable LLMs to recommend diagnostic imaging studies according to expert medical guidelines.

In this study, we first introduce RadCases, the first publicly available dataset of one-liners labeled by the most relevant ACR AC panel and topic. Secondly, we evaluate publicly available, state-of-the-art LLMs on our RadCases dataset and demonstrate how existing tools may be used out-of-the-box for diagnostic imaging support in inpatient settings. Our results suggest that generalist models such as Claude Sonnet-3.5 and Meta Llama 3 can accurately predict ACR AC Topic labels given patient one-liners. We then assess how popular techniques such as model fine-tuning (MFT), retrieval-augmented generation (RAG), in-context learning (ICL), and chain-of-thought prompting (COT) may be effectively leveraged to improve the alignment of existing LLMs with ACR AC, and also enable LLMs to exceed clinician performance in imaging order accuracy in a retrospective study[52,53]. Finally, we conduct a prospective clinical study to show that LLM clinical assistants can improve the accuracy of image ordering by clinicians in simulated acute care environments.

## Methods
### Constructing a benchmarking dataset for image ordering using language models

Prior work in medical natural language processing has primarily focused on tasks such as documentation writing, medical question answering, and chatbot-clinician alignment. In each of these tasks, a relatively complete picture including hospital course, laboratory results, and advanced image studies of a patient presentation is often available. This is not representative of the typically limited patient history to guide acute image ordering in the emergency department. To simulate clinical decision-making contexts with limited patient information, we first curated a dataset of one-liners, which are succinct patient scenario descriptions used by clinicians in medical communication, and corresponding ground-truth labels. We call this resource the "RadCases Dataset" and detail its construction below.

To build the RadCases dataset, we leveraged five publicly available, retrospective sources of textual data. First, we prompted the GPT-3.5 (gpt-3.5-turbo-0125) LLM from OpenAI to generate 16 Synthetic patient cases with a chief complaint corresponding to each of the 11 particular ACR AC Panels in diagnostic radiology. The patient cases in this dataset do not use any real patient data; use of this generated dataset in our study was therefore exempted by the University of Pennsylvania Institutional Review Board (Protocol #856530).

To include more challenging patient cases, we also introduced the Medbullets patient cases consisting of challenging United States Medical Licensing Examination (USMLE) Step 2- and 3- style cases introduced by Chen et al.[54] The original Medbullets dataset consisted of paragraph-form patient cases accompanied by a multiple-choice question; to convert each question to a patient one-liner, we extracted the first sentence of each patient case. Because the Medbullets dataset consists of synthetic patient cases not involving real patient data, use of this dataset in our study was exempted by the University of Pennsylvania Institutional Review Board (Protocol #856530).

Similarly, we leveraged the JAMA Clinical Challenge and NEJM Case Record datasets that include challenging, real-world cases published in the Journal of the American Medical Association (JAMA) and the New England Journal of Medicine (NEJM), respectively. These patient cases are often described as atypical presentations of complex diseases that are noteworthy enough to be published as resources for the broader medical community. The JAMA Clinical Challenge (respectively, NEJM Case Record) dataset was initially introduced by Chen et al.[54] (resp., Savage et al.[22]); we follow the same protocol as for the Medbullets dataset described above to programmatically convert these paragraph-form patient cases into patient one-liners by using the first sentence of each patient case. The authors of each case report were responsible for obtaining informed consent from patients under their local IRB or ethics committee, in accordance with the editorial policies of JAMA and NEJM, which require documented patient consent as a

condition of publication. Given that these case reports were published with patient consent and are publicly available in peer-reviewed journals, and no individual-level identifiers are present in the one-liner summaries, the use of these source datasets in RadCases was exempted by the University of Pennsylvania Institutional Review Board (Protocol #856530).

Finally, we sought to evaluate LLMs on patient summaries written by clinicians in a real-world emergency department. We constructed the BIDMC dataset from anonymized, de-identified patient admission notes introduced by Johnson et al.[55] in the MIMIC-IV dataset by extracting the first sentence of each clinical note as the patient one-liner. Briefly, the original MIMIC-IV dataset includes electronic health record data of patients from the Beth Israel Deaconess Medical Center (BIDMC) admitted to either the emergency department or an intensive care unit (ICU) between 2008 and 2019[55]. Referencing Johnson et al.[55], the Institutional Review Board at the BIDMC granted a waiver of informed consent and approved the sharing of the de-identified MIMIC-IV dataset for research purposes, and the use of this dataset in our study was therefore exempted by the University of Pennsylvania Institutional Review Board (Protocol #856530). We restrict our constructed one-liner dataset to those from the discharge summaries of 100 representative patients.

A patient one-liner was excluded from evaluation if any of the following exclusionary criteria applied: (1) the ACR AC did not provide any guidance for the chief complaint (e.g., a primary dermatologic condition); (2) an appropriate imaging study was performed and/or a diagnosis was already made; (3) the one-liner lacked sufficient information about the patient; or (4) the one-liner did not refer to a specific patient presentation (e.g., one-liners extracted from epidemiology-related USMLE practice questions). Of the original 2513 patient cases, a total of 914 (36.3%) cases were excluded due to the above criteria (719 excluded cases due to criteria (1); 90 due to criteria (2); 49 due to criteria (3); and 56 due to criteria (4)); see Supplementary Table 12 for additional details. All 1599 remaining one-liners in our final dataset were individually reviewed to ensure they were representative of true clinical one-liners in practice by one U.S. attending radiologist and two U.S. medical students.

### Formulating a strategy for LLM evaluation using the RadCases dataset

Our curated RadCases dataset comprises 1599 patient one-liner scenarios constructed from five different sources representing a diverse panel of patient presentations and clinical scenarios: (1) RadCases-Synthetic (156 out of the 1599 total patient cases); (2) RadCases-USMLE (170 patient cases); (3) RadCases-JAMA (965 patient cases); (4) RadCases-NEJM (163 patient cases); and (5) RadCases-BIDMC (145 patient cases).

Our next task was to annotate ground-truth labels to each of the patient scenarios in the RadCases dataset. An intuitive ground-truth label might be to assign a single "best" imaging study (or lack thereof) to order for each patient scenario. However, a single definitive ground-truth label often does not exist: imaging studies can vary significantly based on clinician preferences—even amongst expert physicians[56–58]—and available hospital resources. Furthermore, the ultimate goal of the RadCases dataset is to align LLMs with evidence-based guidelines for image ordering; data points consisting solely of patient scenario-imaging study pairs arguably contain weak, implicit signals on the underlying guidelines that dictate the "best" imaging study. In light of these challenges, we instead labeled the patient one-liners according to the most relevant ACR Appropriateness Criteria Topic. As of June 2024, there are 224 possible diagnostic radiology Topic labels. We investigate the impact of using different versions of the ACR Appropriateness Criteria in Supplementary Fig. 13.

To characterize the baseline alignment of language models with the ACR Appropriateness Criteria, we evaluated 6 state-of-the-art, publicly available LLM models on their ability to predict the most relevant ACR AC Topic based on an input patient one-liner (Fig. 2). The following models were evaluated: (1) DBRX Instruct (databricks/dbrx-instruct) from Databricks is an open-source mixture-of-experts (MoE) model with 132B total parameters[59]. (2) Llama 3 70B Instruct (meta-llama/Meta-Llama-3-

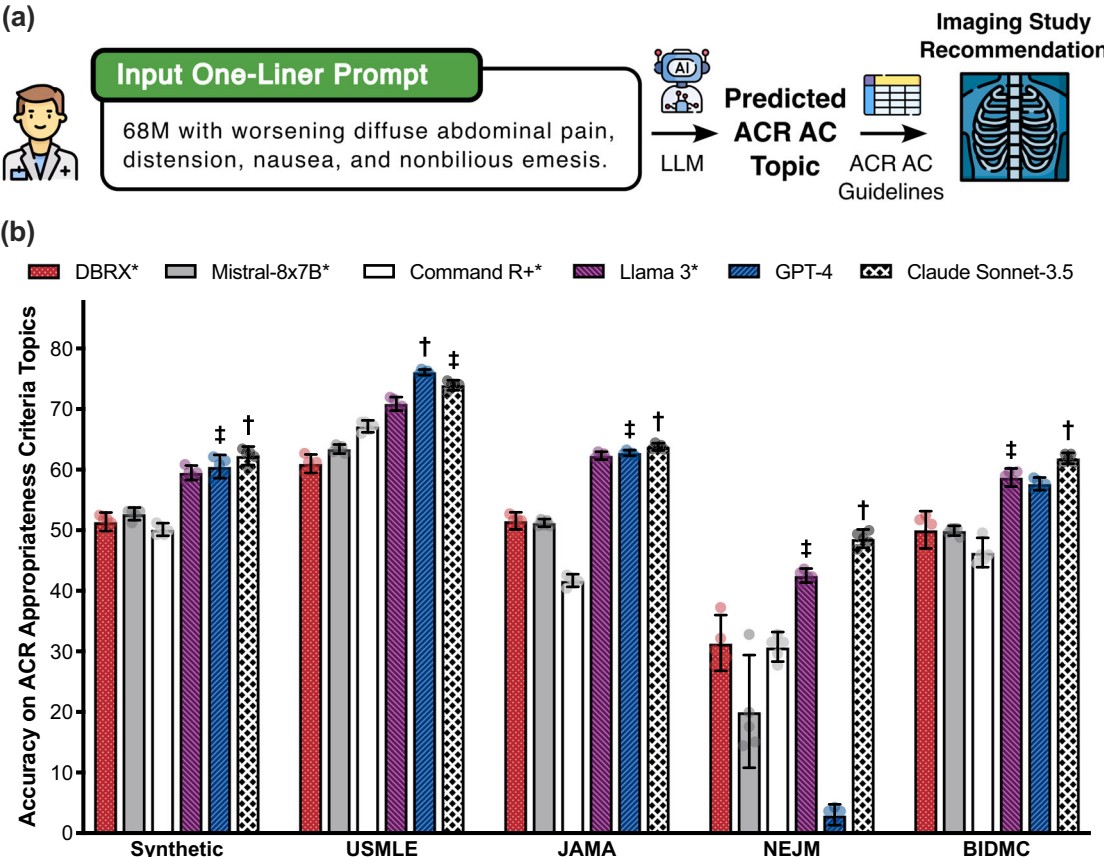

**Fig. 2 | Baseline LLM performance on the RadCases dataset. a** To align LLMs with the evidence-based ACR Appropriateness Criteria (AC), we query a language model to return the most relevant diagnostic radiology ACR AC Topic (224-way classification task) given an input patient one-liner description. We then programmatically query the ACR AC to deterministically return the most appropriate diagnostic imaging study (or lack thereof) given the predicted topic. **b** We evaluate six language models on their ability to correctly identify the ACR AC Topic most relevant to a patient one-liner. Open-source models are identified by an asterisk, and the best (second best) performing model for a RadCases dataset partition is identified by a dagger (double dagger). Error bars represent ± 95% CI over *n* = 5 independent experimental runs. *Icons of doctor, language model, table, and X-ray in (**a**) courtesy of Freepik, used with permission under Premium Flaticon License.*

70B-Instruct) from Meta AI is an open-source LLM with 70B total parameters[60]. (3) Mistral 8 × 7B Instruct (mistralai/Mixtral-8x7B-Instruct-v0.1) from Mistral AI is an open-source sparse MoE model with 47B total parameters[61]. (4) Command R+ (CohereForAI/c4ai-command-r-plus) from Cohere for AI is an open-source retrieval-optimized model with 104B total parameters[62]. (5) GPT-4 Turbo (gpt-4-turbo-2024-04-09) from OpenAI and (6) Claude Sonnet-3.5 (anthropic.claude-3-5-sonnet-20240620-v1:0) from Anthropic AI are proprietary LLMs with confidential model sizes[63,64].

#### Labeling one-liners by ACR Appropriateness Criteria topics
In accordance with this plan, two fourth-year U.S. medical students—supervised by two attending radiologists—manually annotated all RadCases scenarios according to the ACR AC Topic that best described each patient case. The two medical students and radiologists discussed cases where there was disagreement between proposed annotations, and the attending radiologists' decision was final. In scenarios where multiple ACR AC Topics might apply to a single patient case, the more acute, life-threatening scenario was used as the ground-truth label. Patient cases that were not well-described by any of the available ACR AC Topics were excluded from the dataset.

#### Evaluation metrics of language models according to the ACR Appropriateness Criteria
In our experiments, we are interested in evaluation metrics that help us elucidate the performance of LLMs as CDS tools. We detail these metrics and how they are calculated below.

An LLM's accuracy is a score between 0 and 1. We evaluate two accuracy metrics in our experiments: Topic Accuracy (Figs. 2, 3) and Imaging Accuracy (Figs. 4, 5). For a given input patient case with ground truth ACR AC Topic $y$ and model prediction $y_{pred}$, the Topic Accuracy is defined as the binary indicator variable equal to 1 if $y = y_{pred}$ and 0 otherwise. Separately, suppose that according to the ACR AC, the ground truth Topic $y$ is associated with the set of clinically appropriate studies $\mathcal{K}$, and the model-predicted Topic $y_{pred}$ is associated with the set of clinically appropriate studies $\mathcal{K}_{pred}$. The Imaging Accuracy is then defined as

$$\text{Imaging Accuracy}\left(\mathcal{K}_{pred}, \mathcal{K}\right) = \frac{\left|\mathcal{K}_{pred} \cap \mathcal{K}\right|}{\left|\mathcal{K}_{pred}\right|} \quad (1)$$

Using the same notation as above, the rate of unnecessary imaging studies (i.e., false positive rate) ordered by an LLM is a score between 0 and 1 defined as the frequency of evaluated patient cases where (1) the ground truth set of appropriate studies $\mathcal{K}$ is identically equal to {No Imaging}; and (2) No Imaging is not a member of $\mathcal{K}_{pred}$. Similarly, the rate of missed imaging studies (i.e., false negative rate) ordered by an LLM is a score between 0 and 1 defined as the frequency of evaluated patient cases where (1) No Imaging is not a member of $\mathcal{K}$; and (2) $\mathcal{K}_{pred} = $ {No Imaging}. Finally, the $F_1$ score of an LLM is defined as $F_1 = \frac{2 \cdot TP}{2 \cdot TP + (FP + FN)}$, where $TP$ is the number of patient cases where the LLM orders an imaging study that is clinically indicated, $FP$ is the number of patient cases where the LLM orders

**Fig. 3 | Optimizing LLM performance on the RadCases dataset. a** We explore 4 strategies to further improve LLM alignment with the ACR AC: RAG and ICL provide additional context to an LLM as input, COT encourages deductive reasoning, and MFT optimizes the weights of the LLM itself. Each optimization strategy is independently implemented and compared against the baseline prompting results in Fig. 2 for **b** Claude Sonnet-3.5 and **c** Llama 3. Error bars represent ± 95% CI over $n = 5$ independent experimental runs. *Icons in (**a**) courtesy of Freepik, RaftelDesign, and Mehwish, used with permission under Premium Flaticon License.*

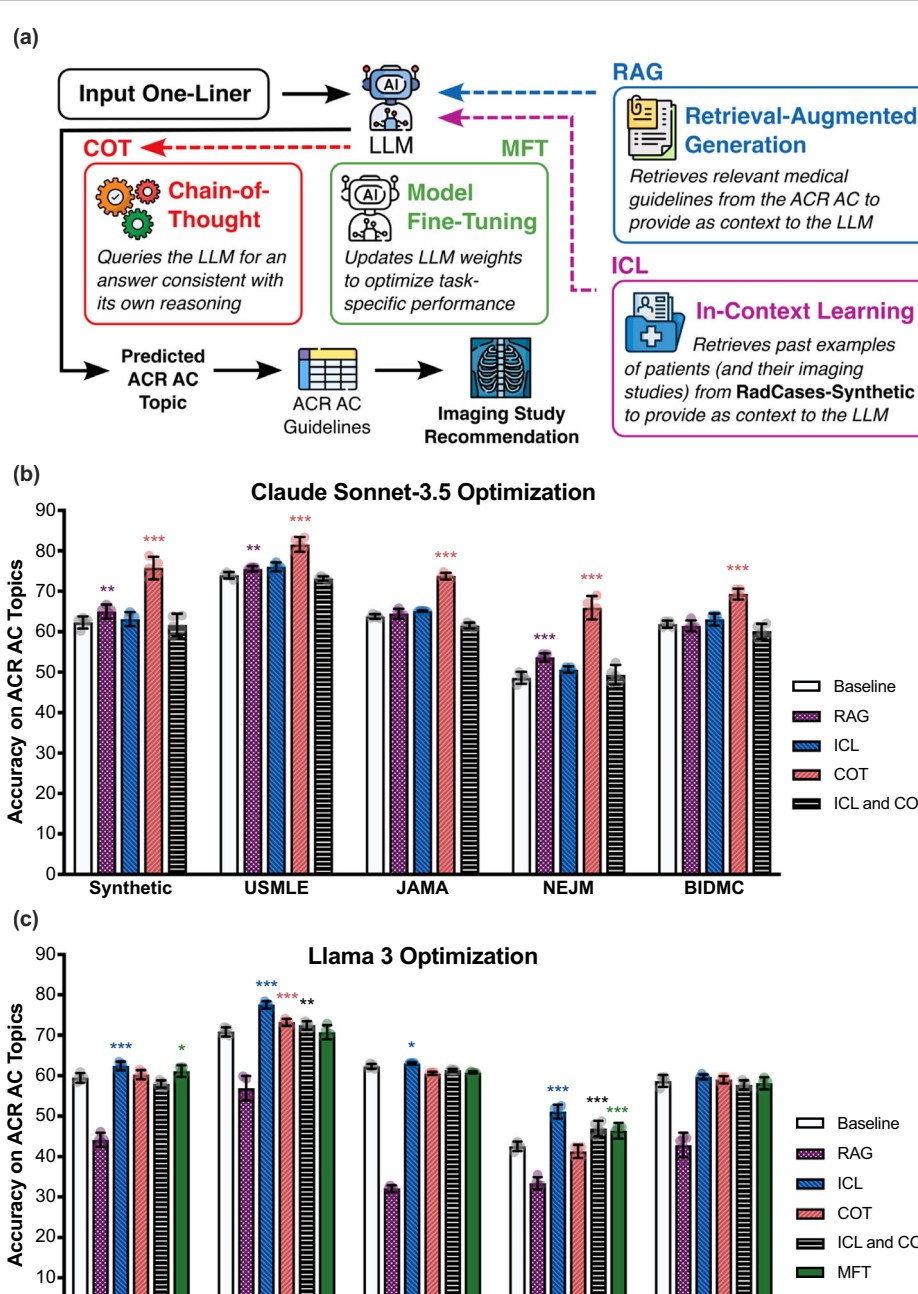

an unnecessary study, and *FN* is the number of patient cases where the LLM incorrectly fails to order an imaging study according to the guidelines.

Importantly, we highlight that the construction of sets $\mathcal{K}$ and $\mathcal{K}_{pred}$ from the Topic labels $y$ and $y_{pred}$ are deterministically constructed and do not involve any LLM queries; instead, we use a custom Python (Python Software Foundation) web-scraping script with the Beautiful Soup (Leonard Richardson) open-source library to define each set of appropriate imaging studies for all Topics in the ACR AC from the URL https://gravitas.acr.org/acportal.

### Optimization of zero-shot prompt engineering and fine-tuning methods

In Fig. 3, we explore 4 distinct LLM optimization strategies—RAG, ICL, chain-of-thought (COT) prompting, and MFT—to improve the ability of LLMs like Claude Sonnet-3.5 and Llama 3 to accurately predict relevant ACR AC Topics from input patient one-liner scenarios. All LLM prompt

templates are included in Supplementary Methods B. For all experiments described herein, LLM prompts were first optimized on a small, holdout set of 10 synthetically generated one-liners that were not a part of the RadCases dataset before being used for all experiments reported herein.

In our RAG approach, we first constructed the relevant reference corpus of guidelines made publicly available by the American College of Radiology (ACR). A link to our custom script is included in our publicly available code repository. Using our custom Python script[65], we first implemented a web scraper, in compliance with the ACR Terms and Conditions, to download relevant Portable Document Format (PDF) narrative files from acsearch.acr.org/list on July 17, 2024. Each ACR AC Topic is associated with one accompanying narrative document, resulting in a total of 224 narrative files extracted. We then used the Unstructured IO open-source library to extract the PDF content into raw text, and segmented the text into 3380 disjoint documents each containing between 1119 and 2048 characters. Our strategy for constructing the retrieval corpus is identical to that used by Xiong et al.[66].

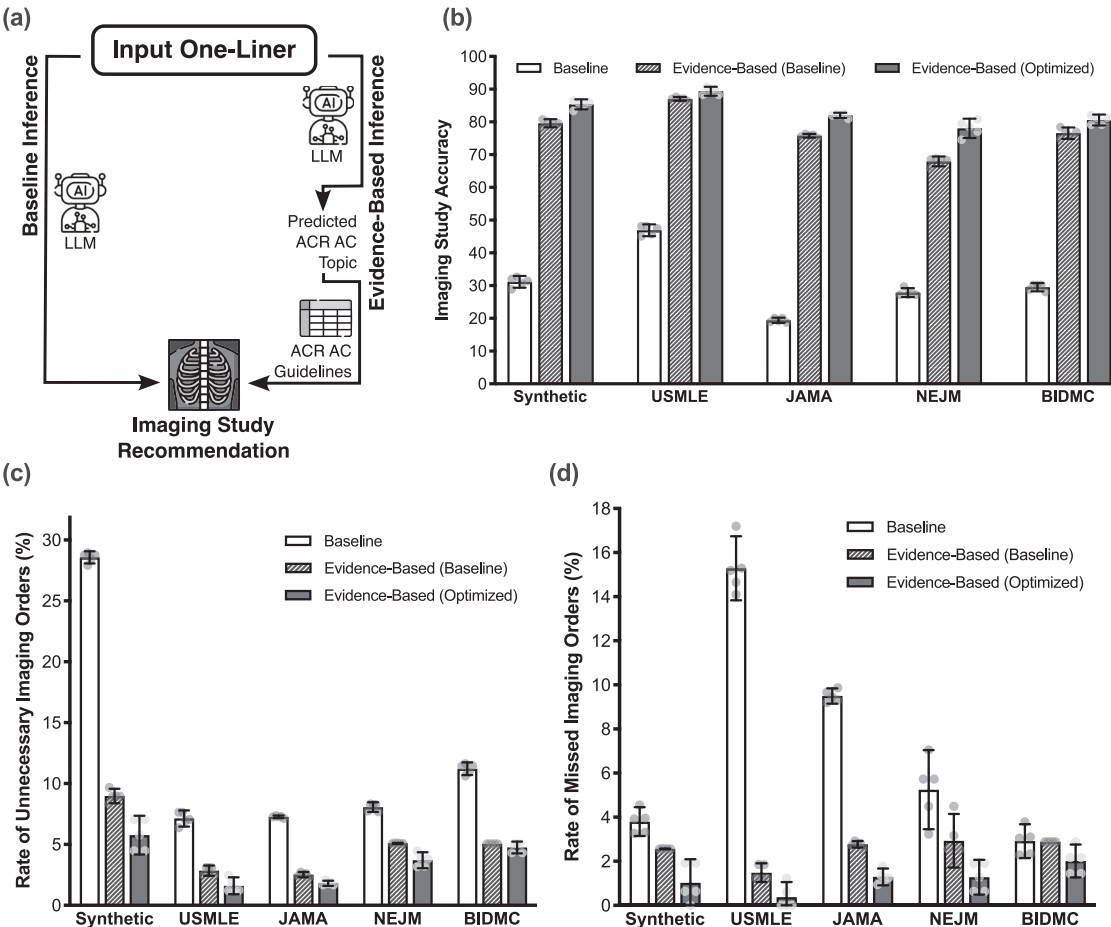

**Fig. 4 | Comparison of baseline and evidence-based inference pipelines with Claude Sonnet-3.5. a** Using our evidence-based inference pipeline, we query the LLM to predict the single ($k = 1$) ACR AC Topic most relevant to an input patient one-liner, and programmatically refer to the evidence-based ACR AC guidelines to make the final recommendation for diagnostic imaging. An alternative approach is the baseline inference pipeline where we query the LLM to recommend a diagnostic imaging study directly without the use of the ACR AC. **b** Our evidence-based pipelines (both using baseline prompting and optimized using chain-of-thought (COT) prompting) significantly outperform the baseline pipeline by up to 62.6% (two-sample, one-tailed homoscedastic $t$-test; Synthetic $p = 2.22 \times 10^{-12}$; USMLE

$p = 9.86 \times 10^{-12}$; JAMA $p = 1.74 \times 10^{-15}$; NEJM $p = 5.08 \times 10^{-11}$; BIDMC $p = 1.26 \times 10^{-12}$). At the same time, our optimized evidence-based pipeline also reduces the rates of both **c** unnecessary imaging orders (two-sample, one-tailed homoscedastic $t$-test; Synthetic $p = 9.18 \times 10^{-13}$; USMLE $p = 1.80 \times 10^{-7}$; JAMA $p = 6.14 \times 10^{-12}$; NEJM $p = 1.52 \times 10^{-8}$; BIDMC $p = 4.53 \times 10^{-10}$) and **d** missed imaging orders (two-sample, one-tailed homoscedastic $t$-test; Synthetic $p = 1.38 \times 10^{-4}$; USMLE $p = 2.74 \times 10^{-9}$; JAMA $p = 3.79 \times 10^{-11}$; NEJM $p = 2.52 \times 10^{-4}$; BIDMC $p = 0.0240$). Error bars represent ± 95% CI over $n = 5$ independent experimental runs. *Icons of language model, table, and X-ray in (**a**) courtesy of Freepik, used with permission under Premium Flaticon License.*

Using this corpus of relevant guidelines written by the ACR, we explored 8 different retriever algorithms to use for RAG: (1) Random, which randomly retrieves $k$ documents from the corpus over a uniform probability distribution; (2) Okapi BM25 bag-of-words retriever[67]; (3) BERT[68] and (4) MPNet[69] trained on unlabeled, natural language text; (5) RadBERT[28] from fine-tuning BERT on radiology text reports; (6) MedCPT[70] leveraging a transformer trained on PubMed search logs; and (7) OpenAI (text-embedding-3-large) and (8) Cohere (cohere.embed-english-v3) embedding models from OpenAI and Cohere for AI, respectively. Retrievers (3)–(8) are embeddings-based retrievers that leverage cosine similarity as the ranking function. These 8 retrievers represent a diverse array of well-studied, domain-agnostic, and domain-specific retrievers for RAG applications. In Fig. 3b, c, we report the results using the best retriever specific to each language model and RadCases dataset subset, fixing the number of retrieved documents to $k = 8$ for each retriever. We include the experimental results for each individual retriever in Supplementary Fig. 3.

Separately in our ICL approach, we use the RadCases-Synthetic dataset partition as the corpus of examples to retrieve from, and experimentally validate the same 8 retrievers used in RAG for retrieving relevant one-liner/ACR AC Topic pairs to provide as context to the language model. In Fig. 3b, c, we report the results using the best retriever specific to each language model and

RadCases dataset subset, fixing the number of retrieved examples to $k = 4$ for each retriever. To evaluate language models on the RadCases-Synthetic dataset using ICL, we constructed a separate corpus of synthetically generated, annotated one-liners for retrieval that was created using the identical prompting strategy as that for the RadCases-Synthetic dataset, except we used the Meta Llama 2 (7B) model (meta-llama/Llama-2-7b-chat-hf). We leveraged this separate corpus for ICL to avoid data leakage in our RadCases-Synthetic ICL evaluation experiments. We include the experimental results for each individual retriever in Supplementary Fig. 4 and explore the effect of different values of $k$ (i.e., the number of retrieved examples) in Supplementary Fig. 5.

In COT prompting, we explore four different reasoning strategies identical to those employed by Savage et al.[22]: (1) Default reasoning, which does not specify any particular reasoning strategy for the LLM to use; (2) Differential diagnosis reasoning, which guides the model to reason through a differential diagnosis to arrive at a final prediction; (3) Bayesian reasoning, which encourages the model to approximate Bayesian posterior updates over the space of ACR AC Topics based on the clinical patient presentation; and (4) Analytic reasoning, which encourages the model to reason through the pathophysiology of the underlying disease process. Examples of each reasoning strategy are included in Supplementary Results C. We include the experimental results for each individual reasoning strategy in

**Fig. 5 | Retrospective Study of Clinician-Ordered versus LLM-Ordered Imaging Studies.** We compare the diagnostic imaging studies ordered by the prompt-optimized LLMs Claude Sonnet-3.5 and Llama 3 against those ordered by clinicians in a retrospective study. Compared with clinicians, Claude Sonnet-3.5 and Llama 3 achieve the same or better **a** accuracy scores; and **b** false positive rates (i.e., the rate at which a patient received at least one unnecessary imaging recommendation); **c** false negative rates (i.e., the rate at which a patient should have received an imaging workup but did not); and **d** $F_1$ scores. **e** However, we observe that Claude Sonnet-3.5 orders a greater number of recommended imaging studies compared to clinicians. **f** According to the Dice-Sørensen Coefficient (DSC) metric, Claude Sonnet-3.5 and Llama 3 order imaging studies that are more similar to one another than to clinicians (two-sample, two-tailed homoscedastic $t$-test; $p = 2.19 \times 10^{-24}$). Error bars in (**e, f**) represent $\pm$ 95% CI over $n = 117$ independent patient cases. *(CL)*: Claude Sonnet-3.5 and Llama 3 pairwise DSC metric. *(CP)*: Claude Sonnet-3.5 and Physician pairwise DSC metric. *(LP)*: Llama 3 and Physician pairwise DSC metric.

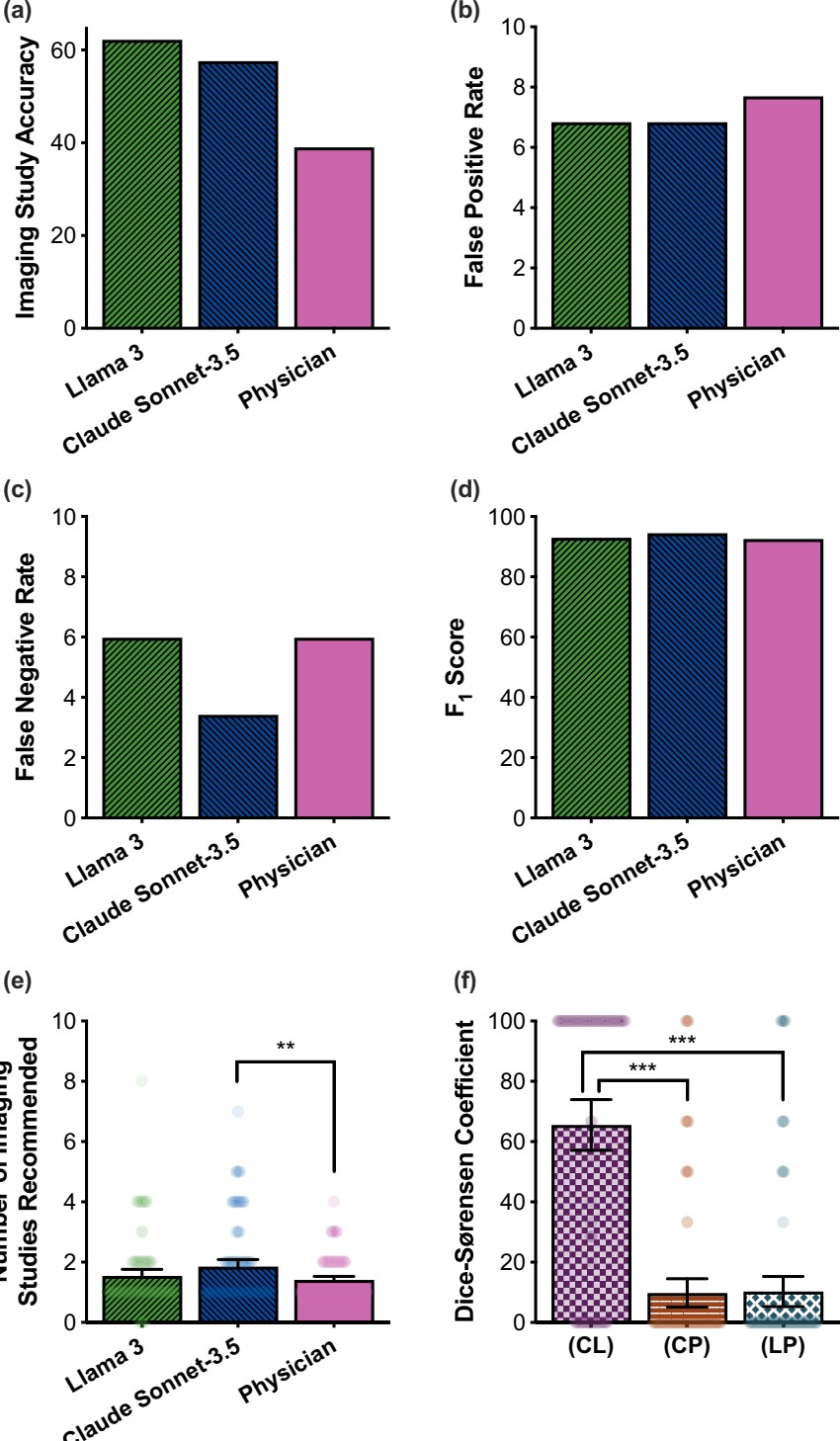

Supplementary Fig. 6, and investigate the utility of combining both ICL and COT (which were individually shown to be effective prompting strategies) in Supplementary Fig. 7. In Fig. 2b, c, we report the results using the best COT reasoning strategy specific to each language model and RadCases dataset subset. In Figs. 3–5, we report results using the Default reasoning strategy when COT is leveraged together with Claude Sonnet-3.5.

For MFT, we explore three different fine-tuning strategies using the Meta Llama 3 base model: (1) Full fine-tuning where all the parameters of the LLM are updated; and (2) Low-Rank Adaptation (LoRA)[71] and (3) Quantized Low-Rank Adaptation (QLoRA)[72] fine-tuning where only a subset of linear LLM parameters are updated. We fix the number of training epochs to 3 and the learning rate to 0.0001. For LoRA (resp., QLoRA), we use a rank of 64 (resp., 512) and an α scaling value of 8 (resp., 8). We chose these particular values according to a hyperparameter grid search over the rank and α hyperparameters, logarithmically ranging from 8 to 512 (resp., 1–512), that maximize the accuracy of the fine-tuned model on a synthetic validation dataset. Due to limitations on local compute availability, we were only able to run the QLoRA fine-tuning experiments on the internal experimental cluster; LoRA and Full fine-tuning experiments were performed using a third-party platform (Together AI). Finally, we also investigate two different fine-tuning datasets for each of the three strategies: (1) Synthetic fine-tuning using the RadCases-Synthetic dataset; and (2) Mixed

fine-tuning using a total 250 cases, where 50 random cases come from each of the five RadCases dataset subsets. To prevent data leakage, we use the Llama 2-generated Synthetic dataset (constructed for a similar purpose for our ICL experiments above) to fine-tune the base Llama 3 model for evaluation on the RadCases-Synthetic dataset in strategy (1) and avoid evaluation on any cases from the individual patients represented in the fine-tuning dataset in strategy (2). In Fig. 3c, we report the results using the LoRA fine-tuning strategy and Mixed fine-tuning dataset of 250 cases described above, as this led to consistently superior fine-tuning results across all datasets and language models that were evaluated. We report additional experimental results in Supplementary Figs. 8, 9.

### Translating ACR AC Topics into imaging study recommendations

In Fig. 4a, we overview our Evidence-Based inference pipeline where we leverage LLMs to assign ACR AC Topics to input patient one-liner scenarios, and then deterministically map Topics to appropriate imaging studies based on the Appropriateness Criteria guidelines. These LLM-generated recommendations were used as the basis of our retrospective and prospective studies described in our work. Constructing this mapping of Topics to imaging studies is a non-trivial task: for any particular Topic, the ACR AC often describes multiple nuanced clinical variants for a single topic. For example, for the "Suspected Pulmonary Embolism" Topic, there are 4 variants in the guidelines as of June 2024: (1) "Suspected pulmonary embolism. Low or intermediate pretest probability with a negative D-dimer. Initial imaging."; (2) "Suspected pulmonary embolism. Low or intermediate pretest probability with a positive D-dimer. Initial imaging."; (3) "Suspected pulmonary embolism. High pretest probability. Initial imaging."; and (4) "Suspected pulmonary embolism. Pregnant patient. Initial imaging." Each of these variants have distinct imaging recommendations: for example, variant (1) does not warrant any imaging study according to the ACR AC, whereas both computed tomography angiography (CTA) of the pulmonary arteries with intravenous (IV) contrast and a ventilation-perfusion (V/Q) scan lung are appropriate studies for variant (3). To define a deterministic mapping of topics to imaging studies, we therefore needed to isolate a single variant for each topic.

We manually reviewed each of the 224 Topics to determine this single variant. This process generally involved reverse engineering a "typical" patient presentation that would be described by a given Topic. In the above example, we reasoned that an acutely presenting patient whose most relevant Topic is "Suspected Pulmonary Embolism" would likely have a high pretest probability for a pulmonary embolism. Furthermore, pregnant patients are less common than non-pregnant patients in the emergency room, and the appropriate imaging studies for variant (3) are also appropriate for variant (4). For this reason, variant (3) was kept and the rest were discarded. As a result, a predicted imaging study of either CTA pulmonary arteries with IV contrast or V/Q scan lung were both considered correct in this example. If no imaging study was deemed appropriate based on the guidelines, then the ground-truth label was defined as "None".

### Retrospective study on autonomous image ordering using LLMs

To power our retrospective study comparing language models with clinicians, we extracted a diverse sample of 242 anonymized, de-identified admission notes derived from the MIMIC-IV dataset[55]. These notes were extracted from the medical records of 100 real patient admissions between 2008-2019 from the Beth Israel Deaconess Medical Center (Boston, MA)[55]. To simulate the limited clinical context typical of acute presentations, we manually truncated each admission note to include only relevant history and vital signs (Supplementary Results D). Admission notes were excluded from our analysis if either (1) the ACR Appropriateness Criteria contained no evidence-based guidance relevant to the patient scenario; or (2) the scenario described a patient admission that was not made in the emergency department (e.g., ICU downgrade to hospital floor). A total of 117 final patient scenarios were included in our analysis.

Using these patient scenarios, we prompted language models to predict up to $m$ ACR AC Topics that may be relevant for a given patient, and

programmatically referenced the ACR AC guidelines to determine the recommended imaging studies based on the LLM-recommended Topic(s). We set $m = 1$ Topic in our experiments in Fig. 5 and evaluated two LLMs from our original RadCases evaluation suite: Claude Sonnet-3.5 from Anthropic AI using COT prompting, and Llama-3 70B Instruct from Meta AI using no special prompt engineering (we further ablate the value of $m$ in Supplementary Fig. 11). We chose to evaluate these two models because they were the best performing proprietary and open-source models on the RadCases benchmarks, respectively (Fig. 2b). Simultaneously, we manually parsed through each of the full, original discharge summaries to determine what imaging study(s) were ordered by the patient's physician. The imaging studies ordered by both clinicians and language models were compared against the ground-truth best imaging study(s) as determined by consensus between two expert radiologists and two fourth-year U.S. medical students at the University of Pennsylvania.

### Constructing the patient cases for prospective user evaluation study

To support our prospective evaluation of LLMs as CDS tools for clinicians, we first constructed a separate dataset of 50 patient one-liners derived from the RadCases BIDMC subset of one-liners. The initially redacted details, such as patient name, age, or gender, were manually replaced with fictitious values. The cases were then reviewed and edited by three separate attending physicians to ensure that the cases were representative of typical real-world patient cases that commonly present in the emergency department.

### Participant recruitment and compensation

We conducted a clinical study with senior U.S. medical students and U.S. emergency medicine physicians to evaluate whether LLMs can assist in determining appropriate imaging studies. Our study was pre-registered on AsPredicted (#185312, available at https://aspredicted.org/x6b9-rcgh.pdf). Study participants were recruited from the Perelman School of Medicine and the Hospital of the University of Pennsylvania, where this study was conducted. Each volunteer participant received a base monetary incentive of $50 USD. To further incentivize performing to the best of their ability, the top 50% of most accurate participants—medical students and physicians, evaluated separately within each treatment arm—received an additional $10 USD. Consistent with prior work[73–76], we chose to offer this monetary compensation to improve recruitment rates and increase the diversity of opt-in participants, especially given the minimal risk posed by the study. A total of 23 medical students and 7 resident physicians participated in our experiment. All participating medical students were required to have passed and completed the emergency medicine clinical rotation at the University of Pennsylvania.

### Participant task in prospective study

Study participants were each tasked with ordering up to 1 diagnostic imaging study for a standardized set of 50 simulated patient case descriptions derived from the MIMIC-IV dataset[55]. Each case was presented on a custom-built website interface to display one patient case at a time; a visual of the interface is shown in Supplementary Fig. 12. For each case, participants selected an imaging study from a dropdown menu containing an alphabetized list of all 1150 diagnostic imaging studies officially recognized in the ACR Appropriateness Criteria. Of the 50 simulated cases, a random subset of 25 cases was chosen at the per-participant level that also showed LLM-generated recommendations for the participant to consult. Study participants were explicitly permitted to consult online resources they would typically use in the emergency department, but were prohibited from seeking assistance from other individuals. In simulated patient cases with more than one possible correct answer, study participants were instructed to select just one correct option.

Separately, study participants were also asked to complete a 5-question multiple-choice survey asking questions about their prior experience with AI tools, and overall sentiment about the use of AI in medicine (Supplementary Table 11). All study participant answers to this short survey and the

overall prospective study were anonymized and aggregated before analysis; participants were informed of this anonymization strategy in the informed consent.

### Statistics and reproducibility

All models and prompting techniques were evaluated on a single internal cluster with 8 NVIDIA RTX A6000 GPUs. The temperature of all language models was set to 0 to minimize variability in the model outputs. Each experiment was run using 5 random seeds, and we computed the mean accuracy of each method with 95% confidence intervals (CIs) against the human-annotated ground truth labels. A $p$ value of $p < 0.05$ was used as the threshold for statistical significance. In all figures, "n.s." represents not significant (i.e., $p \geq 0.05$); a single asterisk $p < 0.05$; double asterisks $p < 0.01$, and triple asterisks $p < 0.001$. All statistical analyses were performed using Python software, version 3.10.13 (Python Software Foundation), the SciPy package, version 1.14.0 (Enthought)[77], and the PyFixest package, version 0.24.2[78].

### Reporting summary

Further information on research design is available in the Nature Portfolio Reporting Summary linked to this article.

## Results

### RadCases: a dataset for evaluating LLM alignment with the ACR Appropriateness Criteria

Prior work evaluating LLMs for medical use cases have primarily relied on datasets that either contain complete pictures of patient presentations and outcomes[22,39,49,66] or are not representative of how clinicians document acute patient cases in practice[20,21]. As a result, such existing datasets do not adequately interrogate the ability of LLMs to take natural medical text written by clinicians as input and produce imaging recommendations that are aligned with the ACR Appropriateness Criteria. To address this limitation, we constructed the RadCases dataset, a labeled dataset of ~1500 patient case descriptions that mimic the structure of one-liner patient scenarios contained in medical documentation written by clinicians. The RadCases dataset is partitioned into 5 subsets: (1) Synthetic; (2) USMLE; (3) JAMA; (4) NEJM; and (5) BIDMC—the source and construction of each subset are detailed in the *Methods*. Each textual description is labeled by the most appropriate ACR Appropriateness Criteria guideline Topic that is most relevant to the patient case as determined by a consensus panel between U.S. attending radiologists and medical students. As an example, the input one-liner "49 M with HTN, IDDM, HLD, and 20 pack-year smoking hx p/w 4 mo hx SOB and non-productive cough" is labeled with the ACR AC Topic "Chronic cough."

Neurologic topics were the most common label in all 5 RadCases subsets, followed by cardiac and gastrointestinal conditions (Supplementary Fig. 1, Supplementary Table 1). We also found that our RadCases patient case descriptions were representative of real-world patient one-liners previously written by physicians in acute clinical workflows (Supplementary Table 2). Of note, while there are 224 unique diagnostic imaging topics in the ACR AC (as of June 2024), only 161 (71.9%) of all topics had nonzero support in the dataset. Furthermore, 73 (32.6%) unique topics are represented in the Synthetic dataset; 61 (27.2%) in the USMLE dataset; 119 (53.1%) in the JAMA dataset; 70 (31.3%) in the NEJM dataset; and 47 (21.0%) in the BIDMC dataset.

Using our RadCases dataset, we evaluated whether LLMs could yield better imaging study predictions if evidence-based guidelines were included as an explicit module in the patient scenario-imaging study inference pipeline. If a language model classified patient scenarios to a specific guideline (i.e., a Topic of the ACR AC), then the corresponding imaging study could then be deterministically identified from the content of the guideline itself. More concretely, we queried LLMs to map input one-liners to output ACR Appropriateness Criteria Topics, and then programmatically map these Topics to their corresponding evidence-based imaging recommendations (Fig. 2a).

### Evaluating large language models on the RadCases dataset

Figure 2b presents the performance of each LLM evaluated on each of the RadCases dataset subsets. Of the language models evaluated, Claude Sonnet-3.5 achieved the highest performance on 4 out of the 5 subsets (i.e., Synthetic, JAMA, NEJM, and BIDMC) and ranked second on the remaining subset (i.e., USMLE). Furthermore, Claude Sonnet-3.5 outperformed all open-source models with statistical significance (two-sample, two-tailed homoscedastic $t$-test; Synthetic $p = 3.68 \times 10^{-3}$; USMLE $p = 2.82 \times 10^{-4}$; JAMA $p = 1.57 \times 10^{-3}$; NEJM $p = 9.93 \times 10^{-4}$; BIDMC $p = 2.12 \times 10^{-5}$). Separately, Llama 3 outperformed all other evaluated open-source models across all 5 RadCases subsets (two-sample, two-tailed homoscedastic $t$-test; Synthetic $p = 2.14 \times 10^{-6}$; USMLE $p = 1.29 \times 10^{-4}$; JAMA $p = 5.25 \times 10^{-8}$; NEJM $p = 1.10 \times 10^{-4}$; BIDMC $p = 1.85 \times 10^{-4}$). Based on these results, we chose to further optimize Claude Sonnet-3.5 and Llama 3 in subsequent experiments as the most promising overall and open-source large language models, respectively. Common failure modes to explain incorrect model predictions by all 6 LLMs are detailed in Supplementary Data 2, and additional classification metrics are described in Supplementary Fig. 2.

### Optimizing large language models for imaging ordering in acute clinical workflows

While Claude Sonnet-3.5 and Llama 3 demonstrated strong baseline accuracy on the RadCases dataset, recent work have introduced techniques to improve the performance of generative language models. For example, RAG provides relevant context to language models retrieved from an information corpus (i.e., the ACR AC narrative medical guidelines written by expert radiologists) to help improve the generative process. ICL provides relevant examples of patient one-liners and their corresponding topic labels (i.e., examples from the RadCases-Synthetic dataset) as context to improve the zero-shot performance of language models. COT prompting is a strategy to improve the complex reasoning abilities of language models by encouraging sequential, logical steps to arrive at a final answer. Finally, MFT directly updates the parameters of a language model to improve its performance on a specific task. We assess all four strategies using Llama 3, and the zero-shot strategies RAG, ICL, and COT using Claude Sonnet-3.5 as there is no publicly available application programming interface (API) to fine-tune the proprietary model as of June 2024 (Fig. 3a).

Figure 3b demonstrates that COT (chain-of-thought prompting) is the most effective strategy for Claude Sonnet-3.5, yielding improvements of up to 17% in ACR AC Topic classification accuracy and consistent improvements across all five RadCases dataset subsets (two-sample, one-tailed homoscedastic $t$-test; Synthetic $p = 1.81 \times 10^{-6}$; USMLE $p = 2.83 \times 10^{-6}$; JAMA $p = 1.45 \times 10^{-9}$; NEJM $p = 3.78 \times 10^{-6}$; BIDMC $p = 6.54 \times 10^{-7}$). Interestingly, this same strategy does not translate well to Llama 3 (Fig. 3c); COT marginally improves upon baseline prompting for Llama 3 only on the USMLE RadCases dataset. Instead, ICL (in-context learning) was the most effective prompt engineering strategy for Llama 3, resulting in improvements of up to 9% on ACR AC Topic classification accuracy compared with naïve prompting (two-sample, one-tailed homoscedastic $t$-test; Synthetic $p = 4.72 \times 10^{-4}$; USMLE $p = 5.67 \times 10^{-7}$; JAMA $p = 0.0141$; NEJM $p = 9.91 \times 10^{-7}$). Additional fine-grained optimization results are included in Supplementary Figs. 3–9.

Our results indicate that while prompt engineering and other optimization techniques can indeed be effective in improving the performance of different language models on this task, the observed performance gains are often model-specific and may not generalize across different LLMs. This finding highlights the inherent challenge in optimizing such models for challenging tasks such as diagnostic image ordering via alignment with the ACR AC.

### Validating the LLM prediction pipeline

In Fig. 2b, we demonstrated that LLMs could achieve promising accuracy on the ACR AC Topic classification task; in Fig. 3b, c, we further optimized two state-of-the-art language models using prompt engineering techniques and MFT. Building on these results, we sought to validate our original hypothesis

and evaluate whether assigning ACR AC Topic predictions to patient one-liners could meaningfully improve LLM performance in diagnostic image study ordering.

We first mapped each ground-truth ACR AC Topic label in the Rad-Cases dataset to the corresponding ground-truth imaging study recommended by the relevant Topic guidelines (see *Methods* for additional details). We evaluated 3 different LLM inference pipelines using Claude Sonnet-3.5: (1) Baseline, which queries an LLM to directly recommend a diagnostic imaging study; (2) Evidence-Based Baseline, which queries an LLM to recommend an ACR AC Topic that is then mapped to the imaging study; and (3) Evidence-Based Optimized, which is the same as (2) but uses the optimized COT prompting strategy from Fig. 3b for Claude Sonnet-3.5 (Fig. 4a).

Our results demonstrate that using LLMs to map patient cases to ACR AC topics significantly improves the imaging accuracy achieved by the model. Across all 5 RadCases dataset subsets, our Evidence-Based Optimized pipeline outperforms the Baseline pipeline by at least 42% on imaging accuracy (two-sample, one-tailed homoscedastic $t$-test; Synthetic $p = 2.22 \times 10^{-12}$; USMLE $p = 9.86 \times 10^{-12}$; JAMA $p = 1.74 \times 10^{-15}$; NEJM $p = 5.08 \times 10^{-11}$; BIDMC $p = 1.26 \times 10^{-12}$) (Fig. 4b). Similarly, our Evidence-Based Baseline pipeline outperforms the Baseline pipeline by at least 40% on imaging accuracy (two-sample, one-tailed homoscedastic $t$-test; Synthetic $p = 3.02 \times 10^{-12}$; USMLE $p = 3.80 \times 10^{-12}$; JAMA $p = 1.47 \times 10^{-15}$; NEJM $p = 6.68 \times 10^{-12}$; BIDMC $p = 3.01 \times 10^{-12}$).

Interestingly, while the Evidence-Based Optimized pipeline outperformed the Evidence-Based Baseline pipeline on ACR AC Topic classification accuracy (Fig. 3b), we observed no statistically significant difference in the optimized and baseline Evidence-Based pipelines on the imaging classification accuracy (two-sample, one-tailed homoscedastic $t$-test; Synthetic $p = 0.111$; USMLE $p = 0.073$; JAMA $p = 0.165$; NEJM $p = 0.237$; BIDMC $p = 0.427$). Qualitatively, we found that although the Evidence-Based Baseline pipeline achieved a lower ACR AC Topic classification accuracy compared to the Evidence-Based Optimized inference strategy, its incorrect predictions were still closely related to the correct answer and underlying patient pathology. For example, a ground truth ACR AC Topic might be "Major Blunt Trauma" and the LLM prediction "Penetrating Torso Trauma;" although the LLM identified the incorrect ACR AC Topic label, both Topics warrant a "Radiography trauma series." As a result, both the optimized and baseline Evidence-Based pipelines achieve comparable imaging accuracy and significantly improve upon the Baseline pipeline.

We also evaluated the false positive and false negative rates in image ordering. Formally, false positives are cases where an imaging study is unnecessarily ordered, and false negatives are cases where a diagnostic imaging study was warranted but not ordered. Both Evidence-Based pipelines again outperformed the Baseline pipeline according to both metrics, significantly reducing the rates of false positives (two-sample, one-tailed homoscedastic $t$-test; Synthetic $p = 9.18 \times 10^{-13}$; USMLE $p = 1.80 \times 10^{-7}$; JAMA $p = 6.14 \times 10^{-12}$; NEJM $p = 1.52 \times 10^{-8}$; BIDMC $p = 4.53 \times 10^{-10}$) and false negatives (two-sample, one-tailed homoscedastic $t$-test; Synthetic $p = 1.38 \times 10^{-4}$; USMLE $p = 2.74 \times 10^{-9}$; JAMA $p = 3.79 \times 10^{-11}$; NEJM $p = 2.52 \times 10^{-4}$; BIDMC $p = 0.0240$).

### Investigating autonomous image ordering using LLMs versus standard of care

Based on the initial results in Fig. 4 and Supplementary Fig. 10, we next assessed whether state-of-the-art, optimized language models could accurately order imaging studies for acutely presenting patients without clinician intervention. Using a set of anonymized, de-identified admission notes derived from the medical records of 100 real patient admissions between 2008-2019 from the Beth Israel Deaconess Medical Center (Boston, MA)[55], we compared the accuracy of diagnostic image ordering of the prompt-optimized versions of Claude Sonnet-3.5 and Llama 3 against that of clinicians. In Fig. 5, our results suggest that autonomous LLMs can be effective tools in ordering diagnostic imaging: Claude Sonnet-3.5 achieved a higher accuracy score of 58.0% and F$_1$ score of 94.4% compared with clinicians (accuracy 39.3%; F$_1$ score 92.5%) (McNemar test; $p = 0.044$). Similarly, there

was no statistically significant difference between Llama 3 (accuracy 61.5%; F$_1$ score 92.9%) and clinicians (McNemar test; $p = 0.099$). Across the patient cases assessed, clinicians ordered an average of 1.41 (95% CI: [1.28–1.53]) imaging studies per case; similarly, Claude Sonnet-3.5 ordered an average of 1.83 (95% CI: [1.60–2.06]) and Llama 3 an average of 1.54 (95% CI: [1.33–1.76]) studies per case. There was no statistically significant difference between the number of imaging studies ordered by Llama 3 and clinicians (two-sample paired $t$-test; Llama 3: $p = 0.269$).

We also evaluated the rates of both unnecessary (false positive) and missed (false negative) imaging studies: both Claude Sonnet-3.5 and Llama-3 were non-inferior to clinicians according to both metrics, achieving a false positive rates (FPR) of 6.90% and 6.90% (clinician FPR = 7.76%) (McNemar test; $p = 1.00$) and false negative rates (FNR) of 3.45% and 6.03% (clinician FNR = 6.03%) (McNemar test; Llama 3: $p = 1$; Claude Sonnet-3.5: $p = 0.549$), respectively (Fig. 5b, c). Altogether, these findings suggest that LLMs are promising tools for image ordering in clinical workflows.

Finally, to gauge the similarity between recommendations made by different language models and clinicians, we computed the pairwise Dice-Sørensen coefficient (DSC) between imaging recommendations made by different decision makers (Fig. 5f). This analysis revealed that recommendations made by different language models (Llama 3 and Claude Sonnet-3.5 DSC 65.5% (95% CI: [57.1–73.9%])) consistently aligned more closely than those made by language models and clinicians ((Llama 3 and Physician DSC 10.3% (95% CI: [5.3–15.3%]); two-sample, two-tailed homoscedastic $t$-test; $p = 1.68 \times 10^{-23}$), (Claude Sonnet-3.5 and Physician DSC 9.8% (95% CI: [5.2–14.5%]); two-sample, two-tailed homoscedastic $t$-test; $p = 2.19 \times 10^{-24}$)).

### Evaluating language models as support tools for clinician diagnostic image ordering

In our experiments above, we assessed LLMs as autonomous agents for clinical decision making. Such retrospective studies help clarify the technical capabilities and limitations of these models compared with standard of care. However, LLMs can also act as assistants for clinicians in diagnostic image ordering.

To evaluate the utility of our evidence-based LLMs as clinical assistants, we conducted a prospective clinical study asking volunteer clinician participants to order diagnostic imaging studies for simulated patient scenarios in an online testing environment. Participants were U.S. medical students and emergency medicine resident physicians recruited from the Perelman School of Medicine and the Hospital of the University of Pennsylvania. This study was reviewed and exempted by the University of Pennsylvania Institutional Review Board (Protocol #856530), as it involved the use of publicly available, de-identified patient data and posed minimal risk to participants. No protected health information was accessed, and no clinical care was affected.

All study participants provided documented informed consent prior to participation. They were informed about the nature and purpose of the research, the voluntary nature of their involvement, the procedures involved, and their right to withdraw at any time. To further protect participant confidentiality, data collected during the study was anonymized and not linked to identifiable individuals. Participant identities were collected only for the purpose of disbursing monetary compensation, and this information was stored separately from study data.

Each study participant was asked to order a single imaging study (or forego imaging if not indicated) for 50 simulated patient cases. For each participant, a random 50% of the patient cases included recommendations generated by Claude Sonnet-3.5 using the evidence-based optimized inference strategy in Fig. 4. To simulate the acuity and high-pressure of many emergency room environments, participants were required to complete the study at an average rate of 1 case per minute in a single setting. We then fitted a regression model according to

$$y_{s,q} = \beta_0 + \left( \beta_1 * \text{WithLLMGuidance}_{s,q} \right) + \theta_q + \chi_s + \varepsilon_{s,q} \qquad (2)$$

where $s$ indexes study participants and $q$ study questions, and $y_{s,q}$ is a binary variable indicating whether participant $s$ answered study question $q$ correctly. Here, $\theta$ is a $q$-vector of study question fixed effects, $\chi_s$ are control variables specific to the study participant (i.e., whether the study participant is a physician or medical student, the participant's personal experience with AI, and the participants sentiment regarding AI), and $\varepsilon_{s,q}$ is the error term. We estimate Eq. 2 using standard errors clustered at the study participant level and question level. Furthermore, WithLLMGuidance$_{s,q}$ is a binary indicator that indicates whether LLM-generated guidance was provided for question $q$ for participant $s$, respectively.

Study participants generally found the study task challenging, with an average accuracy of 15.8% (95% CI: [12.2–19.3%]) without LLM guidance and 25.0% (95% CI: [20.7–29.3%]) with guidance. Offering LLM-based recommendations using our evidence-based optimized pipeline improved image ordering accuracy with statistical significance ($\beta_1 = 0.081$; 95% CI: [0.022–0.140]; $p = 0.011$) for both medical students and resident physicians.

To verify that participants were indeed taking advantage of LLM-generated recommendations when made available, we fitted a separate regression model analogous to that in Eq. 2 that instead measures the binary agreement between LLM recommendations and participant answers as the dependent variable. As expected, the agreement between answers and assistant recommendations increases when the recommendations are made available to the clinician ($\beta_1 = 0.141$; 95% CI: [0.050–0.233]; $p = 0.005$). These results suggest that language models can act as clinical assistants to help clinicians order imaging studies more aligned with evidence-based guidelines.

Similarly, we did not observe statistically significant differences in either the false positive rate ($\beta_1 = 0.008$; 95% CI: [−0.012 to 0.027]; $p = 0.418$) or false negative rate ($\beta_1 = -0.019$; 95% CI: [−0.068 to 0.030]; $p = 0.431$). This ensures that the improvements in accuracy scores with LLM guidance were not at the cost of substantially increasing the number of unnecessary or missed imaging studies ordered by clinicians. Additional analysis is included in Supplementary Tables 4–10, and discussion of experimental results in Supplementary Results A.

## Discussion

Our study investigates the potential of LLMs in the domain of diagnostic image ordering—a task critical to the timely and appropriate management of acute patient presentations. Our results demonstrate how state-of-the-art language models can be used in the context of diagnostic image ordering in acute clinical settings, such as the emergency department. Firstly, we observed that generalist language models—such as Claude Sonnet-3.5 and Meta Llama 3—can accurately predict relevant ACR AC Topic labels to describe patient one-liner descriptions without any domain-specific fine-tuning. By leveraging LLMs to predict Topic labels instead of imaging studies directly, we were able to improve the quality of final imaging recommendations made by LLMs. Comparing language models with clinicians in a retrospective study, we show that LLMs achieve better accuracy with regard to image ordering in the ED without significant changes to the rate of missed imaging (FNR), rate of unnecessary imaging (FPR), or number of recommended imaging studies. Finally, we demonstrate that LLMs can be leveraged by clinicians as a CDS assistant to improve the accuracy of ordered imaging studies without significantly affecting the FPR or FNR in a simulated acute care environment.

Importantly, we demonstrate how integrating evidence-based guidelines (i.e., the ACR AC) directly into the LLM-based inference pipeline can improve the accuracy of clinical recommendations. This approach not only aligns model predictions with established guidelines, but also provides a robust framework for reducing the rates of both unnecessary and missed imaging orders. In theory, such a framework could be readily adapted to make use of available guidelines in other clinical problems, such as the American College of Gastroenterology guidelines to determine clinical indications for endoscopy[79–81], or the American Society of Addiction Medicine guidelines for the management of alcohol withdrawal syndrome[82]. We leave these potential future applications of LLM-based CDS tools for future work.

We also highlight the challenges associated with integrating LLM toolkits into existing clinical workflows. Our prospective study demonstrated preliminary evidence that the utility of LLM clinical assistants can be largely dependent on factors such as user expertise, acuity of care, and existing user attitudes on AI, consistent with prior work[41,83–85]. Nonetheless, we observed that the accuracy of imaging studies ordered by clinicians increased by ~10 percentage points on average, which can potentially translate to hundreds of dollars saved per patient in reducing low-value and unnecessary imaging studies according to recent work[86–89]. That being said, we highlight that our study was limited by a relatively small sample size of only 23 medical students and 7 junior emergency resident physicians. Furthermore, our study participants voluntarily opted in to participate in our study and may not reflect the attitudes and behaviors of clinicians that may have a more conservative predisposition to the use of AI tools in healthcare. Finally, we highlight that ED residents at large academic institutions (such as the University of Pennsylvania where this study was conducted) may not currently be trained to order imaging studies in alignment with the ACR AC, as the benefit-to-cost ratio of obtaining more extensive imaging studies may be different institutionally than as dictated by national guidelines. Given these considerations, future work is warranted to better characterize the impact of these factors across diverse populations of healthcare workers as they affect real-world clinical workflows and physician thinking, ultimately ensuring that LLMs are used responsibly and can improve patient care.

Of note, our results consistently demonstrate that proprietary language models, such as Claude Sonnet-3.5, consistently outperform open-source models. While the performance of Claude Sonnet-3.5 is impressive, it is unlikely that current publicly available inference APIs for the model are sufficient for widespread clinical deployment, as many hospitals understandably express concerns over patient privacy and unknown data handling practices from third-party vendors. In these settings, our experimental results suggest that open-source language models, such as Llama 3, can be potentially viable alternatives. Future work might investigate other strategies that better leverage open-source models to be better compliant with healthcare regulations and best practices.

This study also has its limitations. Firstly, because most of the patient descriptions in our RadCases dataset are derived from real medical sources, they also reflect inherent biases with respect to patient demographics and medical conditions. For example, we found that our dataset most commonly included ground-truth ACR AC Topic labels related to gastrointestinal, cardiac, and neurologic pathologies—while these cases may reflect real-world ED visit patterns, it remains to be seen how LLMs perform on other patient cases sampled from different underlying distributions, such as in rare disease diagnostics and low-resourced patient populations.

Secondly, we note that each of the ACR AC Topics considered in our study are also further stratified into "scenarios" for more nuanced imaging recommendations—for example, the ACR AC Topic "Breast Pain" can be decomposed in 4 related clinical "scenarios" (as of June 2024) each with their own imaging recommendations: (1) Female with clinically insignificant breast pain without other suspicious clinical finding, any age; (2) Female with clinically significant breast pain, age less than 30; (3) Female with clinically significant breast pain, age 30–39; and (4) Female with clinically significant breast pain, age greater than or equal to 40. Due to computational limitations and a restrictive LLM context length, we group all relevant scenarios into a single set of imaging recommendations for the single parent ACR AC Topic in our work; future work might investigate the performance of LLM-based inference pipelines for classification of individual ACR AC scenarios.

Furthermore, closely related topics (e.g., "Major Blunt Trauma" and "Penetrating Torso Trauma") often share clinical indications for the same set of imaging studies (e.g., "Radiography trauma series"). As a result, our choice of imaging accuracy evaluation metrics in Figs. 4, 5 still permits an LLM to predict the correct imaging study even if an incorrect ACR AC Topic was identified. This potential limitation of models achieving the "right answer through the wrong reasoning" is well-documented in prior and

concurrent work examining discrepancies between model reasoning traces and final predictions[34,90–94], and conceivably may degrade model performance if the ACR AC is updated such that the ground-truth and model-predicted ACR AC Topics no longer share the same imaging study. However, we highlight that even if we require our language models to obtain the "right answer through the *right* reasoning," our inference strategy in Fig. 4a still outperforms baseline reasoning strategies (i.e., the ACR AC Topic Prediction accuracy in Fig. 3b, c is greater than the imaging accuracy of baseline LLMs in Fig. 4b).

We also highlight that our main experimental results are reported on the subset of patient cases with at least one ACR AC Topic label. In general, patient one-liners with no matching ground-truth label were excluded from our analysis; see our *Methods* for additional details. In Supplementary Table 3, we investigate the ability of language models to predict whether there exists at least one ACR AC Topic that is relevant for a given patient as a binary classification task. Future work might explore multi-step pipelines involving sequential LLM queries that first determine if a set of ACR AC Topics is relevant before predicting the most relevant Topic within the set.

It is also important to ensure that LLM-based tools operate equitably across diverse patient populations and clinical scenarios before they are deployed across hospitals systems—as clinical medicine and associated evidenced-based guidelines evolve over time, it is important to recognize model drift and concomitant deterioration in the performance of LLMs may affect the accuracy and reliability of model recommendations[95–100]. Future studies are warranted to investigate these problems and their real-world impact on model predictions, and also develop strategies to mitigate them.

Importantly, we also emphasize that even though we leverage the ACR AC as a ground truth symbol in our experiments (Figs. 2, 3), evidence-based guidelines like the ACR AC are ultimately recommendations that should be used in conjunction with clinical expertise to help physicians make the most appropriate decisions regarding the role of diagnostic imaging. Such recommendations may therefore fall short in more challenging patient cases not considered in this work, such as those with multiple medical conditions, complex admissions, and/or prior imaging studies that can drastically affect the appropriateness of different diagnostic methods. For these reasons, we argue that any LLM-based CDS tool–such as the ones that we evaluated in our work–should ultimately be used in the same fashion, where LLM-generated recommendations are used by clinicians together with their individual expertise to best contextualize the role of diagnostic imaging in specific patient scenarios.

Finally, we emphasize that our experiments, while promising, are no substitute for true prospective evaluation of language models as CDS tools in real-world clinical workflows, such as those in the emergency department[101]. We particularly highlight that practical applications of our work might focus on targeting clinical decision making for costly imaging studies (e.g., magnetic resonance imaging) and those associated with relatively higher radiation doses (e.g., computed tomography). Future work is needed in close collaboration with ED physicians across a variety of clinical environments to truly validate the potential clinical utility of the LLM-based pipelines explored in this study.

In conclusion, our study highlights the potential of LLMs to enhance the process of diagnostic image ordering by leveraging evidence-based guidelines. By mapping patient scenarios to ACR AC Topics and using optimized model strategies, LLMs can improve accuracy and efficiency in imaging decisions. Our findings suggest that LLMs could play a transformative role in supporting clinicians and improving patient care in acute diagnostic workflows.

## Data availability

All data are made available within the article, supplementary information, or the source data file provided with this paper. Raw data of all language model predictions are made publicly available without restrictions on Dryad at https://doi.org/10.5061/dryad.p8cz8wb0b[102]. The RadCases dataset is made publicly available at https://doi.org/10.57967/hf/6013[103]. The source data for all figures is in Supplementary Data 1. All other data are available from the corresponding author upon reasonable request.

## Code availability

Custom code used for large language model evaluation is made publicly available without restrictions at https://github.com/michael-s-yao/radGPT[104], and custom code for running our prospective clinical study is made publicly available without restrictions at https://github.com/michael-s-yao/radGPT-UI[105]. Persistent links to the versions of the code described in this paper are available at https://doi.org/10.5281/zenodo.16055681[104] and https://doi.org/10.5281/zenodo.16055916[105], respectively.

## Abbreviations

| | |
|---|---|
| ACR | American College of Radiology |
| ACR AC | American College of Radiology Appropriateness Criteria |
| AI | Artificial Intelligence |
| CDS | Clinical Decision Support |
| CI | Confidence Interval |
| COT | Chain-of-Thought |
| ED | Emergency Department |
| FNR | False Negative Rate |
| FPR | False Positive Rate |
| ICL | In-Context Learning |
| LLM | Large Language Model |
| MFT | Model Fine-Tuning |
| RAG | Retrieval-Augmented Generation |

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

## Acknowledgements

The authors thank Kevin Johnson, M. Dylan Tisdall, and Mark Yatskar (listed alphabetically) at the University of Pennsylvania for their helpful discussions, and the anonymous referees for their valuable input and feedback. We also appreciate the anonymous study participants and Wilma Chan, Lauren Conlon, Michael Abboud, Matthew Magda, and Mira Mamtani in the Department of Emergency Medicine at the University of Pennsylvania for their help and support in conducting the prospective clinician-AI study. This research was funded by the National Science Foundation Division of Computing and Communication Foundations (NSF Award CCF-1917852). M.S.Y. was supported by the NIH (F30 MD020264). A.C. was supported by the 2023 Alpha Omega Alpha Carolyn L. Kuckein Student Research Fellowship and a research grant from the Department of Radiology at the University of Pennsylvania. J.C.G. was supported by the NIH (R01 EB031722). H.S. was supported by the Institute for Translational Medicine and Therapeutics' (ITMAT) Transdisciplinary Program in Translational Medicine and Therapeutics, and by the National Center for Advancing Translational Sciences of the National Institutes of Health under Award Number UL1TR001878. O.B. was supported by NSF Award CCF-1917852. The content is solely the responsibility of the authors and does not necessarily represent the official views of the NIH or the NSF.

## Author contributions

M.S.Y., A.C., H.S., and O.B. conceived the study. M.S.Y. and A.C. planned and performed experiments. P.S. and H.S. supervised the approved the final annotations of the RadCases dataset. M.S.Y. and A.C. analyzed the data with feedback from C.E.K., W.R.W., and J.C.G. M.S.Y. and O.B. wrote the manuscript with input from all other authors. H.S. and O.B. jointly supervised the research.

## Competing interests

The authors declare no competing interests.
