## [Transparent Peer Review file · Communications Medicine]

Evaluating Acute Image Ordering for Real-World Patient Cases via Language Model Alignment with Radiological Guidelines

Corresponding Author: Dr Osbert Bastani

Version 0:

Reviewer comments:

Reviewer #1

(Remarks to the Author)

The authors have extensively responded to the prior reviewers and updated the manuscript. While the concept is interesting, the authors responses have either 1) revealed limitations that are not adequately addressed in the manuscript and/or 2) not clearly or succinctly made some important elements of the study clear. Both the manuscript and the responses are quite long and that makes it difficult to evaluate. I suggest that the authors spend more time in the future producing more succinct and clear responses and a more linear manuscript.

Prior Reviewer 1 Comment Re Line 142: You responded “our study is to evaluate the alignment of language model outputs with existing ACR AC guidelines, not to evaluate whether a language model can identify whether a set of guidelines is applicable or not” – If you do not evaluate whether the LLM can identify the applicable guidelines, how can you say whether the LLM could perform appropriately in a clinical scenario?

Prior Reviewer 1 Comment Re Fig 3: “Qualitatively, we found that although the Evidence-Based Baseline pipeline achieved a lower ACR AC Topic classification accuracy compared to the Evidence-Based Optimized inference strategy, its incorrect predictions were still closely related to the correct answer and underlying patient pathology. For example, a ground truth ACR AC Topic might be “Major Blunt Trauma” and the LLM prediction “Penetrating Torso Trauma;” although the LLM identified the incorrect ACR AC Topic label, both Topics warrant a “Radiography trauma series.” While you may get some form of appropriate answer, the incorrect matching will be a problem whenever ACR guidelines update and the irrelevant errors become relevant errors. Incorrect matching is a fundamental problem that is not getting enough attention in this manuscript and that indicates the conclusions are overstated.

Prior Reviewer 3 Comment 1: I appreciate the revisions to the introduction with the additional evidence. However, “Inappropriate” is loaded with judgement and you do not have the information to pass that judgement post-hoc (i.e. you do not know the knowledge that the provider had or the situation that the provider was in at the time of the order). I recommend that all instances of “inappropriate” be shifted to the concept of “overuse” which is a health services term tied to medical waste. I think it’s still a reach.

Prior Reviewer 3 Comment 7 and 9: The explanation of the LLM is helpful. Is this LLM validated? I do not see an explanation of how the LLM result was tied to the authors checking the ground truth of whether ACR guidelines were actually followed. It seems the authors relied on the LLM exclusively. Some form of ground truth is needed or should be explicitly and clearly explained.

Prior Reviewer 3 Comment 8: These additional experiments are helpful and you have essentially performed a sensitivity analysis – sometimes the LLM is better and sometimes it is not. That information does not support a conclusion that they can be valuable. They might be. But you haven’t proven it. You’ve at most shown that there is potential. And in combination with some of the comments above, the conclusions do not track with what is shown by the data.

Reviewer #2

(Remarks to the Author)

The authors have adequately addressed the comments from Reviewers 1 and 2. The article is suitable for publication.

Reviewer #3

(Remarks to the Author)

Comments to Authors:

Thank you for submitting this important work, which evaluates the potential of large language models (LLMs) to assist in diagnostic imaging study selection by mapping clinical scenarios to ACR Appropriateness Criteria (ACR AC). The integration of techniques like retrieval-augmented generation (RAG), in-context learning (ICL), and chain-of-thought prompting (COT) is a strong feature of the manuscript, and the inclusion of the novel RadCases dataset represents a valuable contribution to the literature. While the work is well-written and addresses a critical issue in emergency departments (EDs), there are areas that require further clarification, refinement, and discussion to strengthen the manuscript. Below, I provide specific comments, followed by general suggestions and strengths.

Specific Comments

Introduction

- ACR Scenarios and Imaging Topics:

- o The paper notes that the ACR AC includes 4000 clinical scenarios but focuses on 263 imaging topics. While this focus is acceptable, it represents a limitation that should be explicitly acknowledged. Additionally, ACR now lists 267 imaging topics; it would be prudent to verify this and update the manuscript accordingly. The grouping of scenarios (e.g., "cervical spine pain") into broader topics is understandable but should be discussed, as the ACR segregates these scenarios due to variations in imaging recommendations based on factors like initial versus subsequent imaging.

Figures and Results

- Figure 1:

- o The rationale for including Claude 3.5 in Figure 1 is unclear, as it appears to focus solely on RadCases performance. If this is due to prior peer-review feedback, the manuscript should explicitly address why only this model is shown and clarify its relevance in the current context. Notably, there is no reference to Figure 1 in the manuscript text.

- Results Section:

- o Portions of the results section, particularly commentary on RadCases, would be more appropriately placed in the discussion. Additionally, content under "Evaluating language models as support tools for clinician diagnostic image ordering" includes elements better suited for the methods or discussion sections. For instance, hypotheses (e.g., "using language models to map patient scenarios to ACR AC Topics...") belong in the introduction or discussion, not the results.

- Ground Truthing and RadCases Labeling:

- o The example provided, "49M with HTN, IDDM, HLD, and 20 pack-year smoking hx p/w 4 mo hx SOB and non-productive cough," is mislabeled as "Lung Cancer Screening." This patient is symptomatic and should receive a CT without contrast rather than screening, which is reserved for asymptomatic individuals aged 50–80 years with a 20-pack-year smoking history. This highlights concerns about ground truthing and labeling. I suggest two or more MDs involved, one of whom should be a subspecialist (e.g. in this case a thoracic radiologist or pulmonologist) to ensure accuracy. Since the dataset has more neurological cases, perhaps adding neuroradiologists and other subspecialists as needed.

- Dataset Imbalance:

- o Neurologic topics dominate the RadCases dataset, followed by cardiac and gastrointestinal conditions. While this imbalance may reflect real-world ED visit patterns, it should be acknowledged and discussed as a potential limitation.

- ACR AC Topic Count:

- o The manuscript states there are 224 diagnostic imaging topics, but ACR lists 247, including 20 interventional radiology topics. Subtracting these leaves 227 diagnostic imaging topics. Please verify the count and update the manuscript as necessary.

- Figure 3:

- o While avoiding data leakage in ICL evaluations is commendable, reserving additional cases separate from training and validation datasets could allow for evaluation of ICL performance on RadCases-Synthetic. Consider augmenting the dataset for this purpose.

- Figure 4:

- o The assumption that selecting the ACR AC Topic is equivalent to selecting the correct imaging study introduces limitations. Many topics encompass multiple scenarios with different imaging recommendations (e.g., "Right Upper Quadrant Pain" includes five clinical variants with differing recommendations). The authors should clarify whether the goal is to identify the most applicable clinical scenario or the general topic. Additionally, Figure 4 comparisons may be misleading, as baseline (LLM selecting the correct imaging study) is inherently different from evidence-based baselines and optimized approaches, which select the ACR AC Topic. Consider revising this figure or ensuring consistent evaluation criteria across methods.

- Page 18, Second Paragraph:

- o The manuscript references Figure 4f, but this appears to be a typographical error as the content aligns with Figure 5f. Please correct this discrepancy.

General Points for Discussion

1. One-Liner Summaries:

- o The reliance on one-liner summaries makes several assumptions, including that all relevant clinical details can be summarized in a single sentence. This may not always be the case, as certain scenarios require multiple lines for full context. The rationale for using one-liners should be elaborated. For example:

- o One-liners are appropriate when clinical records are unavailable and meaningful interaction with an LLM is needed for

specific questions.

When full ED notes are available, having the AI analyze these records directly may be a more streamlined and clinically useful workflow than requiring ED physicians to synthesize one-liners.

2. Real-World ED Integration:

o The retrospective analysis demonstrates LLMs' ability to identify the correct ACR topic but does not fully address their capability to recommend appropriate imaging studies. Future work should include prospective evaluation in real-world ED workflows, engaging ED physicians to validate clinical utility.

3. Focus on High-Impact Imaging Studies:

o It may be worthwhile to consider this in future work: Targeting high resource utilization studies (e.g., PE protocol CTs, head CTs, brain MRIs, spine MRIs)

4. Complex Scenarios:

o Real-world ED cases often involve overlapping conditions and nuanced imaging needs. Additionally, prior imaging results often influence study selection. The current approach assumes ACR topic mapping is sufficient, which may oversimplify decision-making in clinical practice. These limitations should be discussed.

5. Combined Strategies:

o Consider exploring the combined use of the best performing techniques (ICL + COT), or all techniques, as this could further improve performance.

6. Prospective Evaluation:

o The inclusion of prospective evaluation strengthens the study. However, further details are needed:

The low performance of ER resident physicians is surprising. What was their year of training?

Were the scenarios typical or atypical ED cases? If unrepresentative, this could bias the results.

Including board-certified ED physicians in future evaluations would enhance generalizability.

Strengths

o Overordering and inappropriate imaging are major challenges in EDs, and this research addresses a critical issue.

o RadCases is a valuable contribution to the literature and provides a foundation for further research.

o Employing RAG, ICL, COT, and model fine-tuning enhances the manuscript's impact and provides educational value to the audience.

o The study represents an important first step toward integrating AI tools into clinical workflows, laying the groundwork for future innovations.

Reviewer #4

(Remarks to the Author)

This manuscript addresses an important clinical challenge—improving the accuracy of imaging ordering in acute settings by leveraging large language models (LLMs) aligned with evidence-based ACR Appropriateness Criteria. Overall well-written and study design is sound. However, overall novelty of the work appears modest compared to numerous prior publications on radiology appropriateness criteria.

In this regard, it is worth noting that the true novelty of the manuscript may lie in the introduction of the RadCases dataset itself. The authors might consider renaming the manuscript to emphasize the RadCases dataset and refocusing the writing on how this dataset, in combination with the chosen LLM optimization methods and clinical evaluation pipeline, advances current knowledge—rather than rehashing material on Appropriateness Criteria which has been extensively covered in previous work.

Some comments below to help improve the paper:

General:

- Given that many studies have already leveraged appropriateness criteria in radiology, it is essential that your manuscript clearly delineate what is new. Please provide a more comprehensive literature review that contrasts your approach with previous work, and explicitly state how the RadCases dataset, the chosen LLM optimization methods, or the clinical evaluation pipeline advance current knowledge. Emphasize that the primary novelty is the RadCases dataset itself, which could merit renaming the manuscript to highlight this contribution.

Abstract:

- The phrase "...on par with clinicians" would benefit from specifying the metric (e.g., "accuracy on par with clinician recommendations").

Results

- p-values are reported for model comparisons using two-sample, one-tailed homoscedastic t-tests. Please clarify whether you adjusted for multiple comparisons (e.g., using a Bonferroni correction) and provide additional details regarding the statistical tests used.

Materials and Methods

- Page 24, 2nd paragraph RadCases dataset is described as leveraged from five different sources. Please specify the exact number of cases contributed by each source and detail the inclusion/exclusion criteria (e.g., "excluded if the one-liner did not refer to a specific patient presentation").

- Page 26, in the section "Labelling one-liners by ACR Appropriateness Criteria topics", there is a note that two fourth-year

medical students annotated the data under supervision. It would strengthen the manuscript to include inter-rater reliability metrics or a brief description of how disagreements were resolved (e.g., "In cases of disagreement, the attending radiologist's decision was final").

- page 27, the description of the chain-of-thought (COT) prompting strategy is somewhat opaque. Including a concrete example of an optimized prompt with the accompanying chain-of-thought would greatly enhance clarity.
- page 29, LoRA and QLoRA fine-tuning parameters are mentioned. Please justify the chosen parameters (e.g., rank and α scaling values) and, if possible, reference related prior work (such as <https://arxiv.org/abs/2106.09685>; <https://pubmed.ncbi.nlm.nih.gov/39873598/>) to support these decisions.

Figures and tables

- Table 1 lists four common failure modes. This is not clearly defined as "Hallucination" may not be an accurate description of how or why it is failing (retrieval of information) and overall described vaguely. Please refer to established literature for definitions of errors (for example: <https://hdr.mitpress.mit.edu/pub/1yo82mqa/release/2>) and consider adding frequency of such errors to help quantify their impact, with clearer tie to any mitigating steps.
- Figure 5 has a good caption, however is not clearly discussed/described in the accompanying text. There is retrospective comparison of clinicians' imaging orders with those generated by the LLM. While performance metrics (accuracy, F1, FPR, FNR) are provided, please expand on the clinical significance of these differences. For instance, discuss whether a 10–20% improvement in imaging accuracy is likely to translate into measurable benefits for patient outcomes or workflow efficiency.
- The prospective study included 23 medical students and 7 resident physicians—a relatively small sample. Please discuss the limitations of this sample size and its implications for the generalizability of the findings. Also, provide a rationale for the chosen monetary incentives and comment on any potential bias introduced by the recruitment strategy.

Discussion

- Page 22 There is mention of privacy concerns related to proprietary APIs. Consider elaborating on any safeguards or mitigation strategies applied in your experiments.

Novelty and Contextualization Relative to Existing Literature:

In summary, while your study presents an innovative framework and promising preliminary results, at a minimum major revisions are needed to improve methodological transparency, clarity of presentation, and to clearly articulate the novelty and clinical impact relative to existing work. In particular, the manuscript would benefit from a refocusing on the RadCases dataset as the novel contribution. Addressing these points will substantially strengthen your manuscript.

Version 1:

Reviewer comments:

Reviewer #1

(Remarks to the Author)

Reviewer #3

(Remarks to the Author)

Thank you for addressing my prior comments. I appreciate the effort made to clarify the paper, improve the methods, and refine the discussion/conclusions. The additions, including expanded results and discussion of ACR AC topic structure/guideline changes, and validation of labeling with clinical experts further strengthen the manuscript. I do not have any additional comments.

I also reviewed Reviewer 4's initial comments along with the authors' responses and corrections, and I believe these have been adequately addressed.

To: Andreia Cunha, PhD
andreia.cunha@nature.com
Chief Editor
Nature Communications Medicine
The Campus, 4 Crinan Street
London N1 9XW, United Kingdom

November 26, 2024

Dear Dr. Andreia Cunha:

I would like to submit our manuscript titled “Evaluating Image Ordering for Acute Patient Presentations via Language Model Alignment with the ACR Appropriateness Criteria” to be considered for publication as an article in Nature Communications Medicine.

Our work tackles the problem of image ordering in the emergency room, and how large language models can be leveraged to mitigate this issue. We introduce a new method to intelligently query language models and arrive at imaging recommendations for patients that are consistent with evidence-based medical guidelines. To validate our proposed work, we evaluate state-of-the-art language models on a novel set of challenging patient cases, and also conduct a prospective randomized clinical trial that demonstrates how our language model inference strategy can be used as clinical assistants to improve clinician diagnostic imaging workflows.

We assert that our manuscript is original, has not been published before, and is not currently being considered for publication elsewhere. We have also made all data and code required to reproduce our empirical results publicly available in association with our submission to ensure reproducibility of our work. Our paper was also previously considered for publication in Nature Medicine (reference number: NMED-A136194); we have included the comments from the Nature Medicine reviewers, and our response to their comments, below. We have no conflicts of interest associated with this publication, nor is there significant financial support for this work that could have influenced its outcome.

Thank you for your time and consideration of this manuscript.

Sincerely,

Osbert Bastani

General Response:

We sincerely thank the Reviewers for their careful review of our manuscript, and insightful comments and feedback which we believe have substantially improved the quality of our work. We appreciate that the Reviewers find our work significant (Reviewer 1) and extensive (Reviewer 3), and believe that our revised manuscript will be of interest to Nature Communications Medicine readership.

We have made a number of major revisions to the manuscript that we detail here, and include a point-by-point response to individual comments below in blue text.

- We have updated the title of our manuscript to “Evaluating Image Ordering for Acute Patient Presentations via Language Model Alignment with the ACR Appropriateness Criteria” to better reflect the scope of our work.
- We have made significant changes to the Introduction, Discussion, and overall organization of our manuscript to improve the readability and better frame and motivate our contributions presented in our manuscript.
- We have included additional experimental results from our retrospective study comparing autonomous LLMs with clinicians in **Figure 5** of our revised manuscript. Importantly, our results demonstrate that when setting the maximum number of ACR AC Topics requested by an LLM to 1, **LLMs can achieve a dominant strategy with respect to clinicians in our retrospective study**: for example, Llama 3 achieve a better accuracy and F1 score, and reduces the rate of missed imaging studies, without statistically significant increases in the rate of unnecessary imaging or the average number of imaging studies ordered. This are new findings previously not reported in our original submission that we believe will be of interest to the Reviewers. We summarize our findings as follow (lifted from our Discussion section):

“Our study investigates the potential of LLMs in the domain of diagnostic image ordering—a task critical to the timely and appropriate management of acute patient presentations. Our results demonstrate how state-of-the-art language models can be used in the context of diagnostic image ordering in acute clinical settings, such as the emergency department. Firstly, we observed that generalist language models—such as Claude Sonnet-3.5 and Meta Llama 3—can accurately predict relevant ACR AC Topic labels to describe patient one-liner descriptions without any domain-specific fine-tuning. By leveraging LLMs to predict Topic labels instead of imaging studies directly, we achieved significant improvements in the quality of final imaging recommendations made by LLMs. Comparing language models with clinicians in a retrospective study, we show that LLMs achieve better accuracy with regards to image ordering in the ED without significant changes to the rate of missed imaging (FNR), rate of unnecessary imaging (FPR), or number of recommended imaging studies. Finally, we demonstrate that LLMs can be leveraged by clinicians as a CDS assistant to improve the accuracy of ordered imaging studies without significant changes to the FPR or FNR in a simulated acute care environment.”

Figure 5: Retrospective Study of Clinician-Ordered versus LLM-Ordered Imaging Studies. We compare the diagnostic imaging studies ordered by the prompt-optimized LLMs Claude Sonnet-3.5 and Llama 3 against those ordered by clinicians in a retrospective study. We vary the maximum number m of ACR AC Topic predictions requested from each language model on the x-axis. Compared with clinicians, Claude Sonnet-3.5 and Llama 3 achieve better (A) accuracy scores; and (C) false negative rates (i.e., the rate at which a patient should have received an imaging workup but did not); and also achieve (B) false positive rates (i.e., the rate at which a patient received at least one unnecessary imaging recommendation); (D) F1 scores when $m = 1$; and (E) number of recommended imaging studies that are noninferior to that of clinicians. (F) According to the Dice-Sørensen Coefficient (DSC) metric, Claude Sonnet-3.5 and Llama 3 order

imaging studies that are more similar to one another than to clinicians across all values of m (two sample, two-tailed homoscedastic t -test; $p < 0.0001$).

- We have included additional experimental results on evaluating medically fine-tuned language models, such as Meditron, BioMedGPT, and Me-LLaMA, in **Supplementary Figure 8**. These new results suggest that these medical foundation models struggle on ACR AC Topic prediction according to the RadCases dataset when compared with generalist language models, highlighting an opportunity for future work on better medical fine-tuning strategies that help LLMs better generalize to new domain-specific tasks.

Supplementary Figure 8: Evaluating Medical Foundation Models Fine-Tuned on Llama LLMs. Separate from the results presented in **Supplementary Figure 7**, an alternative approach to model fine-tuning is to instead leverage language models fine-tuned on large corpuses of domain-specific medical text. Examples of such foundation models include *BioMedGPT-7B^{SF8-1}*; *MeLLaMA-70B^{SF8-2}*; and *Meditron-70B^{SF8-3}*. We evaluate their accuracies on predicting correct ACR AC Topic labels for RadCases datasets; none of the three medical foundation models evaluated outperformed the base Meta Llama 3 70B model with statistical significance. Our results are consistent with findings reported by prior work^{SF8-4-6} and highlight the challenge in fine-tuning language models specifically for RadCases and other medical tasks. Error bars represent \pm 95% CI over $n = 5$ independent experimental runs.

[SF8-1] Zhang K, Zhou R, Adhikarla E, et al. A generalist vision-language foundation model for diverse biomedical tasks. *Nat Med.* (2024). doi: 10.1038/s41591-024-03185-2

[SF8-2] Xie Q, Chen Q, Chen A, et al. Me LLaMA: Foundation large language models for medical applications. *arXiv Preprint.* (2024). doi: 10.48550/arXiv.2402.12749

[SF8-3] Chen Z, Cano AH, Romanou A, et al. *MEDITRON-70B: Scaling medical pretraining for large language models*. *arXiv Preprint*. (2023). doi: 10.48550/arXiv.2311.16079

[SF8-4] Jeong DP, Garg S, Lipton ZC, Oberst M. *Medical adaptation of large language and vision-language models: Are we making progress?* *Proc Emp Nat Lang Proc* (2024). doi: 10.48550/arXiv.2411.04118

[SF8-5] Dorfner FJ, Dada A, Busch F, Makowski MR, Han T, Truhn D, Kleesiek J, Sushil M, Lammert J, Adams LC, Bressemer KK. *Biomedical large language models seem not to be superior to generalist models on unseen medical data*. *arXiv Preprint* (2024). doi: 10.48550/arXiv.2408.13833

[SF8-6] Hager P, Jungmann F, Holland R, Bhagat K, Hubrecht I, Knauer M, Vielhauer J, Makowski M, Braren R, Kaissis G, Rueckert D. *Evaluation and mitigation of the limitations of large language models in clinical decision making*. *Nat Med* 30: 2613-22. (2024). doi: 10.1038/s41591-024-03097-1

- We have included additional experimental results to better quantify the validity of our RadCases dataset in **Supplementary Table 2**. These results indicate that our RadCases dataset is indeed representative of true one-liners according to common quantitative natural language processing techniques. Qualitatively, all of the RadCases one-liners in our dataset were reviewed by at least one U.S. attending physician and two U.S. final-year medical students to ensure that the dataset is accurately labeled and representative of one-liners in clinical practice: we include this discussion in our revised manuscript.

	Max Similarity	Mean Similarity	Perplexity	Token Count
True One-Liners	100 ⁽¹⁰⁰⁻¹⁰⁰⁾	40.0 ^(38.7-41.3)	115 ^(81.1-148.5)	51.3 ^(44.2-58.3)
arXiv NLP	13.8 ^(13.7-13.9)	6.09 ^(6.04-6.13)	30.0 ^(29.7-30.2)	171 ⁽¹⁷⁰⁻¹⁷²⁾
Wikitext	14.3 ^(14.1-14.4)	6.57 ^(6.48-6.66)	134⁽¹²⁷⁻¹⁴¹⁾	101⁽⁹⁹⁻¹⁰³⁾
PubMed	16.5 ^(16.4-16.6)	8.11 ^(8.04-8.19)	21.5 ^(21.2-21.7)	224 ⁽²²²⁻²²⁷⁾
MedQA	38.4 ^(38.2-38.6)	25.1 ^(24.9-25.2)	14.6 ^(14.5-14.7)	161 ⁽¹⁶⁰⁻¹⁶³⁾
MIMIC-IV Random	31.3 ^(30.1-32.6)	19.4 ^(18.5-20.4)	624 ⁽³¹⁸⁻⁹²⁹⁾	26.4 ^(20.8-32.0)
MIMIC-IV Radiology	26.7 ^(25.3-28.0)	16.1 ^(15.0-17.2)	437 ⁽³²³⁻⁵⁵²⁾	19.3 ^(16.1-22.6)
MIMIC-IV Full Note	37.3 ^(36.5-38.2)	26.1 ^(25.6-26.7)	40.6 ^(39.0-42.1)	3220 ⁽³⁰⁴⁰⁻³³⁹⁰⁾
MIMIC-IV Test	49.4 ^(48.8-50.1)	33.5 ^(33.0-33.9)	317 ⁽²⁶⁵⁻³⁷⁰⁾	40.1 ^(38.5-31.8)
RadCases Synthetic	52.2^(51.0-53.4)	35.3^(34.7-35.9)	60.3^(47.8-72.8)	23.6 ^(22.7-24.4)
RadCases USMLE	47.4 ^(45.9-48.9)	29.9 ^(29.0-30.9)	14.0 ^(12.9-15.2)	21.0 ^(20.1-21.8)
RadCases JAMA	43.1 ^(42.6-43.7)	27.9 ^(27.5-28.2)	20.0 ^(19.2-20.8)	33.0 ^(32.3-33.8)
RadCases NEJM	42.3 ^(41.2-43.4)	27.3 ^(26.6-28.1)	15.0 ^(14.1-16.0)	51.1^(49.1-53.2)
RadCases BIDMC	65.1^(62.1-68.2)	36.2^(35.4-37.2)	173⁽¹⁴⁰⁻²⁰⁷⁾	45.8^(42.0-50.0)

Supplementary Table 2: Comparing the RadCases Dataset with Ground-Truth Patient One-Liner Case Summaries. To validate our RadCases dataset, we first had 3 independent U.S. attending physicians review a set of 50 true one-liners and confirm that they are representative of real-world patient case summaries used in clinical practice. We then computed the (1) Maximum and (2) Mean Similarity Score using the NV-Embed-v2 Retriever, between each of the RadCases datasets and a dataset of true one-liners derived from real patient cases. We also computed the average (3) Perplexity according to GPT-2 Large Medical; and (4) the average number of tokens per one-liner according to the GPT-4o tokenizer. We compare RadCases against other corpora such as arXiv computer science abstracts (**arXiv NLP**); Wikipedia articles (**Wikitext**); **PubMed** articles; and the **MedQA** dataset. Finally, we also compare against **Random** sentences admission notes in the MIMIC-IV dataset; random sentences from **Radiology** imaging reports in the MIMIC-IV dataset, **Full Admission Notes** from the MIMIC-IV dataset; and a separate **Test** set of extracted patient one-liners from the MIMIC-IV dataset. Each metric is reported as $[Mean^{95\% CI}]$, where $[Mean]$ is the mean metric value, and $[95\% CI]$ is the 95% confidence interval. The best (resp., second best) values in each column—and all values with intersecting mean confidence intervals—are **bolded** (resp., underlined). Our results demonstrate that RadCases is a promising dataset of simulated patient one-liners compared with other domain-specific text corpora.

Point-by-Point Response to Comments:

Reviewer #1 (Remarks to the Author):

A. The paper shows that large language models (LLM) have the potential to assist clinicians to make imaging study decisions to patients while complying the ACR AC guideline. The paper reports that by presenting a patient case in one sentence to LLM and prompting it to match a topic in the ACR AC as an "evidence" before prompting it to recommend an imaging study works better than prompting LLM to recommend an imaging study. The paper shows that evidence-based LLM prompting performed on a par with clinicians.

B. The work is original to my knowledge, and significant.

We thank the Reviewer for their feedback and comments, which we believe have significantly improved the quality of our proposed work.

C. The study used public domain data to create one line patient case descriptions and had experienced physicians as curators to create ground truth topic matching and imaging study recommendations. However, the data chosen for the study may not be adequate. Notably the choice of discharge notes as the input (see below).

We thank the Reviewer for this comment, and agree with the Reviewer that it is crucial to encapsulate the limited information about a given patient in evaluating language models for our

intended use case. We have updated our manuscript and results to use the admission notes as our source of patient case summaries in the Methods section of our manuscript:

“We constructed the BIDMC dataset from anonymized, de-identified patient admission notes introduced by Johnson et al.²⁶ in the MIMIC-IV dataset by taking the first sentence of each clinical note as the patient one-liner.”

Of note, none of the main results or extracted one-liners changed as a result of using admissions notes instead of discharge summaries. This is because a widespread practice in clinical medicine is to directly copy over the history of present illness (HPI) of the original admission note directly to the discharge summary for the same patient admission unedited, meaning that the extracted patient description is equivalent in virtually all cases whether the discharge summary or admission note is used.

D. All comparisons include statistical tests, though it is not clear if multiple comparisons were accounted when determining test significance.

We have consulted with two statisticians at the University of Pennsylvania and determined that multiple comparisons corrections are not required in our work.

E. The conclusions stated in the end of Discussion are valid, with sufficient support of experimental results.

F. Suggested improvements:

-- Title: Using the phrase "XXX is all you need" in the title has become somewhat of a cliché. There have been too many of them, making the phrase feel overused and could detract from the originality of your work.

We thank the reviewer for this comment. We have updated the title of our manuscript to “Evaluating Image Ordering for Acute Patient Presentations via Language Model Alignment with the ACR Appropriateness Criteria” to better encapsulate our contributions.

-- "One-liner" is commonly used to refer to a short joke and may not be a right choice of a term to refer to a short one line description of a patient's condition.

Thank you for this comment. “Patient one-liners” is a common term used across all of clinical medicine. For example, a cursory review of the literature shows this term being used in Internal Medicine, academic medical societies such as the AAMC, Psychiatry, and medical education. Because many readers of Nature Medicine and Nature Communications Medicine come from medical and medical-adjacent backgrounds, we believe that labeling short patient descriptions as one-liners is both accurate and appropriate.

Nonetheless, we acknowledge the reviewer's comment and recognize that readers of Nature journals often come from diverse backgrounds. We have clarified the definition of the one-liner explicitly for the reader in our introduction when the "one-liner" term is first introduced in our Introduction (relevant revisions are underlined):

"In this work, we investigate how state-of-the-art LLMs can be used as potential CDS tools to reduce the problem of inappropriate image ordering according to the ACR AC. Our core hypothesis is that while LLMs may struggle to directly recommend imaging studies for patients (a domain-specific task) (Fig. 1), they are often able to accurately describe patient conditions and presentations. In this light, we apply LLMs to analyze patient case summaries and map them to topic categories from the ACR AC. We represent these case summaries as "patient one-liners," which are concise summaries of patient presentations commonly used by clinicians to communicate relevant details quickly to other healthcare providers.^{30,31} We can then programmatically query the ACR AC based on the LLM-recommended topic category (without any explicit LLM usage) to determine the optimal imaging study for a patient. In this fashion, LLMs can be used to recommend diagnostic imaging studies according to recommendations from the guidelines."

-- Line 131: does this mean selecting the first line of a case note?

Yes, the reviewer's assessment is correct here. We have updated the relevant wording in line 131 to better clarify this:

"... we follow the same protocol as for the Medbullets dataset described above to programmatically convert these document-form patient cases into patient one-liners by using the first sentence of each patient case."

-- The use of discharge summaries: When a discharge summary is given the patient is fully evaluated and diagnosed and treated and ready to leave. While a decision for an imaging study is needed when a patient is not fully evaluated. This is a mismatch of a real use case and may given your LLM advantage of knowing more about a patient. Instead, admission notes may be more appropriate. An experiment using admission notes, which should be available in MIMIC data sets, to clarify if the use of note types may lead to different results, is necessary.

We thank the Reviewer for this comment, and agree with the Reviewer that it is crucial to encapsulate the limited information about a given patient in evaluating language models for our intended use case. We have updated our manuscript and results to use the admission notes as our source of patient case summaries in the Methods section of our manuscript (relevant revisions are underlined):

"We constructed the BIDMC dataset from anonymized, de-identified patient admission notes introduced by Johnson et al.²⁶ in the MIMIC-IV dataset by taking the first sentence of each clinical note as the patient one-liner."

Of note, none of the main results or extracted one-liners changed as a result of using admissions notes instead of discharge summaries. This is because a widespread practice in clinical medicine is to directly copy over the history of present illness (HPI) of the original admission note directly to the discharge summary for the same patient admission unedited, meaning that the extracted patient description is equivalent in virtually all cases whether the discharge summary or admission note is used.

-- Line 142: though the authors admitted that this is a limitation, it is not clear why it is not possible to include those matching no ACR AC topic and see if LLM can simply respond that this case has no matching ACR AC topic? If you can exclude them, you have the "labels" and should be feasible to test LLM.

We thank the Reviewer for this thoughtful comment. Assessing for the ability of language models (and machine learning methods in general) to identify whether a potential output is out-of-scope relative to a set of clinical guidelines is a challenging task: in particular, it has been well-characterized in recent work [1, 2, 3] that existing language models are often falsely over-confident in their outputs and are unable to reliably tell if it is unable to respond accurately given an input prompt. Empirically, this is also what we observed in our initial experiments (not reported in this manuscript): the LLMs that we evaluated often predicted almost random ACR AC topics to irrelevant input patient case summaries, instead of returning that the case has no matching ACR AC topic.

We emphasize that the primary objective of our study is to evaluate the alignment of language model outputs with existing ACR AC guidelines, *not* to evaluate whether a language model can identify whether a set of guidelines is applicable or not. This is a nuanced yet critical distinction in the scope of our work that precludes the evaluation of cases that have no relevant ACR AC topic. Our focus is on assessing appropriateness based established criteria, rather than on identifying gaps where criteria do not exist.

We agree with the Reviewer in the importance of future work to enable LLMs to more reliably respond with “negative responses” where guidelines might not be appropriate. We have included additional discussion of this in our Discussion section to better highlight this opportunity for future work (relevant revisions are underlined):

“Firstly, while constructing the RadCases dataset used to evaluate LLMs, we excluded cases where there exist no ACR AC guidelines relevant to the patient scenario. Such cases might include primary dermatologic conditions and cases where insufficient ACR evidence exist. While this strategy allowed us to primarily focus on how existing LLMs perform according to the ACR AC, future work is warranted to investigate how these models handle scenarios where there is a lack of relevant guidelines.”

[1] Liu F et al. Large language models are poor clinical decision makers: A comprehensive benchmark. Proc Emp Nat Lang Proc: 13696-710. (2024). <https://aclanthology.org/2024.emnlp-main.759/>

[2] Agarwal C, Tanneru SH, Lakkaraju H. Faithfulness vs plausibility: On the (un)reliability of explanations from large language models. arXiv Preprint. (2024).

<https://doi.org/10.48550/arXiv.2402.04614>

[3] Omar M et al. Benchmarking the confidence of large language models in clinical questions. medRxiv Preprint. (2024). <https://www.medrxiv.org/content/10.1101/2024.08.11.24311810v2>

-- Line 248 "prompt engineering" was introduced here it is not clear if that refers to which optimization methods? Suggest stick with ICL, COT etc. that have been discussed in the previous paragraphs.

We thank the reviewer for this comment. We have removed the term "prompt engineering" and in other relevant portions of our manuscript, and stick with ICL, COT, etc. instead.

-- Fig 1, 2 the coloring and the legends of the bars are difficult for my aging eyes to tell the difference. Suggest changing colors and consider color-blind friendly color schemes.

Thank you for this comment - we have updated the presentation of Figures 1 and 2 to use more color-blind friendly color schemes for easier legibility.

-- Line 314: Same issue as Line 142, especially when 42% cases have to be excluded.

Please see our response to the Reviewer's comment regarding Line 142 above.

-- Fig 3: How can this be better than Figure 2? So it is possible that a wrong matching of ACR AC topic may still lead to a correct recommendation of imaging study? It is important to also explain how many different types of imaging studies are included in ACR AC, and if these types include no imaging study necessary etc. It is also a question whether it should be considered as a true positive, if the model chooses a wrong ACR AC topic but ends up with a correct imaging study recommendation.

Thank you for this comment. Yes, it is possible that a wrong matching of the ACR AC Topic may still lead to a correct recommendation of the final imaging study. This is because an incorrect ACR AC Topic that is "similar enough" to the ground truth Topic such that the final imaging study is still accurate. We detail this in our manuscript as follows:

““““

Interestingly, while the Evidence-Based Optimized pipeline significantly outperformed the Baseline pipeline on ACR AC Topic classification accuracy (**Fig. 3b**), we did not observe a statistically significant improvement in the optimized versus baseline Evidence-Based pipelines on the imaging classification accuracy (two-sample, one-tailed homoscedastic *t*-test; $p \geq 0.346$ for each of the 4 RadCases dataset subsets). Qualitatively, we found that although the Evidence-Based Baseline pipeline achieved a lower ACR AC Topic classification accuracy compared to the Evidence-Based Optimized inference strategy, its incorrect predictions were still closely related to the correct answer and underlying patient pathology. For example, a

ground truth ACR AC Topic might be “Major Blunt Trauma” and the LLM prediction “Penetrating Torso Trauma;” although the LLM identified the incorrect ACR AC Topic label, both Topics warrant a “Radiography trauma series.”
”””

Given that our ultimate goal is to predict imaging studies using our LLM-based inference strategy, and we use the ACR AC Topic classification task solely as an intermediate step to predict imaging studies, we elected to still consider these cases of correct imaging study but incorrect ACR AC Topic as a true positive in our framework.

-- Line 335: 3 here now how did a true positive determined? as long as one of the top 3 matches the ground truth? Did physicians get to pick their top 3? If LLM is correct when one of top 3 matches, while clinicians only pick 1, how this can be a fair comparison as shown in Fig 4?

We thank the Reviewer for their valuable feedback. Because we looked to historical patient documentation to deduce the imaging study(s) ordered by physicians in our retrospective study, the number of studies ordered by physicians is fixed. As a result, we agree with the Reviewer that fixing the number of studies proposed by LLMs may offer an incomplete picture of the relative performance of autonomous LLMs compared with clinicians.

To better interrogate the performance of the LLMs as a function of the number of proposed imaging studies, we have included additional experiments in **Figure 5** of our updated manuscript where we vary the maximum number of ACR AC Topics requested by an LLM agent in image ordering. This enables us to explore the range of empirical tradeoffs that occur as a function of the number of imaging study recommendations made by LLMs. We include Figure 5 in our updated manuscript below for easy reference:

“Figure 5: Retrospective Study of Clinician-Ordered versus LLM-Ordered Imaging Studies. We compare the diagnostic imaging studies ordered by the prompt-optimized LLMs Claude Sonnet-3.5 and Llama 3 against those ordered by clinicians in a retrospective study. We vary the maximum number m of ACR AC Topic predictions requested from each language model on the x-axis. Compared with clinicians, Claude Sonnet-3.5 and Llama 3 achieve better **(A)** accuracy scores; and **(C)** false negative rates (i.e., the rate at which a patient should have received an imaging workup but did not); and also achieve **(B)** false positive rates (i.e., the rate at which a patient received at least one unnecessary imaging recommendation); **(D)** F1 scores when $m = 1$; and **(E)** number of recommended imaging studies that are noninferior to that of clinicians. **(F)** According to the Dice-Sørensen Coefficient (DSC) metric, Claude Sonnet-3.5 and

Llama 3 order imaging studies that are more similar to one another than to clinicians across all values of m (two sample, two-tailed homoscedastic t -test; $p < 0.0001$).”

Based on these additional experiments (specifically, when setting the maximum number of ACR AC Topics requested by an LLM to 1), **we found that LLMs can achieve a dominant strategy with respect to clinicians in our retrospective study: for example, Llama 3 achieve a better accuracy and F1 score, and reduces the rate of missed imaging studies, without statistically significant increases in the rate of unnecessary imaging or the number of imaging studies ordered.** We believe these new experimental results better support our conclusion that LLMs can serve as effective clinical decision support tools. We include the relevant updates to the revised manuscript below:

““““

Our results suggest that autonomous LLMs are effective tools in ordering diagnostic imaging: when LLMs were queried to return up to $m = 1$ relevant ACR AC Topics, Claude Sonnet-3.5 achieved a higher accuracy score of 58.0% and F1 score of 94.4% compared with clinicians (accuracy 39.3%; F1 score 92.5%) (McNemar test; $p = 0.044$). There was no statistically significant difference between Llama 3 (accuracy 61.5%; F1 score 92.9%) and clinicians (McNemar test; $p = 0.099$) for the same value of m . Across the patient cases assessed, clinicians ordered an average of 1.41 (95% CI: [1.28 – 1.53]) imaging studies per case; similarly, Claude Sonnet-3.5 ordered an average of 1.83 (95% CI: [1.60 – 2.06]) and Llama 3 an average of 1.54 (95% CI: [1.33 – 1.76]) studies per case. There was no statistically significant difference between the number of imaging studies ordered by Llama 3 and its clinician counterparts (two-sample paired t -test; Llama 3: $p = 0.269$).

We also evaluated the rates of both unnecessary and missed imaging studies. For $m = 1$, both Claude Sonnet-3.5 and Llama-3 were non-inferior to clinicians according to both metrics, achieving a false positive rates (FPR) of 6.90% and 6.90% (clinician FPR = 7.76%) (McNemar test; $p = 1.00$) and false negative rates (FNR) of 3.45% and 6.03% (clinician FNR = 6.03%) (McNemar test; Llama 3: $p = 1$; Claude Sonnet-3.5: $p = 0.549$), respectively (**Fig. 4b-c**). As expected, the rate of unnecessary imaging studies increased and the rate of missed imaging studies decreased as the maximum number of queried ACR AC Topics increased. Altogether, these results suggest that LLMs are promising tools for image ordering in clinical workflows.

””””

-- Line 341: if LLM resulted in more unnecessary imaging studies, wouldn't this beat the purpose as the motivation given in the Introduction?

We thank the Reviewer for this comment. Based on our additional experimental results (i.e., **Figure 5** in our updated manuscript) discussed above, we have found that LLMs do not result in more unnecessary imaging studies when the maximum number of requested imaging studies from the language models is set to $m = 1$.

Nonetheless, based on the feedback from the Reviewers, we have significantly rewritten the Introduction and other relevant components of our manuscript. We include the relevant excerpt from our updated Introduction below for easy reference:

“Inappropriate ordering of diagnostic imaging studies is a commonly encountered problem in the emergency department (ED) and other acute-care settings.¹⁻⁴ While diagnostic imaging can play a crucial role in the acute workup of patients, inappropriate or unnecessary imaging studies present mounting concerns regarding resource utilization, radiation exposure, and financial burden to both patients and healthcare systems.⁵⁻⁷ Recent estimates suggest that up to 30% of diagnostic imaging studies ordered in the ED setting could be replaced with more appropriate alternatives.^{6,8}

Multiple factors contribute to this increasing trend of inappropriate imaging. Importantly, emergency medicine physicians often need to make rapid diagnostic decisions with limited clinical context while simultaneously managing high patient volumes and complex patient presentations. Furthermore, there is significant variability between healthcare providers in imaging ordering patterns: recent studies have documented significant differences in the utilization rates of different imaging studies between individual emergency physicians, suggesting that factors beyond pure clinical necessity influence imaging decisions.⁹⁻¹¹ To help clinicians make more informed, evidence-based decisions in image ordering and simultaneously address inter-provider variability in imaging practices, the American College of Radiology (ACR) released the ACR Appropriateness Criteria® (ACR AC), which are a set of evidence-based guidelines that assist referring physicians in ordering the most appropriate diagnostic imaging studies for specific clinical conditions.¹² As of its most recent online release, the ACR AC contains 263 unique imaging topics (i.e., patient scenarios).

However, despite the widespread availability of the ACR AC, improper imaging according to the guidelines remains a challenge in many emergency departments and inpatient settings.^{3,13} Bautista et al. showed that there is low utilization of the ACR AC by clinicians in practice: less than 1% of physicians interviewed in their study use the ACR AC as a first-line resource when ordering diagnostic imaging studies.^{14,15} The limited usage of the ACR AC may be partly due to how the Appropriateness Criteria are presented to clinicians; the evidence-based criteria are dense and can be difficult to parse through even for physician experts, and especially in acute healthcare settings such as the emergency department where decision making is both time-sensitive and critical.”

-- Line 400: Are these LLM generated cases? I think this should be discussed in the limitation section.

The cases used in our human user study are not LLM-generated. We have better clarified this point in our text in the Methods section of our updated manuscript:

“Constructing the patient cases for prospective user evaluation study

To enable our prospective evaluation of LLMs as clinical decision support tools for clinicians, we first constructed a separate dataset of 50 patient one-liners derived from the RadCases BIDMC one-liners. The initially redacted details such as patient name, age, or gender were manually

replaced with fake name, age, and/or gender values. The cases were then reviewed and edited by three separate attending physicians to ensure that the cases were representative of real-world patient cases that commonly present in the emergency room."

--

G. References are appropriate.

H. The presentation is clear, though there are minor issues that can be addressed in the revision.

Reviewer #1 (Remarks on code availability):

The codes were nicely packaged to be easily installed and run following README.md. But it is difficult to set the arguments to run "main.py" without lists of proper values to choose from. It would be nice if lists of pre-set arguments to produce results as shown in Figure 1, 2, 3 in the paper. That said, this is much better than my review experience with other publication venues and i appreciate the efforts made by the authors to share the codes.

We thank the Reviewer for their remarks. In general, it is challenging for us to include all the required constructed datasets to be able to easily allow the interested reader to reproduce our work using a single script. This is because many of the datasets we use cannot be released publicly with our work due to licensing restraints and/or data use and access requirements imposed by the original dataset providers. For example, in the paper that originally introduces the JAMA Clinical Challenges dataset [1], the authors do not make available the actual dataset due to JAMA license constraints. As a result, special steps must be taken as outlined in their implementation to gain access to the dataset. Similar constraints make it challenging to gain access to the NEJM and MIMIC-IV datasets as well. We detail the specific steps that interested readers must take to get setup to use our code in our README.md file associated with our repository, and also specifically describe the commands to run to reproduce our results once these initial setup steps are accomplished.

[1] Chen H et al. Benchmarking large language models on answering and explaining challenging medical questions. arXiv Preprint. (2024). <https://arxiv.org/abs/2402.18060v4>

Reviewer #1 (Remarks on figshare data availability):

Again, if the dataset can be labeled as Fig 1, 2, 3, 4... consistent with the headings given in the article, it will be easier for readers to reproduce the results.

Thank you for this comment. We will update the figshare to more clearly label which datasets are associated with which figures in our manuscript.

Reviewer #2 (Remarks to the Author):

A. Summary of the Key Results

The authors leverage large language models (LLMs) to recommend imaging studies for patient cases, aligned with evidence-based guidelines, using the curated RadCases dataset containing patient “one-liner” summaries. Both retrospective analysis and a prospective randomized controlled trial demonstrate that LLMs perform comparably to clinicians in image ordering, and LLM-based clinical assistants can improve image ordering by clinicians in acute care settings.

We thank the Reviewer for their feedback and comments, which we believe have significantly improved the quality of our proposed work.

B. Originality and Significance

1. The authors attempt to simulate decision-making contexts with limited patient information in an acute clinical setting, which presents an interesting scenario. However, this is not clearly reflected in the title or abstract. To improve clarity, the authors should either include experiments with the full dataset or explicitly state the specific scope of the study in the abstract and title.

We thank the reviewer for this comment. We have updated the title of our manuscript to “Evaluating Image Ordering for Acute Patient Presentations via Language Model Alignment with the ACR Appropriateness Criteria” to better encapsulate our contributions. We have also clarified our scope of the study in our abstract.

2. The RadCases dataset simulates an acute clinical setting by combining GPT-3.5-generated data with the first sentence from publicly available patient cases. However, one-liner summaries do not fully represent a real acute clinical setting. If the authors claim it does, they should provide references or include human evaluation, as done in similar previous works [1].

[1] Hager, P., Jungmann, F., Holland, R., et al. Evaluation and mitigation of the limitations of large language models in clinical decision-making. *Nat Med* 30, 2613–2622 (2024).

<https://doi.org/10.1038/s41591-024-03097-1>

Thank you for this comment. Firstly, we note that all one-liners in our RadCases dataset were individually reviewed to be representative of true clinical one-liners in written and used by clinicians in acute clinical settings by at least one U.S. attending radiologist and two U.S. final-year medical students - we have updated the Methods section of our manuscript to make this clear to readership. To further evaluate our RadCases dataset, we also experimentally evaluated the curated one-liners in **Supplementary Table 2**, also included below:

	Max Similarity	Mean Similarity	Perplexity	Token Count
True One-Liners	100 ⁽¹⁰⁰⁻¹⁰⁰⁾	40.0 ^(38.7-41.3)	115 ^(81.1-148.5)	51.3 ^(44.2-58.3)
arXiv NLP	13.8 ^(13.7-13.9)	6.09 ^(6.04-6.13)	30.0 ^(29.7-30.2)	171 ⁽¹⁷⁰⁻¹⁷²⁾
Wikitext	14.3 ^(14.1-14.4)	6.57 ^(6.48-6.66)	134 ⁽¹²⁷⁻¹⁴¹⁾	101 ⁽⁹⁹⁻¹⁰³⁾
PubMed	16.5 ^(16.4-16.6)	8.11 ^(8.04-8.19)	21.5 ^(21.2-21.7)	224 ⁽²²²⁻²²⁷⁾
MedQA	38.4 ^(38.2-38.6)	25.1 ^(24.9-25.2)	14.6 ^(14.5-14.7)	161 ⁽¹⁶⁰⁻¹⁶³⁾
MIMIC-IV Random	31.3 ^(30.1-32.6)	19.4 ^(18.5-20.4)	624 ⁽³¹⁸⁻⁹²⁹⁾	26.4 ^(20.8-32.0)
MIMIC-IV Radiology	26.7 ^(25.3-28.0)	16.1 ^(15.0-17.2)	437 ⁽³²³⁻⁵⁵²⁾	19.3 ^(16.1-22.6)
MIMIC-IV Full Note	37.3 ^(36.5-38.2)	26.1 ^(25.6-26.7)	40.6 ^(39.0-42.1)	3220 ⁽³⁰⁴⁰⁻³³⁹⁰⁾
MIMIC-IV Test	49.4 ^(48.8-50.1)	33.5 ^(33.0-33.9)	317 ⁽²⁶⁵⁻³⁷⁰⁾	40.1 ^(38.5-31.8)
RadCases Synthetic	52.2 ^(51.0-53.4)	35.3 ^(34.7-35.9)	60.3 ^(47.8-72.8)	23.6 ^(22.7-24.4)
RadCases USMLE	47.4 ^(45.9-48.9)	29.9 ^(29.0-30.9)	14.0 ^(12.9-15.2)	21.0 ^(20.1-21.8)
RadCases JAMA	43.1 ^(42.6-43.7)	27.9 ^(27.5-28.2)	20.0 ^(19.2-20.8)	33.0 ^(32.3-33.8)
RadCases NEJM	42.3 ^(41.2-43.4)	27.3 ^(26.6-28.1)	15.0 ^(14.1-16.0)	51.1 ^(49.1-53.2)
RadCases BIDMC	65.1 ^(62.1-68.2)	36.2 ^(35.4-37.2)	173 ⁽¹⁴⁰⁻²⁰⁷⁾	45.8 ^(42.0-50.0)

Supplementary Table 2: Comparing the RadCases Dataset with Ground-Truth Patient One-Liner Case Summaries. To validate our RadCases dataset, we first had 3 independent U.S. attending physicians review a set of 50 true one-liners and confirm that they are representative of real-world patient case summaries used in clinical practice. We then computed the (1) Maximum and (2) Mean Similarity Score using the NV-Embed-v2 Retriever, between each of the RadCases datasets and a dataset of true one-liners derived from real patient cases. We also computed the average (3) Perplexity according to GPT-2 Large Medical; and (4) the average number of tokens per one-liner according to the GPT-4o tokenizer. We compare RadCases against other corpora such as arXiv computer science abstracts (**arXiv NLP**); Wikipedia articles (**Wikitext**); **PubMed** articles; and the **MedQA** dataset. Finally, we also compare against **Random** sentences admission notes in the MIMIC-IV dataset; random sentences from **Radiology** imaging reports in the MIMIC-IV dataset, **Full Admission Notes** from the MIMIC-IV dataset; and a separate **Test** set of extracted patient one-liners from the MIMIC-IV dataset. Each metric is reported as [Mean^{95% CI}], where [Mean] is the mean metric value, and [95% CI] is the 95% confidence interval. The best (resp., second best) values in each column—and all values with intersecting mean confidence intervals—are **bolded** (resp., underlined). Our results demonstrate that RadCases is a promising dataset of simulated patient one-liners compared with other domain-specific text corpora.

Based on both our quantitative experimental results and expert-in-the-loop data curation process, we believe that our RadCases dataset will be of use to the broader scientific community and future work on LLMs for clinical decision support.

C. Data & Methodology: Validity of Approach, Quality of Data, Quality of Presentation

1. The authors validate their proposed workflow through retrospective and prospective experiments, showing that LLMs can recommend imaging studies according to guidelines. However, the experimental design relies on a key assumption: “LLMs may struggle to directly recommend imaging studies but can accurately describe patient conditions and presentations.” This assumption requires empirical validation or references. For example, Zaki, Hossam A., et al. [2] tested AI models on clinical scenarios from 11 ACR expert panels, with models achieving a high average score, indicating that LLM performance in imaging recommendation is better than implied by the authors. Additionally, the claim that “LLMs can accurately describe patient conditions and presentations” also needs validation.

We thank the Reviewer for this comment. Prior work by Zaki et al. and others evaluating LLMs for medical use cases have primarily reported results on inputs that are not representative of how clinicians discuss acute patient presentations in practice. As a result, such existing datasets cannot help us adequately interrogate the ability of LLMs to take “natural medical text” written by clinicians as input, and produce imaging recommendations that are aligned with the ACR Appropriateness Criteria. For example, in the aforementioned study by Zaki et al. [1], the authors include the inputs used to query the LLM in their Supplementary Data. We include the first three inputs copied verbatim here for easy reference:

1. Breast cancer screening. Average-risk women: women with <15% lifetime risk of breast cancer.
2. Breast cancer screening. Intermediate-risk women: women with personal history of breast cancer, lobular neoplasia, atypical ductal hyperplasia, or 15% to 20% lifetime risk of breast cancer.
3. Breast cancer screening. Average-risk women: women with <15% lifetime risk of breast cancer.

While the study by Zaki et al. and others are important, these textual data are not representative of real texts, such as clinical one-liners, that are written and used by clinicians in real clinical workflows. This discrepancy in the input text also helps to explain the ostensible difference in the performance of LLMs in our work versus what is reported in their study. We believe that our findings help demonstrate the importance of our RadCases dataset to help better evaluate the true clinical utility of language models in real-world acute clinical settings.

Furthermore, to better illustrate the subpar performance of language models on the RadCases dataset without the use of the inference strategy proposed in our work, we include **Figure 1** in our updated manuscript that shows the out-of-the-box performance of a state-of-the-art language model, Claude Sonnet-3.5 (the best performing language model in our experiments). Figure 1 helps to motivate our study, and provides clear experimental evidence that “LLMs may struggle to directly recommend imaging studies.” To see that “LLMs...can accurately describe patient conditions and presentations,” we point the Reviewer to **Figure 2b** in our manuscript. Furthermore, comparing the performance of language models in **Figure 4b-d**, we can see that LLMs that describe patient conditions and presentations as ACR AC Topics significantly outperform LLMs that directly recommend imaging studies: this helps demonstrate the utility of our inference strategy in leveraging LLMs to describe patient presentations instead of directly

recommending imaging studies. Finally, we have also cited the following works in our updated manuscript to help support this claim (new references are underlined below):

“In this work, we investigate how state-of-the-art LLMs can be used as potential CDS tools to reduce the problem of inappropriate image ordering according to the ACR AC. Our core hypothesis is that while LLMs may struggle to directly recommend imaging studies for patients (a domain-specific task) (Fig. 1), they are often able to accurately describe patient conditions and presentations [2-4]. In this light, we apply LLMs to analyze patient case summaries and map them to topic categories from the ACR AC.”

[1] Zaki et al. J Am Coll Radiol (2024). [https://www.jacr.org/article/S1546-1440\(24\)00056-5/fulltext](https://www.jacr.org/article/S1546-1440(24)00056-5/fulltext)

[2] Singhal K et al. Large language models encode clinical knowledge. Nature (2023). <https://www.nature.com/articles/s41586-023-06291-2>

[3] Williams CYK et al. Use of a large language model to assess clinical acuity of adults in the emergency department. JAMA Netw Open (2024). <https://jamanetwork.com/journals/jamanetworkopen/fullarticle/2818387>

[4] Williams CYK et al. Evaluating the use of large language models to provide clinical recommendations in the emergency department. Nat Commun (2024). <https://www.nature.com/articles/s41467-024-52415-1>

2. The dataset quality is a critical concern.

- A portion of the dataset was synthetically generated using GPT-3.5, but it is unclear if the synthetic data were verified by experts. Using synthetic data without expert validation raises concerns about its reliability. Moreover, GPT-3.5 is no longer the state-of-the-art model, and more advanced models like GPT-4 should be considered for generating synthetic data.

Thank you for this comment. Firstly, we note that all one-liners in our RadCases dataset were individually reviewed to be representative of true clinical one-liners in written and used by clinicians in acute clinical settings by at least one U.S. attending radiologist and two U.S. final-year medical students - we have updated the Methods section of our manuscript to make this clear to readership. To further evaluate our RadCases dataset, we also experimentally evaluated the curated one-liners in **Supplementary Table 2**, also included below:

	Max Similarity	Mean Similarity	Perplexity	Token Count
True One-Liners	100 ⁽¹⁰⁰⁻¹⁰⁰⁾	40.0 ^(38.7-41.3)	115 ^(81.1-148.5)	51.3 ^(44.2-58.3)
arXiv NLP	13.8 ^(13.7-13.9)	6.09 ^(6.04-6.13)	30.0 ^(29.7-30.2)	171 ⁽¹⁷⁰⁻¹⁷²⁾
Wikitext	14.3 ^(14.1-14.4)	6.57 ^(6.48-6.66)	134 ⁽¹²⁷⁻¹⁴¹⁾	101 ⁽⁹⁹⁻¹⁰³⁾
PubMed	16.5 ^(16.4-16.6)	8.11 ^(8.04-8.19)	21.5 ^(21.2-21.7)	224 ⁽²²²⁻²²⁷⁾
MedQA	38.4 ^(38.2-38.6)	25.1 ^(24.9-25.2)	14.6 ^(14.5-14.7)	161 ⁽¹⁶⁰⁻¹⁶³⁾
MIMIC-IV Random	31.3 ^(30.1-32.6)	19.4 ^(18.5-20.4)	624 ⁽³¹⁸⁻⁹²⁹⁾	26.4 ^(20.8-32.0)
MIMIC-IV Radiology	26.7 ^(25.3-28.0)	16.1 ^(15.0-17.2)	437 ⁽³²³⁻⁵⁵²⁾	19.3 ^(16.1-22.6)
MIMIC-IV Full Note	37.3 ^(36.5-38.2)	26.1 ^(25.6-26.7)	40.6 ^(39.0-42.1)	3220 ⁽³⁰⁴⁰⁻³³⁹⁰⁾
MIMIC-IV Test	49.4 ^(48.8-50.1)	33.5 ^(33.0-33.9)	317 ⁽²⁶⁵⁻³⁷⁰⁾	40.1 ^(38.5-31.8)
RadCases Synthetic	52.2 ^(51.0-53.4)	35.3 ^(34.7-35.9)	60.3 ^(47.8-72.8)	23.6 ^(22.7-24.4)
RadCases USMLE	47.4 ^(45.9-48.9)	29.9 ^(29.0-30.9)	14.0 ^(12.9-15.2)	21.0 ^(20.1-21.8)
RadCases JAMA	43.1 ^(42.6-43.7)	27.9 ^(27.5-28.2)	20.0 ^(19.2-20.8)	33.0 ^(32.3-33.8)
RadCases NEJM	42.3 ^(41.2-43.4)	27.3 ^(26.6-28.1)	15.0 ^(14.1-16.0)	51.1 ^(49.1-53.2)
RadCases BIDMC	65.1 ^(62.1-68.2)	36.2 ^(35.4-37.2)	173 ⁽¹⁴⁰⁻²⁰⁷⁾	45.8 ^(42.0-50.0)

Supplementary Table 2: Comparing the RadCases Dataset with Ground-Truth Patient One-Liner Case Summaries. To validate our RadCases dataset, we first had 3 independent U.S. attending physicians review a set of 50 true one-liners and confirm that they are representative of real-world patient case summaries used in clinical practice. We then computed the (1) Maximum and (2) Mean Similarity Score using the NV-Embed-v2 Retriever, between each of the RadCases datasets and a dataset of true one-liners derived from real patient cases. We also computed the average (3) Perplexity according to GPT-2 Large Medical; and (4) the average number of tokens per one-liner according to the GPT-4o tokenizer. We compare RadCases against other corpora such as arXiv computer science abstracts (**arXiv NLP**); Wikipedia articles (**Wikitext**); **PubMed** articles; and the **MedQA** dataset. Finally, we also compare against **Random** sentences admission notes in the MIMIC-IV dataset; random sentences from **Radiology** imaging reports in the MIMIC-IV dataset, **Full Admission Notes** from the MIMIC-IV dataset; and a separate **Test** set of extracted patient one-liners from the MIMIC-IV dataset. Each metric is reported as [Mean^{95% CI}], where [Mean] is the mean metric value, and [95% CI] is the 95% confidence interval. The best (resp., second best) values in each column—and all values with intersecting mean confidence intervals—are **bolded** (resp., underlined). Our results demonstrate that RadCases is a promising dataset of simulated patient one-liners compared with other domain-specific text corpora.

Based on both our quantitative experimental results and expert-in-the-loop data curation process, we believe that our RadCases dataset will be of use to the broader scientific community and future work on LLMs for clinical decision support.

- The datasets from USMLE, JAMA, and NEJM use only the first sentence of each patient case, which does not ensure completeness or quality. Prior work has shown that summarizing patient cases using more sophisticated models like GPT-4 has been effective [3-4]. Although the authors provide exclusion criteria for case selection, a more systematic quality assessment is necessary.

Please see our response to Reviewer 2, Comment 2 above. Specifically, we note that all one-liners in our RadCases dataset were individually reviewed to be representative of true clinical one-liners in written and used by clinicians in acute clinical settings by one U.S. attending radiologist and two U.S. medical students - we have updated the Methods section of our manuscript to make this clear to readership. To further evaluate our RadCases dataset, we also experimentally evaluated the curated one-liners in **Supplementary Table 2**. According to semantic similarity, perplexity, and token count metrics, our RadCases dataset is consistently similar to ground-truth, physician-verified one-liners than other scientific and medical textual data are.

- The dataset distribution is uneven, as shown in Supplementary Figure 1 and Table 1, which could introduce bias. Additional metrics like AUC and F1 should be included to provide a more comprehensive evaluation.

Thank you for this comment - we have included the F1 scores of the top performing language models in **Figure 5d** in our updated manuscript. Because our language models output a single predicted ACR AC Topic label and not a probability distribution over the topic space, we are unable to plot a meaningful AUROC curve.

3. The evaluation of LLMs on the RadCases dataset is limited. The authors selected six state-of-the-art general models but did not include medical-specific models such as Meditron [5], BiomedGPT [6], and Me-LLaMA [7]. Including specialized models would offer a more meaningful assessment of LLM performance in the medical domain.

[2] Hager, P., Jungmann, F., Holland, R., et al. Evaluation and mitigation of the limitations of large language models in clinical decision-making. *Nat Med* 30, 2613–2622 (2024).

<https://doi.org/10.1038/s41591-024-03097-1>

[3] Ma, Chong, et al. An Iterative Optimizing Framework for Radiology Report Summarization with ChatGPT. *IEEE Transactions on Artificial Intelligence* (2024).

[4] Goff, Daniel J., and Thomas W. Loehfelm. Automated radiology report summarization using an open-source natural language processing pipeline. *Journal of Digital Imaging* 31 (2018): 185-192.

[5] Chen, Zeming, et al. Meditron-70b: Scaling medical pretraining for large language models. *arXiv preprint arXiv:2311.16079* (2023).

[6] Zhang, Kai, et al. A generalist vision–language foundation model for diverse biomedical tasks. *Nature Medicine* (2024): 1-13.

[7] Xie, Qianqian, et al. Me LLaMA: Foundation large language models for medical applications. *arXiv preprint arXiv:2402.12749* (2024).

We thank the Reviewer for this comment. We have evaluated medical-specific language models such as Meditron, BioMedGPT, and Me-LLaMA on the RadCases and include the results of our evaluations in **Supplementary Figure 8** of our updated manuscript. All three medical-specific foundation models struggled on the RadCases dataset; notably, this finding where medical fine-tuned foundation models perform worse than their generalist counterparts has been known empirically within the Natural Language Processing community in recent years, and recently corroborated by Jeong et al. [1] and others [2,3]. We hypothesize that this could be due to the fact that these models are often fine-tuned on corpora of scientific literature from PubMed, and paragraph-form questions and patient cases. Because we focus on acute patient presentations where limited patient information is available, our RadCases dataset is significantly different from the text data used to fine-tune these medical models. We empirically validate this claim in **Supplementary Table 2**. We also include a copy of Supplementary Figure 8 here for easy reference:

Supplementary Figure 8: Evaluating Medical Foundation Models Fine-Tuned on Llama LLMs. Separate from the results presented in **Supplementary Figure 7**, an alternative approach to model fine-tuning is to instead leverage language models fine-tuned on large corpuses of domain-specific medical text. Examples of such foundation models include BioMedGPT-7B; MeLLaMA-70B; and Meditron-70B. We evaluate their accuracies on predicting correct ACR AC Topic labels for RadCases datasets; none of the three medical foundation models evaluated outperformed the base Meta Llama 3 70B model with statistical significance. These results highlight the challenge in fine-tuning language models specifically for RadCases task performance. Error bars represent \pm 95% CI over $n = 5$ independent experimental runs.

[1] Jeong DP et al. Proc EMNLP (2024). <https://arxiv.org/abs/2411.04118>

[2] Hager P et al. Nat Med (2024). <https://www.nature.com/articles/s41591-024-03097-1>

[3] Dorfner FJ et al. arXiv Preprint (2024). <https://arxiv.org/abs/2408.13833v1>

D. Appropriate Use of Statistics and Treatment of Uncertainties

The authors have appropriately applied statistical methods, with clear use of metrics and confidence intervals to support their conclusions.

E. Conclusions: Robustness, Validity, Reliability

The evaluation lacks robustness and reliability. The authors explore LLM performance in an acute clinical setting, but this is not clearly emphasized in the title or abstract. There is a notable gap between the one-liner summary data used in this study and real-world scenarios.

Additionally, using accuracy as the primary metric may be unreliable due to the dataset's uneven distribution. Specialized medical LLMs should be included in the evaluation, or the authors should provide justification for their exclusion.

Thank you for this comment. To clarify, the goal of our study is to specifically study LLM performance in the setting of limited patient information characteristic of patient presentations in acute care settings such as the emergency room. To better clarify this, we have updated the title of our manuscript to: "Evaluating Image Ordering for Acute Patient Presentations via Language Model Alignment with the ACR Appropriateness Criteria." We have also revised the abstract to more clearly emphasize our specific intended application context. We discuss this in further detail in our response to Reviewer 2, Comment G.

F. Suggested Improvements: Experiments, Data for Possible Revision

The evaluation of LLMs on the RadCases dataset is incomplete, focusing solely on general models. Specialized medical models should also be assessed. Furthermore, the core assumption underlying the experiments needs further justification. While the labeling process is clear, the rationale behind the initial data collection should be explained in greater detail.

Moreover, the experiments focus on limited patient information scenarios. The authors should include experiments with full datasets from USMLE, JAMA, and NEJM as mentioned in the manuscript.

We thank the Reviewer for raising this point. To clarify, the goal of our study is to specifically study LLM performance in the setting of limited patient information characteristic of patient presentations in acute care settings such as the emergency room. To better clarify this, we have updated the title of our manuscript to: "Evaluating Image Ordering for Acute Patient Presentations via Language Model Alignment with the ACR Appropriateness Criteria." We have also revised the abstract to more clearly emphasize our specific intended application context. We discuss this in further detail in our response to Reviewer 2, Comment G, and hope that the changes better clarify the purpose and motivation of our study.

G. Clarity and Context: Lucidity of Abstract/Summary, Appropriateness of Abstract, Introduction, and Conclusions

The authors aim to simulate decision-making contexts with limited patient information to test LLM performance. However, the title and abstract do not emphasize this specific application context clearly. The authors should either revise the abstract to reflect this focus or include experiments that cover scenarios beyond limited patient information, providing a more comprehensive evaluation.

We thank the Reviewer for raising this point. To clarify, the goal of our study is to specifically study LLM performance in the setting of limited patient information characteristic of patient presentations in acute care settings such as the emergency room. Multiple prior works have explored how LLMs may be leveraged as diagnostic tools when a more complete clinical picture of patients is available [1-4]; given this body of work, we specifically tackle the related but separate domain where patient information is limited. In accordance with the Reviewer's feedback, we have updated the title of our manuscript to: "Evaluating Image Ordering for Acute Patient Presentations via Language Model Alignment with the ACR Appropriateness Criteria." We have also revised the abstract to more clearly emphasize our specific intended application context (relevant revisions are underlined):

"Diagnostic imaging studies are an increasingly important component of the workup and management of acutely presenting patients. However, ordering appropriate imaging studies in the emergency room according to evidence-based medical guidelines is a challenging task with a high degree of variability between healthcare providers. To address this issue, recent work has investigated if generative AI and large language models can be leveraged to help clinicians order relevant imaging studies for patients. However, it is challenging to ensure that these tools are correctly aligned with medical guidelines, such as the American College of Radiology's Appropriateness Criteria (ACR AC), especially given the limited diagnostic information available in acute care settings. In this study, we introduce a framework to intelligently leverage language models by recommending imaging studies for patient cases that are aligned with evidence-based guidelines. We make available a novel dataset of patient case summaries to power our experiments, and optimize state-of-the-art language models to achieve an accuracy on par with clinicians in image ordering. Finally, we demonstrate that our language model-based pipeline can be used as intelligent assistants by clinicians to support image ordering workflows and improve the accuracy of acute image ordering according to the ACR AC. Our work demonstrates and validates a strategy to leverage AI-based software to improve trustworthy clinical decision making in alignment with expert evidence-based guidelines."

[1] Savage T et al. NPJ Digit Med (2024). <https://www.nature.com/articles/s41746-024-01010-1>

[2] Savage T et al. JAMIA (2024). <https://doi.org/10.1093/jamia/ocae254>

[3] Kim Y et al. ACL BioNLP (2024). <https://arxiv.org/abs/2406.06331v2>

[4] Omiye JA et al. NPJ Digit Med (2024). <https://www.nature.com/articles/s41746-023-00939-z>

Reviewer #2 (Remarks on code availability):

Yes, the code is accessible from the given link.

Reviewer #2 (Remarks on figshare data availability):

I did not see the real data. But, the authors provides the data preprocessing code, which looks fine to me.

We thank the Reviewer for their remarks. In general, it is challenging for us to include all the required constructed datasets to be able to easily allow the interested reader to reproduce our work using a single script. This is because many of the datasets we use cannot be released publicly with our work due to licensing restraints and/or data use and access requirements imposed by the original dataset providers. For example, in the paper that originally introduces the JAMA Clinical Challenges dataset, the authors do not make available the actual dataset due to JAMA license constraints. As a result, special steps must be taken as outlined in their implementation to gain access to the dataset. Similar constraints make it challenging to gain access to the NEJM and MIMIC-IV datasets as well. We detail the specific steps that interested readers must take to get setup to use our code in our README.md file associated with our repository, and also specifically describe the commands to run to reproduce our results once these initial setup steps are accomplished.

Reviewer #3 (Remarks to the Author):

This multi-component study describes the development of an LLM methodology for identifying appropriate image test orders based on medical chart information and then evaluates the methodology in several ways. I commend the authors for the extensive work.

We thank the Reviewer for their careful consideration of our manuscript and their insightful comments, which we believe have significantly improved the quality and impact of our work. Please find our responses to the Reviewer's major comments below.

Major Comments:

1. The introduction contains many emphatic statements that reflect preference/opinion more than fact. For example, "inappropriate imaging" implies that ordering providers are negligent in the exams they order. It is always easier to consider an exam to be unnecessary after-the-fact than in-the-moment. Furthermore, the citations listed in support of this statement are primarily opinion pieces and do not support the statement with data. Another example: "up to 50% of imaging studies performed in the ED every year are not clinically indicated..." That's a loaded assertion, the citation is for a cost-effectiveness analysis review manuscript, and the citation does not support the statement. I would argue that it is plausible to suggest that even more imaging would be clinically indicated if the resource is available and the goal is more rapid patient disposition or reduced liability or reduced diagnostic error. The introduction would benefit from substantial tempering of the language and by supporting statements with high quality data.

We thank the Reviewer for this insightful comment. We have significantly rewritten the Introduction and other relevant components of our manuscript in alignment with the Reviewer's

feedback to more appropriately highlight the challenges in ordering appropriate diagnostic imaging studies in real-world imaging workflows, as highlighted by numerous recent studies from emergency medicine physicians and others. We include the relevant excerpt from our updated Introduction below for easy access:

“Inappropriate ordering of diagnostic imaging studies is a commonly encountered problem in the emergency department (ED) and other acute-care settings.¹⁻⁴ While diagnostic imaging can play a crucial role in the acute workup of patients, inappropriate or unnecessary imaging studies present mounting concerns regarding resource utilization, radiation exposure, and financial burden to both patients and healthcare systems.⁵⁻⁷ Recent estimates suggest that up to 30% of diagnostic imaging studies ordered in the ED setting could be replaced with more appropriate alternatives.^{6,8}

Multiple factors contribute to this increasing trend of inappropriate imaging. Importantly, emergency medicine physicians often need to make rapid diagnostic decisions with limited clinical context while simultaneously managing high patient volumes and complex patient presentations. Furthermore, there is significant variability between healthcare providers in imaging ordering patterns: recent studies have documented significant differences in the utilization rates of different imaging studies between individual emergency physicians, suggesting that factors beyond pure clinical necessity influence imaging decisions.⁹⁻¹¹ To help clinicians make more informed, evidence-based decisions in image ordering and simultaneously address inter-provider variability in imaging practices, the American College of Radiology (ACR) released the ACR Appropriateness Criteria® (ACR AC), which are a set of evidence-based guidelines that assist referring physicians in ordering the most appropriate diagnostic imaging studies for specific clinical conditions.¹² As of its most recent online release, the ACR AC contains 263 unique imaging topics (i.e., patient scenarios).

However, despite the widespread availability of the ACR AC, improper imaging according to the guidelines remains a challenge in many emergency departments and inpatient settings.^{3,13} Bautista et al. showed that there is low utilization of the ACR AC by clinicians in practice: less than 1% of physicians interviewed in their study use the ACR AC as a first-line resource when ordering diagnostic imaging studies.^{14,15} The limited usage of the ACR AC may be partly due to how the Appropriateness Criteria are presented to clinicians; the evidence-based criteria are dense and can be difficult to parse through even for physician experts, and especially in acute healthcare settings such as the emergency department where decision making is both time-sensitive and critical.”

2. Many ACR Appropriateness Criteria, while termed “evidence-based”, are largely based on professional opinion informed by weak evidence. Some discussion would be welcome on how any CDS system can function well in the absence of high quality evidence.

We thank the Reviewer for this comment. In general, the role of clinical guidelines such as the ACR AC in clinical workflows can certainly be debated—especially in cases where there is limited evidence available. In our work, we assume that ACR AC incorporates expert consensus and the expertise of a panel of seasoned clinicians where high-quality evidence is lacking, and choose to leverage the ACR AC across all diagnostic Topics available in the ACR AC. In

situations where high quality evidence is absent, there are a number of possible solutions: for instance, we might flag the Topics with limited evidence in a user interface portal so that clinicians are aware that evidence is limited if the Topic is predicted by an LLM. Instead, we might also choose to completely eliminate these Topics from the database of possible predictions all together. We might also choose to develop a user interface to display condensed summaries of the guideline recommendations associated with all predicted Topics so that clinicians are able to synthesize the available information and contextualize recommendations for individual patient presentations. Given that these challenges are largely associated with how information is presented to clinicians in deployed applications of language models, we leave the task of determining the optimal strategy of adapting CDS systems to situations with limited high-quality evidence as future work.

3. “we investigate how state-of-the-art LLMs can be used as potential CDS tools to reduce the problem over-imaging according to the ACR AC.” -> The assumption of over imaging is largely philosophical.

We have updated the referenced statement to “We investigate how state-of-the-art LLMs can be used as potential CDS tools to reduce the problem of inappropriate image ordering according to the ACR AC.” Importantly, we also highlight that inappropriate image ordering and over-imaging is a well-documented and well-recognized issue in clinical medicine [1-6], and a focus point of many recent discussions by professional medical groups [6-11].

- [1] Kwee et al. Eur J Radiol (2024). <https://pubmed.ncbi.nlm.nih.gov/38820950/>
- [2] Baloescu. Annals of Emerg Med (2018). <https://pubmed.ncbi.nlm.nih.gov/30146444/>
- [3] Salerno et al. Radiol Med (2019). <https://pubmed.ncbi.nlm.nih.gov/30900132/>
- [4] Tung et al. Am J Emerg Med (2017). <https://pmc.ncbi.nlm.nih.gov/articles/PMC5815889/>
- [5] Kjelle et al. BMC Health Serv Res. (2021). <https://doi.org/10.1186/s12913-021-07004-z>
- [6] Sadigh et al. J Am Coll Radiol (2022). <https://pubmed.ncbi.nlm.nih.gov/35051412/>
- [7] Hendee et al. Radiology (2010). <https://pubs.rsna.org/doi/full/10.1148/radiol.10100063>
- [8] Mezrich. J Am Coll Radiol (2017). [https://www.jacr.org/article/S1546-1440\(17\)30677-4/fulltext](https://www.jacr.org/article/S1546-1440(17)30677-4/fulltext)
- [9] Lexa et al. J am Coll Radiol (2024). <https://doi.org/10.1016/j.jacr.2023.06.044>
- [10] R-SCAN Initiative. Accessed 2024 Nov 11. <https://edhub.ama-assn.org/acr-lifelong-learning/module/2729284>
- [11] Venkatesh et al. Am J Emerg Med (2021). <https://pubmed.ncbi.nlm.nih.gov/32014376/>

4. There is a lot of mixing of Methods and Results. For example, the entire “Constructing a benchmarking dataset...” subsection of the Results is descriptive of what was done and belongs in the Methods. This descriptive approach makes it challenging for the reader to parse the technical aspects of what was performed from what was actually learned. The manuscript would benefit from a very clear Methods section that carefully describes what was performed, step-by-step, in a way that they could be replicated by another group, and a much more abbreviated Results section that focuses on the generated data.

We thank the Reviewer for this comment. We have significantly updated the structure of our manuscript in accordance with this comment to improve its readability and overall flow.

5. It seems the primary result of interest (at least to me, and a bit buried) is: “Our results suggest that autonomous LLMs achieve comparable accuracy to that of clinicians in ordering diagnostic imaging: there was no statistically significant difference between the performance of Claude Sonnet-3.5 (accuracy 40.8%) and clinicians (accuracy 39.0%)”. So, from this statement, I am gathering that even the best LLM performed no better than the ordering providers, but no interpretation is provided so I’m not sure if I’m correct.

Based on feedback from additional Reviewers, we have included additional experiments in Figure 5 of our updated manuscript where we vary the maximum number of ACR AC Topics requested by an LLM agent in image ordering. Based on these additional experiments (specifically, when setting the maximum number of ACR AC Topics requested by an LLM to 1), we found that LLMs can achieve a dominant strategy with respect to clinicians in our retrospective study: for example, Llama 3 achieve a better accuracy and F1 score, and reduces the rate of missed imaging studies, without statistically significant increases in the rate of unnecessary imaging or the number of imaging studies ordered. We believe these new experimental results better support our conclusion that LLMs can serve as effective clinical decision support tools. We include the relevant updates to the revised manuscript below:

““““

Our results suggest that autonomous LLMs are effective tools in ordering diagnostic imaging: when LLMs were queried to return up to $m = 1$ relevant ACR AC Topics, Claude Sonnet-3.5 achieved a higher accuracy score of 58.0% and F1 score of 94.4% compared with clinicians (accuracy 39.3%; F1 score 92.5%) (McNemar test; $p = 0.044$). There was no statistically significant difference between Llama 3 (accuracy 61.5%; F1 score 92.9%) and clinicians (McNemar test; $p = 0.099$) for the same value of m . Across the patient cases assessed, clinicians ordered an average of 1.41 (95% CI: [1.28 – 1.53]) imaging studies per case; similarly, Claude Sonnet-3.5 ordered an average of 1.83 (95% CI: [1.60 – 2.06]) and Llama 3 an average of 1.54 (95% CI: [1.33 – 1.76]) studies per case. There was no statistically significant difference between the number of imaging studies ordered by Llama 3 and its clinician counterparts (two-sample paired t-test; Llama 3: $p = 0.269$).

We also evaluated the rates of both unnecessary and missed imaging studies. For $m = 1$, both Claude Sonnet-3.5 and Llama-3 were non-inferior to clinicians according to both metrics, achieving a false positive rates (FPR) of 6.90% and 6.90% (clinician FPR = 7.76%) (McNemar test; $p = 1.00$) and false negative rates (FNR) of 3.45% and 6.03% (clinician FNR = 6.03%) (McNemar test; Llama 3: $p = 1$; Claude Sonnet-3.5: $p = 0.549$), respectively (**Fig. 4b-c**). As expected, the rate of unnecessary imaging studies increased and the rate of missed imaging studies decreased as the maximum number of queried ACR AC Topics increased. Altogether, these results suggest that LLMs are promising tools for image ordering in clinical workflows.

””””

6. The LLMs triggered more imaging orders, not less, but throughout the introduction and discussion you talk about unnecessary imaging and waste. It seems your model would add to what you are trying to reduce.

Please see our response to Reviewer 3, Comment 5 above.

7. “We also evaluated the rates of both unnecessary and missed imaging studies.” I cannot find a description of the methods of how you defined or determined what studies were unnecessary or missed.

Thank you for this comment. We have updated our methods section with additional discussion on how each of our evaluation metrics were calculated. We include the relevant methods description in our updated manuscript below:

““““

Evaluation metrics of language models according to the ACR Appropriateness Criteria

In our experiments, we are interested in evaluation metrics that help us elucidate the performance of LLMs as clinical decision support tools. We detail these metrics and how they are calculated below.

An LLM’s *accuracy* is a score between 0 and 1. We evaluate two accuracy metrics in our experiments: Topic Accuracy (**Figs. 2-3**) and Imaging Accuracy (**Figs. 4-5**). For a given input patient case with ground truth ACR AC Topic y and model prediction y_{pred} , the Topic Accuracy is defined as the binary indicator variable equal to 1 if $y_{pred} = y$ and 0 otherwise. Separately, suppose that according to the ACR AC, the ground truth Topic y is associated with the set of clinically appropriate studies K , and the model-predicted Topic y_{pred} is associated with the set of clinically appropriate studies K_{pred} . The Imaging Accuracy is then defined as

$$\text{Imaging Accuracy} = |K_{pred} \cap K| / |K_{pred}|$$

Using the same notation as above, the rate of unnecessary imaging studies (i.e., false positive rate) ordered by an LLM is a score between 0 and 1 defined as the frequency of evaluated patient cases where (1) the ground truth set of appropriate studies K is identically equal to {No Imaging}; and (2) No Imaging is not a member of K_{pred} . Similarly, the rate of missed imaging studies (i.e., false negative rate) ordered by an LLM is a score between 0 and 1 defined as the frequency of evaluated patient cases where (1) No Imaging is not a member of K ; and (2) $K_{pred} = \{\text{No Imaging}\}$. Finally, the F1 score of an LLM is defined as $2 * TP / (2 * TP + (FP + FN))$, where TP is the number of patient cases where the LLM orders an imaging study that is clinically indicated, FP is the number of patient cases where the LLM orders an unnecessary study, and FN is the number of patient cases where the LLM inappropriately does not order an imaging study.

””””

8. First paragraph of the Discussion “we conduct both retrospective and prospective studies that demonstrate that LLMs such as Claude Sonnet-3.5 can be valuable tools for clinical decision support in acute care settings.” Your results that I outline in comments #5 and #6 do not support this statement.

Please see our response to Reviewer 3, Comment 5 above. Namely, based on feedback from additional Reviewers, we have included additional experiments in Figure 5 of our updated manuscript where we vary the maximum number of ACR AC Topics requested by an LLM agent in image ordering. Based on these additional experiments (specifically, when setting the maximum number of ACR AC Topics requested by an LLM to 1), we found that LLMs can indeed achieve a dominant strategy with respect to clinicians in our retrospective study: for example, Llama 3 achieve a better accuracy and F1 score, and reduces the rate of missed imaging studies, without statistically significant increases in the rate of unnecessary imaging or the number of imaging studies ordered. We believe these new experimental results better support our conclusion that LLMs can serve as effective clinical decision support tools.

9. “Importantly, we demonstrate how integrating evidence-based guidelines (i.e., the ACR AC) directly into the LLM-based inference pipeline can significantly improve the accuracy of clinical recommendations.” I cannot find a description of the methods of how you defined or calculated accuracy. I assume it’s in relation to the ACR AC. But even then, the ACR AC are guidelines and all guidelines have the caveat that provider discretion is needed, so it’s difficult to apply accuracy analyses in the context of guidelines adherence.

Please see our response to Reviewer 3, Comment 7 above.

Importantly, we also acknowledge the Reviewer’s point that the ACR AC and other evidence-based guidelines are ultimately recommendations that should be used in conjunction with clinical expertise to help physicians make the most appropriate decisions regarding the role of diagnostic imaging. We believe that an LLM-based clinical decision support tool—such as the ones that we evaluated in our work—should ultimately be used in the same fashion, where LLM-generated recommendations are used by clinicians together with their individual expertise to ultimately decide on the role of diagnostic imaging in specific patient scenarios. For this reason, we decided to focus on aligning language models solely with the ACR AC for the purposes of our study. We also note that in our retrospective study where language models were indeed used to autonomously order imaging studies for previous patients without a clinician-in-the-loop, our ground truth was dictated by an attending U.S. radiologists with access to the ACR AC, and not just the ACR AC alone.

We discuss this in detail in our revised Discussion section.

10. There’s no discussion about model drift. It’s great that it’s tested and (somewhat, albeit by medical students and residents) validated, but what happens as the model drifts over time?

We thank the Reviewer for this comment - indeed, model drift is an important phenomenon that can impact the quality of model predictions over time. While experimental evaluation of the impact of model drift is outside the scope of our work, we include additional discussion about model drift in our Discussion; the relevant updated paragraph is included below:

“...While this strategy allowed us to primarily focus on how existing LLMs perform according to the ACR AC, future work is warranted to investigate how these models handle scenarios where there is a lack of relevant guidelines. It is also important to ensure that LLM-based tools operate equitably across diverse patient populations and clinical scenarios before they are deployed across hospitals systems—as clinical medicine and associated evidenced-based guidelines evolve over time, it is important to recognize model drift and concomitant deterioration in the performance of LLMs may affect the accuracy and reliability of model recommendations. Future studies are warranted to investigate these problems and their real-world impact on model predictions—and also develop strategies to mitigate them.”

Response Letter

We sincerely appreciate that the reviewers find our work “interesting” and “important” in using language models to provide evidence-based recommendations for acute image ordering. We believe your comments were insightful and informative, and ultimately helped improve the quality of our work. Please find our point-by-point response to your comments below.

Reviewer #1 (Remarks to the Author):

The authors have extensively responded to the prior reviewers and updated the manuscript. While the concept is interesting, the authors’ responses have either 1) revealed limitations that are not adequately addressed in the manuscript and/or 2) not clearly or succinctly made some important elements of the study clear. Both the manuscript and the responses are quite long and that makes it difficult to evaluate. I suggest that the authors spend more time in the future producing more succinct and clear responses and a more linear manuscript.

We thank the Reviewer for their comments and careful consideration of our work, and appreciate the opportunity to better clarify and improve the quality of our manuscript. Please find our point-by-point responses to your comments below.

1. Prior Reviewer 1 Comment Re Line 142: You responded “our study is to evaluate the alignment of language model outputs with existing ACR AC guidelines, not to evaluate whether a language model can identify whether a set of guidelines is applicable or not” – If you do not evaluate whether the LLM can identify the applicable guidelines, how can you say whether the LLM could perform appropriately in a clinical scenario?

Thank you for this comment. We have revised the manuscript to investigate whether the LLM can identify the applicable guidelines as you recommended. To evaluate whether an LLM can identify applicable guidelines, we formulated the problem as a binary classification task: given an input patient one-liner and the set of existing ACR AC Topics, we asked an LLM to predict whether there exists at least one ACR AC Topic that is applicable to the patient. We conducted this additional experiment using the entire RadCases dataset, including patient one-liners that are not associated with a relevant ACR AC topic.

Our results are shown in **Supplementary Table 3**, included below for easy reference:

Claude Sonnet-3.5	Synthetic	USMLE	JAMA	NEJM	BIDMC
Balanced Accuracy	97.3 ^(96.1-98.5)	92.5 ^(91.5-93.4)	88.0 ^(87.7-88.3)	81.6 ^(80.3-83.0)	93.9 ^(92.2-95.6)
F ₁ Score	97.3 ^(96.2-98.4)	92.9 ^(91.6-94.3)	86.5 ^(86.1-86.9)	78.9 ^(76.5-81.4)	95.9 ^(94.5-97.3)
False Positive Rate	0.1 ^(0.0-0.2)	6.4 ^(5.2-7.6)	1.1 ^(0.2-1.9)	8.1 ^(6.5-9.7)	8.4 ^(7.0-9.7)

False Negative Rate	5.3 ^(3.1-7.5)	8.5 ^(6.7-10.2)	22.9 ^(22.4-23.4)	28.6 ^(27.0-30.2)	3.7 ^(1.5-5.9)
Llama 3	Synthetic	USMLE	JAMA	NEJM	BIDMC
Balanced Accuracy	99.0 ^(98.6-99.3)	94.9 ^(93.6-96.2)	92.2 ^(91.9-92.4)	80.2 ^(79.7-80.8)	87.4 ^(87.4-87.4)
F ₁ Score	99.0 ^(98.7-99.3)	95.6 ^(95.0-96.3)	92.5 ^(92.4-92.6)	77.9 ^(77.6-78.2)	87.6 ^(87.5-87.6)
False Positive Rate	0.0 ^(0.0-0.0)	7.3 ^(4.7-9.9)	3.8 ^(3.3-4.3)	11.0 ^(10.1-11.8)	9.1 ^(9.1-9.1)
False Negative Rate	2.1 ^(0.9-3.3)	3.4 ^(3.1-3.7)	11.6 ^(11.5-11.7)	28.6 ^(28.2-29.0)	16.1 ^(16.0-16.2)

In general, we find that both Claude Sonnet-3.5 and Meta Llama 3 are able to achieve promising balanced accuracy and F1 classification scores, which suggest that they are able to determine whether a set of guidelines is applicable or not. Nonetheless, we agree with the Reviewer that restricting our attention to patient cases with a ground-truth ACR AC Topic label is a potential limitation of our setting; we discuss this in future detail in our revised **Discussion** section: *“We also highlight that our main experimental results are reported on the subset of patient cases with at least one ACR AC Topic label. In general, patient one-liners with no matching ground-truth label were excluded from our analysis; see our **Materials and Methods** for additional details. In **Supplementary Table 3**, we investigate the ability of language models to predict whether there exists at least one ACR AC Topic that is relevant for a given patient as a binary classification task. Future work might explore multi-step pipelines involving sequential LLM queries that first determine if a set of ACR AC Topics is relevant before predicting the most relevant Topic within the set.”*

2. Prior Reviewer 1 Comment Re Fig 3: “Qualitatively, we found that although the Evidence-Based Baseline pipeline achieved a lower ACR AC Topic classification accuracy compared to the Evidence-Based Optimized inference strategy, its incorrect predictions were still closely related to the correct answer and underlying patient pathology. For example, a ground truth ACR AC Topic might be “Major Blunt Trauma” and the LLM prediction “Penetrating Torso Trauma;” although the LLM identified the incorrect ACR AC Topic label, both Topics warrant a “Radiography trauma series.” While you may get some form of appropriate answer, the incorrect matching will be a problem whenever ACR guidelines update and the irrelevant errors become relevant errors. Incorrect matching is a fundamental problem that is not getting enough attention in this manuscript and that indicates the conclusions are overstated.

Thank you for this comment. We acknowledge that incorrect Topic matching is a potential limitation of our work and highlight this in our Discussion section: *“Furthermore, closely related topics (e.g., “Major Blunt Trauma” and “Penetrating Torso Trauma”) often share clinical indications for the same set of imaging studies (e.g. “Radiography trauma series”). As result, our choice of imaging accuracy evaluation metrics in **Figures 4-5** still permit an LLM to predict the correct imaging study even if an incorrect ACR AC Topic was identified. This potential limitation of models achieving the “right answer through the wrong reasoning” is well-documented in prior*

*and concurrent work examining discrepancies between model reasoning traces and final predictions,^{34,66-70} and conceivably may degrade model performance if the ACR AC is updated such that the ground-truth and model-predicted ACR AC Topics no longer share the same imaging study. However, we highlight that even if we require our language models to obtain the “right answer through the right reasoning,” our inference strategy in **Figure 4a** still outperforms baseline reasoning strategies (i.e., the ACR AC Topic Prediction accuracy in **Figure 3b-c** is greater than the imaging accuracy of baseline LLMs in **Figure 4b**).”*

3. Prior Reviewer 3 Comment 1: I appreciate the revisions to the introduction with the additional evidence. However, “Inappropriate” is loaded with judgement and you do not have the information to pass that judgement post-hoc (i.e. you do not know the knowledge that the provider had or the situation that the provider was in at the time of the order). I recommend that all instances of “inappropriate” be shifted to the concept of “overuse” which is a health services term tied to medical waste. I think it’s still a reach.

Thank you for this comment. In accordance with this feedback and other feedback from reviewers, we have re-framed our contributions in our revised Introduction to instead focus on addressing the **cognitive burden** of diagnostic image ordering: in diagnosing and managing acutely presenting patients, LLMs can provide guideline-aligned recommendations as helpful suggestions for clinicians managing multiple complex patients simultaneously. We have removed all references to “inappropriate” imaging from our manuscript to better motivate our work.

4. Prior Reviewer 3 Comment 7 and 9: The explanation of the LLM is helpful. Is this LLM validated? I do not see an explanation of how the LLM result was tied to the authors checking the ground truth of whether ACR guidelines were actually followed. It seems the authors relied on the LLM exclusively. Some form of ground truth is needed or should be explicitly and clearly explained.

Thank you for this question. We apologize for the initial confusion; the LLM is only used to predict the ACR AC Topic given a patient one-liner. To determine the most appropriate imaging study after obtaining the LLM-generated ACR AC Topic prediction, we consult the ACR AC guidelines programmatically via web-scraping the ACR AC portal, which definitively provides us with the ground-truth imaging study recommendation given an ACR AC Topic. Importantly, we have verified that our use of the ACR AC for research purposes abides by the ACR Terms and Conditions.

We further clarify this in our revised **Materials and Methods** section: *“Importantly, we highlight that the construction of sets K and K_{pred} from the Topic labels y and y_{pred} are deterministically constructed and do not involve any LLM queries; instead, we use a custom Python (Python Software Foundation) web-scraping script with the Beautiful Soup (Leonard Richardson) open-source library to define each set of appropriate imaging studies for all Topics in the ACR AC from the URL <https://gravitas.acr.org/acportal>.”*

5. Prior Reviewer 3 Comment 8: These additional experiments are helpful and you have essentially performed a sensitivity analysis – sometimes the LLM is better and sometimes it is not. That information does not support a conclusion that they can be valuable. They might be. But you haven't proven it. You've at most shown that there is potential. And in combination with some of the comments above, the conclusions do not track with what is shown by the data.

Thank you for this comment. We have revised our manuscript to ensure not to overstate any conclusions. Notably, the purpose of the additional retrospective study experiments in **Figure 5** and **Supplementary Fig. 11** is simply to compare the performance of our LLM-based inference pipelines to that of clinicians. We have revised the manuscript to not incorrectly claim that these results “prove” that LLMs have clinical utility—in fact, we agree with the Reviewer and argue in our Discussion that substantial future work is needed to validate such a claim: “*Finally, we emphasize that our experiments, while promising, are no substitute for true prospective evaluation of language models as clinical decision support tools in real-world clinical workflows, such as those in the emergency department.*”⁷⁷ We particularly highlight that practical applications of our work might focus on targeting clinical decision making for costly imaging studies (e.g., magnetic resonance imaging) and those associated with relatively higher radiation doses (e.g., computed tomography). Future work is needed in close collaboration with ED physicians across a variety of clinical environments to truly validate the potential clinical utility of the LLM-based pipelines explored in this study.”

We apologize if our presentation of our previous version of **Figure 5** was confusing. To clarify, the point of the ablation study now included as **Supp. Fig. 11** is to demonstrate a potential trade-off with our language model inference strategy: namely, as we increase the number of predicted ACR AC Topics, we are less likely to mistakenly forgo image ordering when one is actually indicated, but at the expense of ordering additional (possibly unnecessary) imaging studies. Tuning the number of ACR AC Topics is analogous to tuning the threshold of a binary classifier to balance the tradeoff between false negative and false positive rates. Similarly, we can choose to limit the number of LLM-predicted ACR AC Topics to 1 to obtain more accurate predictions without sacrificing other performance metrics included in **Figure 5**. To this end, we have updated **Figure 5** in our original submission to highlight the results specifically when language models are prompted to propose up to a *single* ACR AC Topic that might be relevant for a patient. The original figure ablating the number of ACR AC Topics proposed by a language model has moved to **Supplementary Fig. 11**.

We hope this better clarifies the purpose of our study and our experimental results and also hope the revised conclusions are better phrased to align with our experimental results.

Reviewer #2 (Remarks to the Author):

The authors have adequately addressed the comments from Reviewers 1 and 2. The article is suitable for publication.

We thank and sincerely appreciate the Reviewer for their thoughtful feedback and consideration of our work.

Reviewer #3 (Remarks to the Author):

Comments to Authors:

Thank you for submitting this important work, which evaluates the potential of large language models (LLMs) to assist in diagnostic imaging study selection by mapping clinical scenarios to ACR Appropriateness Criteria (ACR AC). The integration of techniques like retrieval-augmented generation (RAG), in-context learning (ICL), and chain-of-thought prompting (COT) is a strong feature of the manuscript, and the inclusion of the novel RadCases dataset represents a valuable contribution to the literature. While the work is well-written and addresses a critical issue in emergency departments (EDs), there are areas that require further clarification, refinement, and discussion to strengthen the manuscript. Below, I provide specific comments, followed by general suggestions and strengths.

We thank the Reviewer for their careful review of our work, and appreciate their thoughtful comments and suggestions which we believe have enabled us to substantially improve both the quality and impact of our work. Please find our point-by-point responses to your comments below.

Specific Comments

Introduction

- ACR Scenarios and Imaging Topics:

1. The paper notes that the ACR AC includes 4000 clinical scenarios but focuses on 263 imaging topics. While this focus is acceptable, it represents a limitation that should be explicitly acknowledged. Additionally, ACR now lists 267 imaging topics; it would be prudent to verify this and update the manuscript accordingly. The grouping of scenarios (e.g., "cervical spine pain") into broader topics is understandable but should be discussed, as the ACR segregates these scenarios due to variations in imaging recommendations based on factors like initial versus subsequent imaging.

Thank you for the opportunity to clarify this point of confusion in our work. As of June 2024 (when our initial study was conducted), there were indeed 224 imaging topics - we have clarified this where relevant in our manuscript.

As of February 2025, there are now 267 total ACR AC topics - 233 of which are diagnostic imaging topics. To confirm that our experimental findings do not change significantly even as the ACR AC is updated over time, we relabelled the RadCases-Synthetic and RadCases-BIDMC datasets using the new set of 233 diagnostic topics, and re-evaluated both Meta Llama 3 and Claude Sonnet 3.5 on this more recent version of the ACR AC topics:

Our figure above plots the mean (\pm 95% confidence interval) accuracy of each model on ACR AC Topic prediction. Furthermore, only 8/176 (4.5%) of ground-truth labels in the Synthetic dataset and 6/241 (2.5%) of the ground-truth labels in the BIDMC dataset were modified due to the changes in the ACR AC from June 2024 to February 2025. Our results show that there was no statistically significant difference between in model performance depending on whether the June 2024 or February 2025 ACR AC was used. Importantly, our main findings have not changed significantly due to these recent changes in the ACR AC.

Regarding the grouping of ACR AC scenarios into topics, we thank the Reviewer for raising this point. We have better clarified this in our revised Discussion section: *“Secondly, we note that each of the ACR AC Topics considered in our study are also further stratified into “scenarios” for more nuanced imaging recommendations—for example, the ACR AC Topic “Breast Pain” can be decomposed in 4 related clinical “scenarios” (as of June 2024) each with their own imaging recommendations: (1) Female with clinically insignificant breast pain without other suspicious clinical finding, any age; (2) Female with clinically significant breast pain, age less than 30; (3) Female with clinically significant breast pain, age 30 to 39; and (4) Female with clinically significant breast pain, age greater than or equal to 40. Due to compute limitations and a restrictive LLM context length, we group all relevant scenarios into a single set of imaging recommendations for the single parent ACR AC Topic in our work; future work might investigate the performance of LLM-based inference pipelines for classification of individual ACR AC scenarios.”*

Figures and Results

- Figure 1:

2. The rationale for including Claude 3.5 in Figure 1 is unclear, as it appears to focus solely on RadCases performance. If this is due to prior peer-review feedback, the manuscript should explicitly address why only this model is shown and clarify its relevance in the current context. Notably, there is no reference to Figure 1 in the manuscript text.

Thank you for this comment. Our rationale for and reference to Figure 1 is included in page 6 of our original submission: *“Our core hypothesis is that while LLMs may struggle to directly recommend imaging studies for patients (a domain-specific task) (Fig. 1), ...”* In alignment with the Reviewer’s feedback and with prior peer-review feedback from other reviewers, we have revised this portion of our Introduction to better clarify both (1) the rationale for including Claude 3.5; and (2) how the results in Figure 1 help motivate our study: *“In Figure 1, we first motivate this problem by demonstrating how a state-of-the-art language model, such as Claude Sonnet-3.5, fails to accurately recommend diagnostic imaging studies that align with the ACR AC for a variety of input patient descriptions. Given these initial findings, we hypothesize that while LLMs may struggle to directly recommend imaging studies for patients (a domain-specific task), they may be able to accurately describe patient conditions and presentations (phrased as ACR AC Topics).^{47–49}”*

- Results Section:

3. Portions of the results section, particularly commentary on RadCases, would be more appropriately placed in the discussion. Additionally, content under "Evaluating language models as support tools for clinician diagnostic image ordering" includes elements better suited for the methods or discussion sections. For instance, hypotheses (e.g., “using language models to map patient scenarios to ACR AC Topics...”) belong in the introduction or discussion, not the results.

We thank the Reviewer for this suggestion to improve the clarity and organization of our work. In alignment with this feedback, we have moved the majority of the content initially under “Evaluating language models as support tools for clinician diagnostic image ordering” to “Formulating a strategy for LLM evaluation using the RadCases dataset” in the Materials and Methods section. Separately, we have removed the cited hypothesis (and other related hypotheses) from the Results section, and instead include them in the Introduction: *“Given these initial findings, we hypothesize that while LLMs may struggle to directly recommend imaging studies for patients (a domain-specific task), they may be able to accurately describe patient conditions and presentations (phrased as ACR AC Topics).^{47–49} ... Given a patient one-liner, we can then programmatically query the ACR AC based on the LLM-recommended topic category (without any explicit LLM usage) to determine the optimal imaging study for a patient. In this fashion, LLMs can be used to recommend diagnostic imaging studies according to recommendations from the guidelines.”*

- Ground Truthing and RadCases Labeling:

4. The example provided, “49M with HTN, IDDM, HLD, and 20 pack-year smoking hx p/w 4 mo hx SOB and non-productive cough,” is mislabeled as "Lung Cancer Screening." This patient is symptomatic and should receive a CT without contrast rather than screening, which is reserved for asymptomatic individuals aged 50–80 years with a 20-pack-year smoking history. This

highlights concerns about ground truthing and labeling. I suggest two or MDs involved, one of whom should be a subspecialist (e.g. in this case a thoracic radiologist or pulmonologist) to ensure accuracy. Since the dataset has more neurological cases, perhaps adding neuroradiologists and other subspecialists as needed.

Thank you for raising the point on the mislabelled example one-liner. Incidentally, this one-liner was indeed incorrectly annotated in an earlier draft of our work and revised by our authorship team, although the revision was accidentally missed and excluded from our final submission. We sincerely apologize for this oversight and have corrected this error in our manuscript.

Regarding the inclusion of additional MD experts for ground truth annotation, we thank the Reviewer for this suggestion. In alignment with this recommendation, we recruited Dr. Piya Saraiya as an additional member of our research and authorship team to help validate our labels for our dataset. Dr. Saraiya is a seasoned neuroradiologist at the University of Pennsylvania. We have received her help in further refining our constructed RadCases dataset, and have updated our reported experimental results where appropriate according to her expert ground-truth annotations. **Importantly, there were no significant changes to the results of our work after these changes were made.** With the addition of Dr. Saraiya, we are now privileged to have 2 US MD radiologists—with collectively over 35 years of clinical experience—approving of the construction and labelling of our RadCases dataset, which will be publicly released alongside our manuscript.

- Dataset Imbalance:

5. Neurologic topics dominate the RadCases dataset, followed by cardiac and gastrointestinal conditions. While this imbalance may reflect real-world ED visit patterns, it should be acknowledged and discussed as a potential limitation.

Thank you for raising this point. We have acknowledged this potential limitation in our revised Discussion section: *“This study also has its limitations. Firstly, because most of the patient descriptions in our RadCases dataset are derived from real medical sources, they also reflect inherent biases with respect to patient demographics and medical conditions. For example, we found that our dataset most commonly included ground-truth ACR AC Topic labels related to gastrointestinal, cardiac, and neurologic pathologies—while these cases may reflect real-world ED visit patterns, it remains to be seen how LLMs perform on other patient cases sampled from different underlying distributions, such as in rare disease diagnostics and low-resourced patient populations.”*

- ACR AC Topic Count:

6. The manuscript states there are 224 diagnostic imaging topics, but ACR lists 247, including 20 interventional radiology topics. Subtracting these leaves 227 diagnostic imaging topics. Please verify the count and update the manuscript as necessary.

Thank you for raising this point. We have updated our manuscript to clarify that the 224 diagnostic imaging topics used in our work are from the June 2024 version of the ACR AC. We refer the reviewer to our response to Reviewer 3, Comment 1 above for additional discussion.

• Figure 3:

7. While avoiding data leakage in ICL evaluations is commendable, reserving additional cases separate from training and validation datasets could allow for evaluation of ICL performance on RadCases-Synthetic. Consider augmenting the dataset for this purpose.

We thank the Reviewer for this suggestion. In accordance with this recommendation, we constructed a separate corpus of synthetically generated, annotated one-liners to retrieve from that was created using the identical prompting strategy as that for the RadCases-Synthetic dataset, except we used the Meta Llama 2 (7B) model (meta-llama/Llama-2-7b-chat-hf). This separate Llama 2-generated dataset was then used as the corpus of cases for ICL evaluation on the RadCases-Synthetic dataset, and also for model finetuning (MFT) of the base Llama 3 model to evaluate on the RadCases-Synthetic dataset. We have updated our Materials and Methods section in accordance with this change.

We have updated the experimental results in **Figure 3** and **Supplementary Figures 4-5** and **8** to include these new experimental results. Importantly, no major conclusions have changed as a result of these additional experiments.

• Figure 4:

8. The assumption that selecting the ACR AC Topic is equivalent to selecting the correct imaging study introduces limitations. Many topics encompass multiple scenarios with different imaging recommendations (e.g., "Right Upper Quadrant Pain" includes five clinical variants with differing recommendations). The authors should clarify whether the goal is to identify the most applicable clinical scenario or the general topic. Additionally, Figure 4 comparisons may be misleading, as baseline (LLM selecting the correct imaging study) is inherently different from evidence-based baselines and optimized approaches, which select the ACR AC Topic. Consider revising this figure or ensuring consistent evaluation criteria across methods.

Thank you for this comment. Regarding the distinction between ACR AC Topics and scenarios, we have clarified this in our revised Discussion section - please refer to our earlier response to Reviewer 3, Comment 1 above. To clarify explicitly, the goal of our study is to identify the most applicable ACR AC Topic.

Separately, thank you for your comment regarding Figure 4. To clarify, the metric being evaluated across all methods in Figure 4 is the imaging accuracy (resp., imaging FPR and FNR) as defined in the subsection "Evaluation metrics of language models according to the ACR Appropriateness Criteria". The selection of the ACR AC Topic by the evidence-based baseline and optimized approaches are only an intermediate step to the selection of a predicted imaging study to compute and report the imaging metrics in Figure 4. We also refer the Reviewer to our responses to Reviewer 1, Comments 2 and 4 above.

• Page 18, Second Paragraph:

9. The manuscript references Figure 4f, but this appears to be a typographical error as the content aligns with Figure 5f. Please correct this discrepancy.

We thank the Reviewer for catching this typographical error - we have corrected all instances of this discrepancy in our revised manuscript.

General Points for Discussion

One-Liner Summaries:

10. The reliance on one-liner summaries makes several assumptions, including that all relevant clinical details can be summarized in a single sentence. This may not always be the case, as certain scenarios require multiple lines for full context. The rationale for using one-liners should be elaborated. For example:

One-liners are appropriate when clinical records are unavailable and meaningful interaction with an LLM is needed for specific questions.

When full ED notes are available, having the AI analyze these records directly may be a more streamlined and clinically useful workflow than requiring ED physicians to synthesize one-liners.

Thank you for this comment. We have clarified our choice in using one-liners in our revised Introduction: *“Prior work has examined the ability of LLMs to rapidly process and contextualize large volumes of information, which could help transform the ACR Appropriateness Criteria into a more accessible, real-time clinical decision support tool... Other studies, including Savage et al.²⁰; Kim et al.³⁴; Jin et al.³⁵; and Krithara et al.³⁶, work with more realistic examples of real-world patient descriptions; **however, they also simultaneously (1) assume that all relevant medical information is provided to make a diagnostic decision; and (2) are phrased as multiple choice questions. Neither of these characteristics are representative of how clinicians might use LLMs for clinical decision support in practice, especially in acute emergency medicine settings that are notably characterized by the lack of a complete patient information.**”*

In particular, we note that many ED physicians draft one-liners while working up a new patient presentation even when there is limited patient information available to help organize differential diagnoses and communicate with other clinicians. **Because one-liners are reflective of real-world clinical communication and the limited patient information available in acute care settings, we believe they are a natural and realistic choice for inputs into clinical decision support tools in acute patient presentations that we study in our work.**

Real-World ED Integration:

11. The retrospective analysis demonstrates LLMs' ability to identify the correct ACR topic but does not fully address their capability to recommend appropriate imaging studies. Future work should include prospective evaluation in real-world ED workflows, engaging ED physicians to validate clinical utility.

Thank you for this suggestion. We agree with the Reviewer on the significance of this future work, and discuss this in our revised Discussion section as a (critically important) opportunity for future work: *“Finally, we emphasize that our experiments, while promising, are no substitute for true prospective evaluation of language models as clinical decision support tools in real-world clinical workflows, such as those in the emergency department.”*⁷⁷ We particularly highlight that *practical applications of our work might focus on targeting clinical decision making for costly imaging studies (e.g., magnetic resonance imaging) and those associated with relatively higher radiation doses (e.g., computed tomography). Future work is needed in close collaboration with ED physicians across a variety of clinical environments to truly validate the potential clinical utility of the LLM-based pipelines explored in this study.”*

Focus on High-Impact Imaging Studies:

12. It may be worthwhile to consider this in future work: Targeting high resource utilization studies (e.g., PE protocol CTs, head CTs, brain MRIs, spine MRIs)

Thank you for this suggestion. We have discussed this potential opportunity for our proposed work in our revised Discussion section - please see our response to the comment above for additional details.

Complex Scenarios:

13. Real-world ED cases often involve overlapping conditions and nuanced imaging needs. Additionally, prior imaging results often influence study selection. The current approach assumes ACR topic mapping is sufficient, which may oversimplify decision-making in clinical practice. These limitations should be discussed.

Thank you for raising this point. We agree with the Reviewer on this important limitation and discuss it in further detail in our revised Discussion section: *“Importantly, we also emphasize that even though we leverage the ACR AC as a ground truth symbol in our experiments (Fig. 2-3), evidence-based guidelines like the ACR AC are ultimately recommendations that should be used in conjunction with clinical expertise to help physicians make the most appropriate decisions regarding the role of diagnostic imaging. Such recommendations may therefore fall short in more challenging patient cases not considered in this work, such as those with multiple medical conditions, complex admissions, and/or prior imaging studies that can drastically affect the appropriateness of different diagnostic methods.”*

Combined Strategies:

14. Consider exploring the combined use of the best performing techniques (ICL + COT), or all techniques, as this could further improve performance.

Thank you for this suggestion. We combine ICL and COT in **Supplementary Fig. 7** of our revised manuscript (see a copy of the figure below). Interestingly, we observe that using the zero-shot prompting strategies together does not necessarily improve performance when compared to using ICL or COT individually for both Claude Sonnet-3.5 and Meta Llama 3.

Prospective Evaluation:

15. The inclusion of prospective evaluation strengthens the study. However, further details are needed:

The low performance of ER resident physicians is surprising. What was their year of training? Were the scenarios typical or atypical ED cases? If unrepresentative, this could bias the results. Including board-certified ED physicians in future evaluations would enhance generalizability.

Thank you for this comment. 6 out of the 7 ER resident physicians who participated in our study were in their first two years of residency training. We agree with the Reviewer that the observed performance of the study participants was initially surprising—in our conversations with ED physicians after the conclusion of our prospective study, we hypothesized that this is likely due to two reasons: (1) independent, appropriate ordering of diagnostic imaging studies is considered a challenging task even for senior ED residents; and (2) ED residents are not necessarily trained to order imaging studies that align with the ACR AC. This is particularly the case at a large academic institution such as the University of Pennsylvania, where the

benefit-to-cost ratio of obtaining more extensive imaging may be different than that dictated by the ACR AC. We have discussed this in detail in our revised Discussion section.

The scenarios included in our prospective study were classified as *typical* ED cases according to two U.S. attending physicians. Importantly, we also note that each case was annotated with a set of possible “correct” imaging studies—a participant answer was classified as correct if the answer was contained within this set. We have better clarified this in our revised Materials and Methods section:

“Constructing the patient cases for prospective user evaluation study

To enable our prospective evaluation of LLMs as clinical decision support tools for clinicians, we first constructed a separate dataset of 50 patient one-liners derived from the RadCases BIDMC one-liners. The initially redacted details such as patient name, age, or gender were manually replaced with fictitious name, age, and/or gender values. The cases were then reviewed and edited by three separate attending physicians to ensure that the cases were representative of typical real-world patient cases that commonly present in the emergency department.”

We agree with the Reviewer that including board-certified ED physicians in future evaluations is warranted and have discussed this in our revised Discussion section: *“That being said, we highlight that our study was limited by a relatively small sample size of only 23 medical students and 7 junior emergency resident physicians. Furthermore, our study participants voluntarily opted in to participate in our study and may not reflect the attitudes and behaviors of clinicians that may have a more conservative predisposition to the use of AI tools in healthcare. Finally, we highlight that ED residents at large academic institutions (such as the University of Pennsylvania where this study was conducted) may not currently be trained to order imaging studies in alignment with the ACR AC, as the benefit-to-cost ratio of obtaining more extensive imaging studies may be different institutionally than as dictated by national guidelines. Given these considerations, future work is warranted to better characterize the impact of these factors across diverse populations of healthcare workers as they affect real-world clinical workflows and physician thinking, ultimately ensuring that LLMs are used responsibly and can improve patient care.”*

Strengths

- o Overordering and inappropriate imaging are major challenges in EDs, and this research addresses a critical issue.
- o RadCases is a valuable contribution to the literature and provides a foundation for further research.
- o Employing RAG, ICL, COT, and model fine-tuning enhances the manuscript's impact and provides educational value to the audience.
- o The study represents an important first step toward integrating AI tools into clinical workflows, laying the groundwork for future innovations.

Reviewer #4 (Remarks to the Author):

This manuscript addresses an important clinical challenge—improving the accuracy of imaging ordering in acute settings by leveraging large language models (LLMs) aligned with evidence-based ACR Appropriateness Criteria. Overall well-written and study design is sound. However, overall novelty of the work appears modest compared to numerous prior publications on radiology appropriateness criteria.

In this regard, it is worth noting that the true novelty of the manuscript may lie in the introduction of the RadCases dataset itself. The authors might consider renaming the manuscript to emphasize the RadCases dataset and refocusing the writing on how this dataset, in combination with the chosen LLM optimization methods and clinical evaluation pipeline, advances current knowledge—rather than rehashing material on Appropriateness Criteria which has been extensively covered in previous work.

We thank the Reviewer for their thoughtful feedback and careful consideration of our work and believe that their insights have allowed us to greatly improve the quality of our manuscript. We appreciate that the Reviewer finds our work novel, well-written, and sound. Please find our point-by-point responses to your comments below.

Some comments below to help improve the paper:

General:

1. Given that many studies have already leveraged appropriateness criteria in radiology, it is essential that your manuscript clearly delineate what is new. Please provide a more comprehensive literature review that contrasts your approach with previous work, and explicitly state how the RadCases dataset, the chosen LLM optimization methods, or the clinical evaluation pipeline advance current knowledge. Emphasize that the primary novelty is the RadCases dataset itself, which could merit renaming the manuscript to highlight this contribution.

We thank the Reviewer for this comment, and have included a more comprehensive literature review in our revised Introduction to better differentiate our manuscript from prior work: *“Prior work has examined the ability of LLMs to rapidly process and contextualize large volumes of information, which could help transform the ACR Appropriateness Criteria into a more accessible, real-time clinical decision support tool. For example, Nazario-Johnson et al.²⁰ and Zaki et al.²¹ evaluate the alignment of LLMs with the ACR Appropriateness Criteria; however, both studies leverage inputs that are not representative of the vernacular used in real-world clinical workflows. Other studies, including Savage et al.²²; Kim et al.³⁹; Jin et al.⁴⁰; Rau et al.⁴¹; and Krithara et al.⁴², work with more realistic examples of real-world patient descriptions; however, these LLM inputs either (1) assume that all relevant medical information is provided to make a diagnostic decision; or (2) are phrased as multiple choice questions. Neither of these characteristics are representative of how clinicians might use LLMs for clinical decision support in practice, especially in acute emergency medicine settings that are notably characterized by*

the lack of a complete patient information. Finally, Liu et al.⁴³; Zhang et al.⁴⁴; Zhang et al.⁴⁵; Zhang et al.⁴⁶; and Singhal et al.⁴⁷ introduce a number of performant models for medical tasks; however, these models again assume access to a relatively complete picture of the patient’s clinical status and past medical history, which is rarely the case for acutely presenting patients in the emergency department.” Importantly, our inference pipeline to predict diagnostic imaging studies for input patient descriptions via intermediate ACR AC Topics is a key novel contribution that distinguishes our work from existing literature: by explicitly leveraging ACR AC Topics, we can significantly improve the imaging accuracy of generalist LLMs without any additional model training. This approach—while perhaps obvious in hindsight—is novel and an important additional contribution of our work, which we are able to empirically evaluate following the construction of the RadCases dataset. Furthermore, our approach is *scalable* and is compatible with hundreds of ACR AC Topics, and can also be readily adaptable in other medical domains, too. We highlight these primary contributions in the Discussion section of our revised manuscript.

Separately, we appreciate that the Reviewer highlights the novelty of our RadCases dataset, and hope that it will be a valuable contribution to the broader community. We have renamed our manuscript to “**Evaluating Acute Image Ordering for Real-World Patient Cases via Language Model Alignment with Radiological Guidelines**” to better highlight the contribution of our RadCases dataset.

Abstract:

2. The phrase “...on par with clinicians” would benefit from specifying the metric (e.g., “accuracy on par with clinician recommendations”).

Thank you for this suggestion. We have revised the sentence in our abstract to the following:

“We make available RadCases—a novel dataset of patient case summaries to power our experiments—and leverage our framework to enable state-of-the-art language models to achieve an accuracy on par with clinicians in image ordering.”

Results

3. p-values are reported for model comparisons using two-sample, one-tailed homoscedastic t-tests. Please clarify whether you adjusted for multiple comparisons (e.g., using a Bonferroni correction) and provide additional details regarding the statistical tests used.

Thank you for this comment. During a previous round of peer-review, we consulted with two statisticians at the University of Pennsylvania and determined that multiple comparisons corrections were not required in our work. This is because for all our experiments, either (1) the experiment was considered exploratory analysis of a secondary outcome (e.g., **Supp. Tables 8-10** and [1-2]); (2) a sufficiently small family of experiments was used to support any single pre-planned inference; and/or (3) the experiment followed similar setups in prior work where multiple comparisons correction was not needed (e.g., [3-5]).

Regarding additional details for the statistical tests used, we refer the Reviewer to the “Experimental evaluation and statistical analysis” subsection of our Materials and Methods section: *“Each experiment was run using 5 random seeds, and we computed the mean accuracy of each method with 95% confidence intervals (CIs) against the human-annotated ground truth labels. A p-value of $p < 0.05$ was used as the threshold for statistical significance. In all figures, “n.s.” represents not significant (i.e., $p \geq 0.05$); a single asterisk $p < 0.05$; double asterisks $p < 0.01$, and triple asterisks $p < 0.001$. All statistical analyses were performed using Python software, version 3.10.13 (Python Software Foundation), the SciPy package, version 1.14.0 (Enthought),⁷⁴ and the PyFixest package, version 0.24.2.⁹⁹”*

- [1] Li G et al. An introduction to multiplicity issues in clinical trials: The what, why, when and how. *Int J Epidemiol* 46(2): 746-55. (2017). doi: 10.1093/ije/dyw320
- [2] Zarbock A, et al. Effect of regional citrate anticoagulation vs systemic heparin anticoagulation during continuous kidney replacement therapy on dialysis filter life span and mortality among critically ill patients with acute kidney injury: A randomized clinical trial. *JAMA* 324(16):1629-39. (2020). doi: 10.1001/jama.2020.18618
- [3] Goh E et al. GPT-4 assistance for improvement of physician performance on patient care tasks: a randomized controlled trial. *Nat Med* 31: 1233-8. (2025). doi: 10.1038/s41591-024-03456-y
- [4] Singhal K et al. Large language models encode clinical knowledge. *Nature* 620: 172-80. (2023). doi: 10.1038/s41586-023-06291-2
- [5] Zhang K et al. A generalist vision-language foundation model for diverse biomedical tasks. *Nat Med* 30: 3129-41. (2024). doi: 10.1038/s41591-024-03185-2

Materials and Methods

4. Page 24, 2nd paragraph RadCases dataset is described as leveraged from five different sources. Please specify the exact number of cases contributed by each source and detail the inclusion/exclusion criteria (e.g., “excluded if the one-liner did not refer to a specific patient presentation”).

Thank you for this comment. We have revised our description of the RadCases dataset in the “Formulating a strategy for LLM evaluation using the RadCases dataset” subsection to specify the exact number of cases from each source:

“Our curated RadCases dataset consists of 1,599 patient one-liner scenarios constructed from five different sources representing a diverse panel of patient presentations and clinical scenarios: (1) RadCases-Synthetic (156 out of the 1,599 total patient cases); (2) RadCases-USMLE (170 patient cases); (3) RadCases-JAMA (965 patient cases); (4) RadCases-NEJM (163 patient cases); and (5) RadCases-BIDMC (145 patient cases).”

We have also included a breakdown of the number of excluded cases by each exclusion criteria in our revised “Constructing a benchmarking dataset for image ordering using language models” subsection:

“A patient one-liner was excluded from evaluation if any of the following exclusionary criteria applied: (1) the ACR AC did not provide any guidance for the chief complaint (e.g., a primary dermatologic condition); (2) an appropriate imaging study was performed and/or a diagnosis was already made; (3) the one-liner did not include sufficient information about the patient; or (4) the one-liner did not refer to a specific patient presentation (e.g., one-liners extracted from epidemiology-related USMLE practice questions). Of the original 2,513 patient cases, a total 914 (36.3%) cases were excluded due to the above criteria (719 excluded cases due to criteria (1); 90 due to criteria (2); 49 due to criteria (3); and 56 due to criteria (4)); see **Supplementary Table 12** for additional details.”

We also include a more detailed breakdown of the prevalence of exclusion criteria within each of the individual RadCases datasets in **Supplementary Table 12**, included below for easy reference:

RadCases Dataset	(1) No ACR AC Guidance	(2) Imaging or Diagnosis Already Performed	(3) Insufficient Information	(4) No Specific Patient Presentation	Total Number of Excluded Cases
Synthetic	19	0	1	0	20
USMLE	57	2	33	39	131
JAMA	522	3	5	8	538
NEJM	45	76	1	1	123
BIDMC	76	9	9	8	102
Total Number of Excluded Cases	719	90	49	56	914

5. Page 26, in the section “Labelling one-liners by ACR Appropriateness Criteria topics”, there is a note that two fourth-year medical students annotated the data under supervision. It would strengthen the manuscript to include inter-rater reliability metrics or a brief description of how disagreements were resolved (e.g., “In cases of disagreement, the attending radiologist’s decision was final”).

Thank you for this comment. We have added the following clarifying statement in the referenced section: “*The two medical students and radiologists discussed cases where there was disagreement between proposed annotations, and the attending radiologists’ decision was final.*”

6. page 27, the description of the chain-of-thought (COT) prompting strategy is somewhat opaque. Including a concrete example of an optimized prompt with the accompanying chain-of-thought would greatly enhance clarity.

Thank you for this comment. We include all prompts used for our experimental results in **Appendix B** of our supplementary material. In particular, the COT prompts can be found on pages S26 and S27 in our supplementary material. We also include examples of the reasoning traces for different COT reasoning strategies in our revised **Appendix C** of our supplementary material.

7. page 29, LoRA and QLoRA fine-tuning parameters are mentioned. Please justify the chosen parameters (e.g., rank and α scaling values) and, if possible, reference related prior work (such as <https://arxiv.org/abs/2106.09685>; <https://pubmed.ncbi.nlm.nih.gov/39873598/>) to support these decisions.

The values of our rank and α hyperparameters were experimentally determined according to a hyperparameter grid search over the rank and α hyperparameters, logarithmically ranging from 8 to 512 (resp., 1 to 512), that maximize the accuracy of the fine-tuned model on a synthetic validation dataset. These wide ranges of hyperparameters were chosen in accordance with prior work (e.g., Christophe et al., Yang et al., Van Veen et al.) and documented expert recommendations (i.e., Hugging Face, Lightning AI) in addition to our own experiences with model fine-tuning. We have clarified this in our revised Materials and Methods section.

Figures and tables

8. Table 1 lists four common failure modes. This is not clearly defined as "Hallucination" may not be an accurate description of how or why it is failing (retrieval of information) and overall described vaguely. Please refer to established literature for definitions of errors (for example: <https://hdsr.mitpress.mit.edu/pub/1yo82mqa/release/2>) and consider adding frequency of such errors to help quantify their impact, with clearer tie to any mitigating steps.

Thank you for this suggestion and sharing the helpful reference. We have revised the error titles and descriptions in Table 1 to better align with existing work and more precisely describe the observed failure modes. We also analyzed 100 cases where Claude Sonnet-3.5 made an inaccurate prediction across all 5 RadCases datasets, and reported their frequency in our revised Table 1.

9. Figure 5 has a good caption, however is not clearly discussed/described in the accompanying text. There is retrospective comparison of clinicians' imaging orders with those generated by the LLM. While performance metrics (accuracy, F1, FPR, FNR) are provided, please expand on the clinical significance of these differences. For instance, discuss whether a 10–20% improvement in imaging accuracy is likely to translate into measurable benefits for patient outcomes or workflow efficiency.

Thank you for this suggestion. The clinical significance of the improvements we observed is challenging to extrapolate from our relatively small study. This is especially the case given that our study participants opted in to participate in the study (see our response to the next comment). That being said, according to recent estimates from radiological societies and systematic reviews in the literature, a 10%-20% improvement in imaging accuracy "*can*

potentially translate to hundreds of dollars saved per patient in reducing low-value and unnecessary imaging studies according to recent work.^{62–65} as discussed in our revised Discussion section.

10. The prospective study included 23 medical students and 7 resident physicians—a relatively small sample. Please discuss the limitations of this sample size and its implications for the generalizability of the findings. Also, provide a rationale for the chosen monetary incentives and comment on any potential bias introduced by the recruitment strategy.

Thank you for this comment. We discuss the limitations of our initial prospective study in our revised Discussion section:

“That being said, we highlight that our study was limited by a relatively small sample size of only 23 medical students and 7 junior emergency resident physicians. Furthermore, our study participants voluntarily opted in to participate in our study and may not reflect the attitudes and behaviors of clinicians that may have a more conservative predisposition to the use of AI tools in healthcare.”

Separately, we explain the rationale behind our chosen monetary incentives in our revised Materials and Methods section:

“Following prior work,⁹⁴⁻⁹⁷ we chose to offer this monetary compensation to improve recruitment rates and increase the diversity of opt-in participants, especially given the fact that our study posed minimal risk to the participants.”

Discussion

11. Page 22 There is mention of privacy concerns related to proprietary APIs. Consider elaborating on any safeguards or mitigation strategies applied in your experiments.

Thank you for this comment. Privacy concerns related to the use of proprietary APIs are important, and anecdotally affect AI usage policies at the authors’ home institution. In our study, we attempt to mitigate this by (1) reporting the performance of open-source models like Llama 3 in addition to proprietary models; and (2) not including patient name or ethnicity in input queries to LLMs. Future work might explore additional, more complex mitigation strategies - we have revised our Discussion to include these points.

In summary, while your study presents an innovative framework and promising preliminary results, at a minimum major revisions are needed to improve methodological transparency, clarity of presentation, and to clearly articulate the novelty and clinical impact relative to existing work. In particular, the manuscript would benefit from a refocusing on the RadCases dataset as the novel contribution. Addressing these points will substantially strengthen your manuscript.

We hope that we have adequately addressed your concerns above, and would be happy to answer any remaining questions or comments.

Response to Reviewers

Reviewer #3 (Remarks to the Author):

Thank you for addressing my prior comments. I appreciate the effort made to clarify the paper, improve the methods, and refine the discussion/conclusions. The additions, including expanded results and discussion of ACR AC topic structure/guideline changes, and validation of labeling with clinical experts further strengthen the manuscript. I do not have any additional comments.

We thank Reviewer #3 and the other referees for their feedback and thoughtful consideration of our work throughout the peer review process!